# A foundation machine learning potential with polarizable long-range interactions for materials modelling

Rongzhi Gao [1], ChiYung Yam [2,3], Jianjun Mao[2], Shuguang Chen[2,4], GuanHua Chen [1,2] ✉ & Ziyang Hu [1,2] ✉

Long-range interactions are essential determinants of chemical system behavior across diverse environments. We present a foundation framework that integrates explicit polarizable long-range physics with an equivariant graph neural network potential. It employs a physically motivated polarizable charge equilibration scheme that directly optimizes electrostatic interaction energies rather than partial charges. The foundation model, trained across the periodic table up to Pu, demonstrates strong performance across key materials modeling challenges. It effectively captures long-range interactions that are challenging for traditional message-passing mechanisms and accurately reproduces polarization effects under external electric fields. We have applied the model to mechanical properties, ionic diffusivity in solid-state electrolytes, ferroelectric phase transitions, and reactive dynamics at electrode-electrolyte interfaces, highlighting the model's capacity to balance accuracy and computational efficiency. Furthermore, we show that as a foundation model, it can be efficiently finetuned to achieve high-level accuracy for specific challenging systems.

Molecular dynamics (MD) simulations are indispensable in describing phenomena at the atomic level in disciplines such as chemistry, materials science, and biology[1–4]. While ab initio MD (AIMD)[5] simulation approaches offer unparalleled accuracy, their computational demands constrain their application to small timescales and length scales. On the other hand, classical MD (CMD)[6,7] simulations offer computational efficiency, but their accuracy is contingent upon empirical parameters. Machine learning interatomic potentials (MLIPs)[8–11] provide a solution that balances accuracy and computational efficiency. In particular, MLIPs based on equivariant graph neural networks, such as NequIP[12], DimeNet[13], and MACE[14], achieve excellent performance by introducing equivariant and invariant symmetries. Moreover, universal foundation models trained on the periodic table, such as M3GNet[15], CHGNet[16], GNoME[17], MACE-MP-0[18], MatterSim[19],

SevenNet[20], and ORB[21] have emerged, showing remarkable prospects for materials discovery.

While existing local MLIPs with a cut-off of around 5 Å perform well in simulating interactions within localized chemical environments, they may fail to capture long-range phenomena. This hinders their ability to understand and elucidate the behaviors of complex materials[22]. One approach is to implicitly incorporate the effects of long-range interactions through message-passing mechanisms[23], which can extend the cut-off through layer-wise propagation. However, if certain parts of the system are disconnected on the graph, such as two molecules separated by a distance beyond the cut-off value, the message-passing scheme becomes ineffective[24]. A promising direction involves investigating the potential advantages of explicitly including long-range interactions in the model formulation.

[1]Department of Chemistry, The University of Hong Kong, Pokfulam, Hong Kong SAR, China. [2]Hong Kong Quantum AI Lab Limited, Pak Shek Kok, Hong Kong SAR, China. [3]Shenzhen Institute for Advanced Study, University of Electronic Science and Technology of China, Shenzhen, China. [4]MattVerse Limited, Pak Shek Kok, Hong Kong SAR, China. ✉e-mail: ghc@everest.hku.hk; hzy@yangtze.hku.hk

These behaviors encompass electrostatic interactions, involving forces between charged particles, and dispersion terms, which arise from dynamic fluctuations in electron distribution within molecules or atoms.

Various explicit models have been proposed to address the challenges associated with long-range interactions by including electrostatic interactions. In these methods, charges are usually obtained through charge equilibration (QEq)[25] principles, including the Charge Equilibration via Neural Network Technique (CENT) method[26], Fourth-generation High-Dimensional Neural Network Potential (4G-HDNNP)[27,28], and Charge Equilibration Layer for Long-range Interactions (CELLI)[29] method. Generally, electrostatic parameters were trained to minimize the difference between QEq charges and reference partial charges derived from quantum mechanics (QM) calculations. Notably, partial charges obtained from QM calculations are either derived from the direct partitioning of the molecular wave function into atomic contributions or calculated based on the analysis of physical observables. However, due to the incompleteness of basis sets and variances in the partitioning methods, properties calculated based on partial charges may not always be reliable[16,30]. The ambiguity in DFT-assigned charges suggests that directly learning of them may be inessential for constructing accurate interatomic potentials[26,31,32]. On the contrary, the polarizable charge equilibration (PQEq)[30,33–36] method proposed by Naserifar et al. enhances the QEq method by using QM electrostatic interaction energies instead of partial charges as targets for evaluating interatomic potentials. Additionally, PQEq has proven to be effective in capturing polarization effects by explicitly introducing core-shell displacement. Another way to solve the partition-dependent issue is to take the dipole moment into the loss function, as done by SpookyNet[37]. However, it still uses point-like

charges, making it behaves like QEq-based methods when dealing with polarization effects.

In this work, we introduce a framework that integrates the equivariant message-passing neural network potentials with the explicit polarizable long-range potential. We initially conducted benchmark tests on datasets containing systems with various charge states proposed by references[28,38], showcasing the efficacy of our framework in handling both long-range interactions and varying charge environments. Building on these results, we extended the framework to develop a foundation model encompassing the periodic table up to Pu, while maintaining computational efficiency. We have applied this foundation model across different areas of applications, including the predictions of mechanical properties of materials, phase transitions in ferroelectric materials, and reactive molecular dynamics of solid-state batteries interface, thus leading to significant advancements in the fields of materials science and chemical simulations. The model also demonstrates transferability, accurately predicting interactions between clusters in various charge states and molecular polarizability. Furthermore, we show that the foundation model can be efficiently finetuned to achieve ab initio accuracy for specific challenging systems.

## Results
### Theoretical framework

Our framework, which integrates neural network potential and polarizable long-range interactions, is depicted in Fig. 1. For a given chemical system and boundary conditions, our goal is to construct a mapping from atomic coordinates $\mathbf{r}_i$ and atomic types $Z_i$ to the total potential energy $E_{\text{pot}}$. The potential energy is expressed as the sum of the second-order expansion[30] with respect to charge fluctuations and the density functional theory[39] D3 (DFT-D3) van der Waals dispersion energies correction

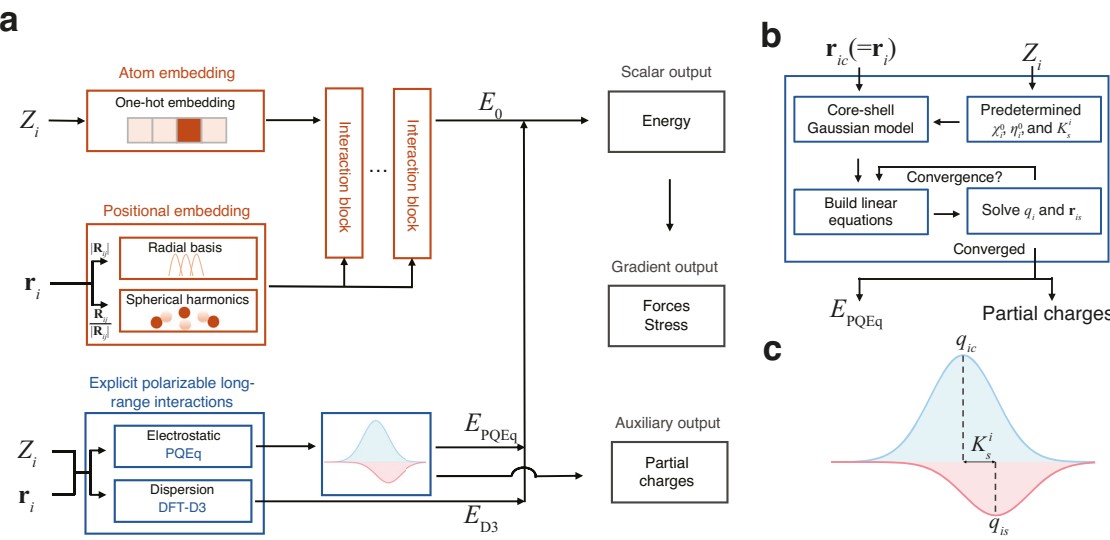

**Fig. 1 | Overview of the framework. a** The framework takes atomic numbers ($Z_i$) and coordinates ($\mathbf{r}_i$) as inputs and integrates a neural network block and an explicit polarizable long-range interactions block. For the neural network block, atomic numbers are converted into feature vectors via one-hot encoding, while the coordinates are transformed into a representation of the local environment using radial basis functions and spherical harmonics to capture the distance ($|\mathbf{R}_{ij}|$) and directional ($\mathbf{R}_{ij}/|\mathbf{R}_{ij}|$) information. These embeddings are processed through an equivariant graph neural network to output the scalar machine learning potential energy $E_0$. Explicit, physically-grounded calculations are performed for long-range effects. An electrostatic energy component ($E_{\text{PQEq}}$) is calculated using the polarizable charge equilibration (PQEq) method, and a dispersion energy component ($E_{\text{D3}}$) is calculated using the DFT-D3 method. The total energy is the sum of these

components. Gradients of the energy yield forces and stress on the system. The long-range block also provides partial charges as an auxiliary output. **b** A flowchart detailing the iterative self-consistent procedure for the PQEq method. Starting with atomic core positions ($\mathbf{r}_{ic}$), atomic numbers ($Z_i$), and predetermined atomic parameters (electronegativities $\chi_i^0$, chemical hardness $\eta_i^0$, and spring constant $K_s^i$), the method uses a core-shell Gaussian model to build and solve a system of linear equations. This loop is repeated until the partial charges ($q_i$) and shell positions ($\mathbf{r}_{is}$) converge, yielding the final electrostatic energy ($E_{\text{PQEq}}$) and the converged partial charges. **c** Partition of core-shell Gaussian charge model used in long-range electrostatics. The partial charge ($q_i$) of each atom is represented as the sum of the core ($q_{ic}$) and shell ($q_{is}$) charge. The harmonic spring constant, $K_s^i$, couples the core and shell.

**Table 1 | A comparison of root-mean-square test error metrics across different charge-state datasets[28,38] was performed among 4G-HDNNP[28], Marfu's NequIP[38], and our work**

| Dataset | | 4G-HDNNP[28] | Maruf's NequIP[38] | This work |
|---|---|---|---|---|
| $C_{10}H_2/C_{10}H_3^+$ | Energy (meV·atom$^{-1}$) | 1.19 | 1.33 | **0.44** |
| | Force (eV·Å$^{-1}$) | 0.078 | 0.071 | **0.023** |
| $Na_{8/9}Cl_8^+$ | Energy (meV·atom$^{-1}$) | 0.48 | N. A. | **0.16** |
| | Force (eV·Å$^{-1}$) | 0.033 | N. A. | **0.005** |
| $Ag_3^{+/-}$ | Energy (meV·atom$^{-1}$) | **1.32** | 498.39 | 4.87 |
| | Force (eV·Å$^{-1}$) | 0.032 | 2.145 | **0.028** |
| $Au_2$-MgO | Energy (meV·atom$^{-1}$) | 0.22 | 1.03 | **0.14** |
| | Force (eV·Å$^{-1}$) | 0.066 | 0.034 | **0.021** |
| BTA-Cu | Energy (meV·atom$^{-1}$) | N. A. | 0.48 | **0.30** |
| | Force (eV·Å$^{-1}$) | N. A. | 0.008 | **0.004** |
| BTA ($H_2O$)-Cu | Energy (meV·atom$^{-1}$) | N. A. | 0.71 | **0.19** |
| | Force (eV·Å$^{-1}$) | N. A. | 0.010 | **0.005** |

Best results in bold.

term[40,41] $E_{D3}$,

$$E_{pot} = \sum_i \left( E_0^i(\mathbf{r}_i, Z_i) + \chi_i^0 q_i + \frac{1}{2}\eta_i^0 q_i^2 + \frac{1}{2}K_s^i|\mathbf{r}_{ic} - \mathbf{r}_{is}|^2 \right)$$
$$+ \sum_{ik>jl} C_{ik,jl}\left(\mathbf{r}_{ik,jl}\right)q_{ik}q_{jl} + E_{D3} \qquad (1)$$
$$= \sum_i E_0^i + E_{PQEq} + E_{D3}.$$

The zeroth-order atomic energy $E_0^i$ corresponds to the final layer scalar output of the equivariant graph neural networks, while the higher-order self- and interatomic Coulomb interactions represent the polarizable long-range electrostatic interactions $E_{PQEq}$. To account for charge transfer and polarization effects, the partial atomic charge $q_i$ of atom $i$ is the sum of the nuclear core charge $q_{ic}$ and shell charge $q_{is}$, both of which assume a Gaussian charge distribution form. The first-order coefficients $\chi^0$ are electronegativities, commonly defined as half of the sum of ionization potential (IP) and electron affinities (EA). The second-order coefficients $\eta^0$ signifies idempotential or chemical hardness, defined as IP − EA. The spring constant $K_s^i$ denotes the isotropic harmonic connectivity between the shell position $\mathbf{r}_{is}$ and core position $\mathbf{r}_{ic}$ (equal to $\mathbf{r}_i$) of atom $i$. The Gaussian electrostatic energy is given by $C\left(\mathbf{r}_{ik,jl}\right)q_{ik}q_{jl}$, where $i$ and $j$ are the atomic indices, and $k$ and $l$ represent the core ($c$) or shell ($s$), respectively. The displacement vector is given by $\mathbf{r}_{ik,jl} = \mathbf{r}_{ik} - \mathbf{r}_{jl}$. The detailed derivation of the response to the external electric field can be found in the Supplementary Note 1. In previous works, PQEq parameters for 102 elements have been derived from experimental data or high-level QM calculations[25,42], and have accurately reproduced QM electrostatic interaction energies[30]. In the equivariant neural networks[12], the initial features are generated using a trainable one-hot embedding that operates on the atomic types. The interatomic distance of atom $i$ and atom $j$, denoted as $|\mathbf{R}_{ij}|$, is expanded by radial basis functions. Concurrently, the directional component of the interatomic vectors, expressed as $\mathbf{R}_{ij}/|\mathbf{R}_{ij}|$, is expanded by spherical harmonics functions. These features are utilized to construct the atomic graph. Then, through the equivariant message passing[23] schemes, the atomic features are updated. A multilayer perceptron layer is then connected to derive the scalar output. It should be noted that the neural network potential component is specifically designed to capture energy contributions distinct from electrostatic interactions, thereby ensuring no overlap in energy accounting between different components. Subject to the conservation of the net charge and the equality of chemical potentials for all atoms, the linear equations can be used to update

partial charges ($q_i$) and shell positions ($\mathbf{r}_{is}$). The partial charges can be obtained self-consistently in each MD step. The forces on atoms and virial stress on the cell can be calculated via automatic differentiation after partial charges are determined, as detailed in Supplementary Note 2.

## Validation on diverse charge-state datasets

To evaluate the capability of our framework in capturing different charge states and long-range interactions, we validated our framework against the dataset developed by Ko et al.[28] and Maruf et al.[38], which encompass various charge states and charge transfer systems. They contain six distinct subsets: Ag cluster with positive and negative total charge, ($Ag_3^{+/-}$), Na-Cl ionic cluster with one neutral Na removed ($Na_{8/9}Cl_8^+$), hydrogenated carbon chains in both neutral and cationic states ($C_{10}H_2/C_{10}H_3^+$), a periodic system consisting of Au clusters adsorbed on an MgO-(001) surface, and benzotriazole interactions with Cu-(111) surfaces in dry (BTA-Cu) and aqueous (BTA ($H_2O$)-Cu) environments. We demonstrate the advantage of integrating equivariant message-passing networks with explicitly polarizable long-range interaction models by benchmarking against both 4G-HDNNP, a local MLIP with explicit long-range interactions, and NequIP, which implicitly captures long-range effects through its message-passing mechanism. 4G-HDNNP differentiates charged systems by explicitly training DFT partial charges and incorporating them as descriptors into neural networks for short-range interactions. We employed the NequIP models trained in the reference[38] (Maruf's NequIP) as our baseline equivariant model. Our framework explicitly incorporates long-range interactions while adopts Maruf's NequIP as the $E_0$ component without touching its settings, thereby ensuring a fair comparative analysis. The results are presented in Table 1, with detailed neural network architectural specifications provided in the Supplementary Table 3.

For the carbon chains, the introduction of additional H$^+$ causes some C atoms to exhibit opposite charge states[43], as shown in Supplementary Fig. 1a. Although we did not fit partial charges, our model qualitatively captures the distribution of DFT partial charges, as shown also in Supplementary Fig. 1a. In the linear $C_{10}H_2$ molecule, our model demonstrates symmetric charge distribution around the center which shows agreement with DFT. After explicitly incorporating physical long-range interactions, our framework achieved additional improvements in error metrics compared to NequIP and 4G-HDNNP. For positively charged NaCl clusters, the predicted energy and forces by our model are still more accurate compared to 4G-HDNNP, as well as the potential energy surface shown in Supplementary Fig. 1b. For $Au_2$ adsorption on MgO surfaces and benzotriazole interactions with Cu-(111) surfaces in dry and aqueous environments, our model

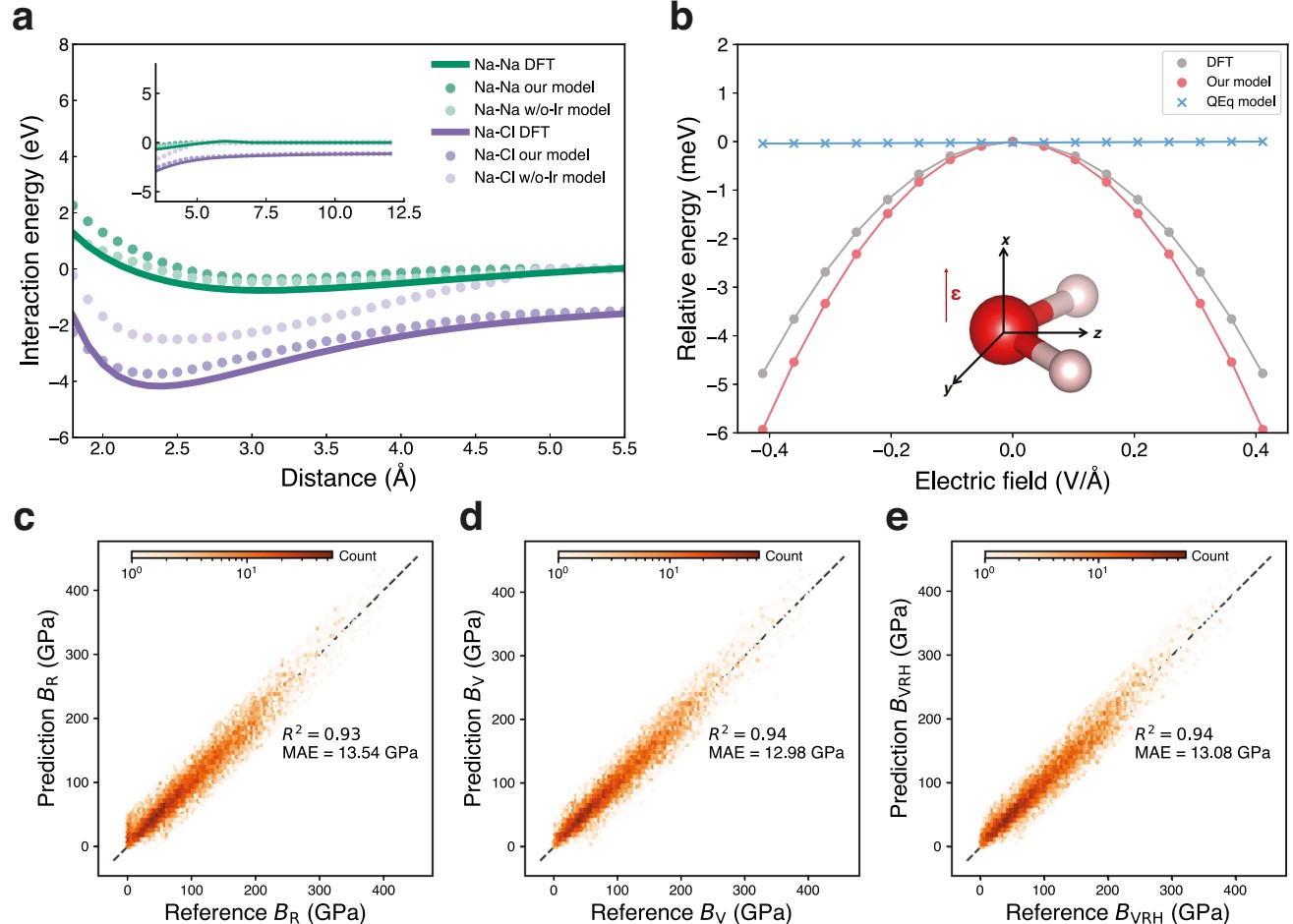

**Fig. 2 | Benchmarking of the foundation model. a** The interaction energies for Na-Na (green) and Na-Cl (purple) dimers predicted by our model, the model without polarizable long-range interactions (w/o-lr model), and reference calculations using density functional theory (DFT)[39]. **b** Evaluation of polarizable interactions in water molecules. The water molecule model is oriented in the *yz*-plane, with oxygen and hydrogen atoms shown in red and white, respectively. The energy response curves as a function of external electric field (**ε**) strength applied along the *x*-axis. Results are compared among our model (red), the Charge Equilibration (QEq)-based model[25] (blue), and the reference DFT calculations (gray) serving as the reference values. **c**–**e** Comparison of bulk modulus from DFT calculations (reference values) and our model (prediction values): (**c**) Voigt approach ($B_V$), (**d**) Reuss method ($B_R$), and (**e**) Hill average ($B_{VRH}$). The R-squared values ($R^2$) and mean absolute errors (MAE) of the bulk modulus are also shown.

outperforms NequIP significantly. Additionally, our model correctly predicted the preferential adsorption geometries for both doped and undoped MgO, as listed in Supplementary Table 1. As for the charged Ag clusters, due to their small system size and lacking long-range effects, the entire structure falls within the typical cutoff radius used by the MLIP. The energy dependence on the overall charge state of clusters leads to degeneracy between atomic structures and potential energy surfaces, resulting in poor performance of the charge-independent NequIP model. We acknowledge that for such small, highly charged systems, the 4G-HDNNP approach to learning charge equilibration parameters is more convenient. Nonetheless, we also achieved reduced force error with the fixed physical parameters compared to 4G-HDNNP. Although PQEq partial charges are not directly comparable to DFT Hirshfeld ones, we compare them to show their relevance. Varying degrees of agreement across different chemical systems are observed, as shown in Supplementary Table 2.

Our comprehensive evaluation demonstrates that explicit incorporation of physical long-range interactions significantly enhances the performance of MLIPs across diverse charge-state systems. Notably, our proposed framework outperforms 4G-HDNNP in most cases for energies and forces without fitting partial charges. This advantage likely stems from a more physically meaningful partition of the total energy.

## Foundation model benchmark

We trained a foundation model for all the periodic table elements up to Pu using the MPtrj dataset[16] following our framework (our model), as described in the Methods section. We also trained a model maintaining the same MLIPs framework and dataset, but without the polarizable long-range interactions (w/o-lr model). The former one demonstrated mean absolute errors (MAE) of 18 meV per atom for energies, 0.065 eV·Å$^{-1}$ for forces, and 0.301 GPa for stresses across the test set, outperforming the w/o-lr model, as shown in Supplementary Table 4. These proved the benefit of incorporating long-range interactions in the enhancement of prediction accuracy.

Message-passing mechanisms will fail when graph connections break in the system. For instance, in dimers, when the interatomic distance exceeds the graph cut-off, the atoms no longer interact with each other. We analysed interaction energies for Na-Na and Na-Cl dimers across interatomic distances ranging from 1.8 Å to 12.0 Å. As shown in Fig. 2a, both models (our model and the w/o-lr model) trained on the MPtrj dataset accurately reproduce the equilibrium bond lengths of the two dimers. However, the w/o-lr model failed to differentiate interaction energies beyond the MLIPs cutoff distance, i.e., 5.0 Å, while our model successfully captured the DFT potential energy surface throughout the entire distance range, demonstrating its capability in describing extended ionic interactions. Moreover, the message-passing mechanism

exhibits deficiencies when simulating layered materials. Taking lithium iron phosphate (LiFePO$_4$) as a case study, we observed that the w/o-lr model predicted an irreversible phase transition, as shown in Supplementary Fig. 3. In contrast, our model correctly reproduced the experimentally[44] and AIMD[45] thermally stable olivine structure. These results collectively demonstrate that the explicit long-range interactions are crucial for accurately describing both ionic systems and layered materials, particularly in predicting structural stability and thermal behavior.

Polarization plays a fundamental role in determining molecular responses to external electric fields. Conventional QEq-based models have limitations in their response to an external electric field applied orthogonally to the molecular dipole. We validated this capability through a benchmark study examining a water molecule's response to external electrostatic fields[46]. The water molecule was fixed in the *yz*-plane, and we applied varying external electric fields along the *x*-axis to derive energy curves, with the results presented in Fig. 2b. The model with polarizable long-range interactions demonstrates a trend consistent with the reference values obtained from DFT. Due to the external electrostatic field direction consistently being orthogonal to the molecule, the electrostatic energy in the QEq-based model (like 4G-HDNNP) is 0 and is completely independent of field strength (i.e., $\mu \cdot \epsilon = 0$, where $\mu$, $\epsilon$ are dipole moment and electric field, respectively). This is because the QEq model treats atoms in the molecule as point charges or Gaussian charges, without distinguishing between core and shell components. We also compared the response of O atom partial charges across DFT Hirshfeld calculations, our model, and the conventional QEq model, as shown in Supplementary Fig. 4. The agreement between the PQEq model and DFT is observed, while the QEq model completely lacks response. This comparative analysis establishes that the explicit polarizable interactions are crucial for modeling molecular systems subject to external electric fields.

Besides the failure of response in static conditions, the QEq method may exhibit non-physical behavior in dynamics[47]. As illustrated in Supplementary Fig. 5, we conducted additional investigations of two water molecules under an external electric field of 0.25 V·Å$^{-1}$. In charge distribution models, it is essential to allow systems to form an internal electric field opposing the external field. The QEq model assumes the system behaves as a perfect conductor without penalizing charge transfer as a function of distance. This limitation of the QEq method leads to non-physical charge transfer between molecules separated by large distances[47]. In contrast, within the PQEq model, the individual water molecules can polarize, generating internal electric fields that counteract the external field, thereby capturing more realistic electrostatic responses in molecular systems. Under the QEq scheme, the water molecules accumulate non-zero net charges and migrate to the top and bottom of the simulation box. In contrast, our model maintains charge neutrality of the water molecules, which remain stationary despite the applied electric field, aligning with DFT results.

We conducted benchmarking using properties that were not labeled during the training process. We applied our foundation model to predict the mechanical properties of 10,154 materials from the Materials Project[48]. Figure 2c–e illustrates the comparison of the bulk modulus $B$ determined by the foundation model and DFT. Our model demonstrated impressive performance, achieving an $R^2$ of 0.94 with the Voigt approach $B_V$ and Hill average method $B_{VRH}$, and 0.93 using the Reuss method $B_R$. In comparison, as shown in Supplementary Fig. 2, the w/o-lr model achieved lower $R^2$ values of 0.89, 0.86, and 0.88 for $B_V$, $B_R$, and $B_{VRH}$, respectively. These not only highlight the robust performance of the model but also establish a solid foundation for the high-throughput screening of materials with exceptional mechanical properties.

## Transferability of the foundation model
In terms of foundation model transferability, evaluations were conducted across diverse systems. The model accurately reproduces the potential energy surfaces and successfully distinguishes between neutral and ionic states in OH·OH systems, capturing the long-range Coulombic repulsion of ionic state as shown in Supplementary Fig. 8. Further validation using a periodic water system demonstrates the model's capability to predict polarization effects under varying electric fields, achieving comparable performance to specialized models as depicted in Supplementary Fig. 10. Additionally, as shown in Supplementary Fig. 11, when tested on 7211 molecules from the QM-7b dataset[49], the model shows remarkable accuracy in predicting molecular polarizability with a mean absolute error of 4.57 atomic units (a.u.), highlighting its robust transferability in capturing polarization.

## Computational efficiency
Despite the incorporation of additional long-range interaction calculations, our foundation model maintains high computational efficiency. Performance benchmarks were conducted on a single NVIDIA H100 GPU to quantitatively assess the computational cost. As illustrated in Supplementary Fig. 6, for a periodic system comprising 2160 atoms, our model requires approximately 0.07 s per molecular dynamics time step, representing a significant improvement over conventional universal MLIPs, which require approximately 0.91 s. This computational advantage extends to larger systems, as demonstrated by simulations of a 24000-atom system, where our model maintains efficiency with only 1.49 s per time step. Such computational performance enables nanosecond-scale molecular dynamics simulations of systems containing tens of thousands of atoms, making it practical for more realistic modeling of materials.

## Materials modeling applications
To demonstrate the capability of our model to reproduce results from AIMD simulations, we simulate the kinetic transport properties of a solid-state electrolyte. Specifically, we investigated lithium-ion diffusivity within the cubic phase of Li$_7$La$_3$Zr$_2$O$_{12}$ (c-LLZO)[50], a crystalline superionic conductor known for its remarkable stability as a lithium conductor. Owing to the efficiency of our foundation model, we are able to conduct MD simulations on the c-LLZO comprising 64 formula units, with a duration of 2 ns for each temperature range from 800 K to 1800K. Figure 3a presents the Arrhenius plots derived from our model, w/o-lr model, and AIMD[51] simulations. The diffusion coefficients were determined from the slope of the logarithmic mean square displacements (MSD) versus logarithmic time within the Fickian regime[52]. MSD curves of our model are depicted in Fig. 3b. Compared to previous work using AIMD simulations, our model clearly reproduces the diffusion coefficients and activation energy. The performance surpasses that of the w/o-lr model, which lacks explicit long-range interactions. With the ability to reach larger model sizes and longer simulation times, our model achieves these results with significantly reduced uncertainties. This capability not only delivers comparable results to AIMD simulations but also enables larger-scale simulations that cannot be assessed by AIMD, enhancing the statistical significance of MD studies. Consequently, the framework developed in this work provides more reliable and comprehensive data for diffusion analysis in solid-state electrolytes.

To show the application of polarizable potentials, we performed MD simulations on a typical perovskite ferroelectric material, BaTiO$_3$. As temperature increases, BaTiO$_3$ undergoes a sequential phase transition process, transforming from rhombohedral to orthorhombic, then to tetragonal, and finally to a cubic structure[53]. Accurately simulating phase changes in ferroelectric substances requires precise potential energy functions that can respond to the small atomic shifts and structural changes, as well as account for the free energy landscape under the conditions of finite-temperature thermodynamics[54]. The phase sequence of BaTiO$_3$ has been extensively studied using effective Hamiltonians[55–58], CMD[59,60], reactive force fields (ReaxFF)[61], specialized MLIPs models trained on the BaTiO$_3$ system[54,62], and

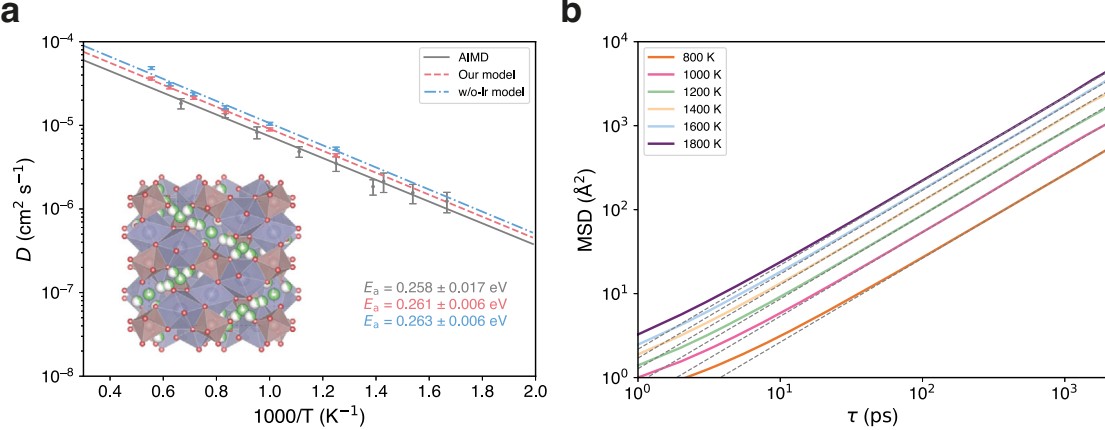

**Fig. 3 | Ionic diffusivity of cubic phase Li$_7$La$_3$Zr$_2$O$_{12}$. a** Crystal structure of cubic phase Ia3d Li$_7$La$_3$Zr$_2$O$_{12}$ (c-LLZO) and Arrhenius plots depicting the lithium-ion diffusion coefficients across varying temperatures (T). The dark blue polyhedron signifies La located at the 24(c) site, and the light brown polyhedron indicates Zr at the 16(a) site. Li fraction occupies the 24(d) and 96(h) sites. Predicted diffusion coefficients (D) of ab initio molecular dynamics (AIMD)[51], our model, and the model without polarizable long-range interactions (w/o-lr model) are presented to calculate activation energies ($E_a$). The error bars represent the standard deviation of the diffusivity, calculated based on the total number of effective ion hops observed in the MD simulation, following the methodology proposed by the reference[51]. **b** 2-ns mean square displacements (MSD) verse time ($\tau$) using our model of lithium-ion in c-LLZO with different temperatures ranging from 800 K to 1800 K in an increment of 200 K. The linear dashed gray lines, with a slope of 1, are also plotted.

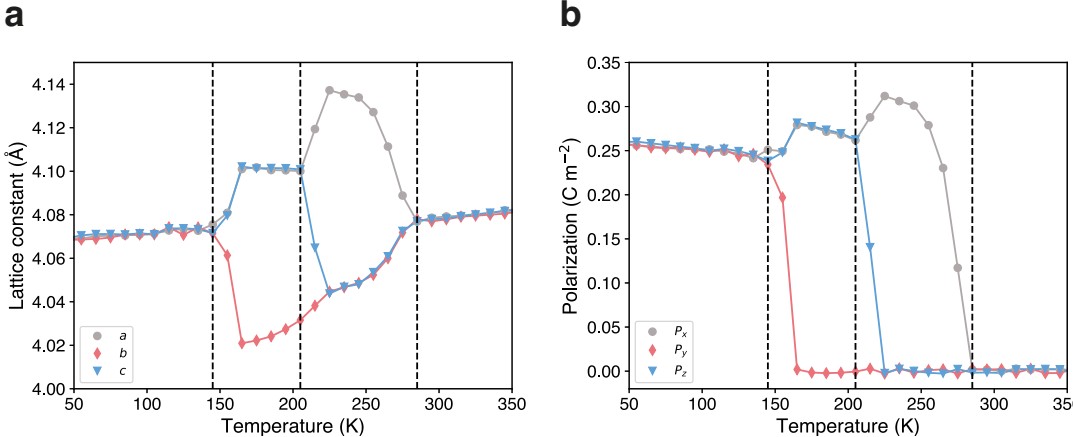

**Fig. 4 | Phase transition of BaTiO$_3$.** The temperature dependence of (**a**) lattice constants (a, b, and c in gray, red, and blue, respectively), and (**b**) local polarizations of unit cells in each direction ($P_x$, $P_y$, and $P_z$ in gray, red, and blue, respectively) exhibit notable changes during the phase transitions observed from molecular dynamics simulations on $10 \times 10 \times 10$ supercell of BaTiO$_3$.

experiments[63]. To detect lattice distortions that distinguish different phases, a $10 \times 10 \times 10$ supercell of BaTiO$_3$ was simulated. The use of larger simulation cells ($10 \times 10 \times 10$ in our work versus $4 \times 4 \times 4$ in Gigli et al.[54]) may provide some advantages for phase transition studies. Larger supercells can effectively reduce the impact of temperature fluctuations, enabling better temperature control for sampling the free energy landscape, thereby yielding results with greater statistical significance. Figure 4 illustrates the simulation results, which clearly identify four distinct phases and the three first-order phase transitions within the simulated temperature range. Below 145 K, the overall average polarization of the supercell aligns with the (111) direction, indicative of the rhombohedral phase. At 145 K, a phase transition to orthorhombic is suggested as the y-component of polarization approaches zero. With further increase to 205 K, the polarization predominantly orients along the x-direction, indicating the tetragonal phase. This phase persists until 285 K, where the cubic paraelectric phase is observed. The obtained phase transition temperatures align closely with those predicted by various models and experiments, as

listed in Table 2. It is worth noting that the supercell size can affect phase transition temperatures, so that direct comparison between different simulations is not expected. Utilizing first principles methods or fitting potential energy surfaces to first principles data often underestimates these temperatures, which is attributed to the approximated exchange-correlation functional[59] employed in DFT. Although the predicted phase transition temperatures are lower than the experimentally observed values, the model effectively captures the sequence of phase transitions as measured in experiments, demonstrating the efficacy of the foundation model in modeling highly polarized phase transitions in ferroelectric materials.

The final application studied in this work involves all-solid-state batteries, which represent a breakthrough in the evolution of next-generation energy solutions, owing to their superior energy density and inherent safety features[64]. Lithium thiophosphate, known for its exceptional ionic conductivity ($\sim 10^{-3}$ S cm$^{-1}$), is deemed the most promising candidate for solid electrolytes and has been widely studied through experiments[65–67] and computer modeling[68–70]. Notably, the

**Table 2 | The phase transition temperatures of BaTiO₃ obtained by effective Hamiltonian[56,57], second principles[58], reactive force field (ReaxFF)[61], specialized machine learning potential (MLIP)[54] and experiments[63]. The size of the supercell is also displayed. $T_{c, R-O}$, $T_{c, O-T}$, and $T_{c, T-C}$ are the phase transition temperatures of rhombohedral–orthorhombic, orthorhombic–tetragonal, and tetragonal–cubic, respectively**

| Method | $T_{c, R-O}$ (K) | $T_{c, O-T}$ (K) | $T_{c, T-C}$ (K) |
|---|---|---|---|
| This work (10×10×10) | 145 | 205 | 285 |
| Effective Hamiltonian (16×16×16)[56] | 119 | 158 | 257 |
| Effective Hamiltonian* (12×12×12)[57] | 150 ± 10 | 195 ± 5 | 265 ± 5 |
| Second Principles (16×16×16)[58] | 140 | 180 | 224 |
| ReaxFF (6×6×6)[61] | N.A. | N.A. | 240 |
| MLIP (4×4×4)[54] | 18.6 | 91.4 | 182.4 |
| Experiments[63] | 183 | 278 | 403 |

*Deviations from quantum Monte Carlo simulations.

nanoporous β-Li₃PS₄ has been validated to exhibit outstanding cycling stability[71], presumably attributed to the formation of Li₂S and Li₃P solid-electrolyte interphases (SEI) during the initial battery cycles, which serves to passivate further degradation of the electrolyte[69,72–74].

To investigate the formation of SEI, we utilized the foundation model to simulate the interfacial reactive MD of the solid electrolyte β-Li₃PS₄ in contact with the lithium metal anode, encompassing a system of 13,760 atoms. The initial β-Li₃PS₄/Li interfacial structure, measuring 33.15 nm × 3.1 nm × 2.6 nm, was utilized in the MD simulations, as depicted in Fig. 5a. The highly reactive Li metal began to interact with the PS₄ tetrahedron at the interface, triggering the formation of the SEI layer as illustrated in Fig. 5b. With the growth of the SEI layer, electrons were transferred from the Li anode to the Li₃PS₄ electrolyte, which causes a gradual increase of the partial charges of Li as they migrate from the anode, across the SEI, and into the electrolyte. This results in a transition from metal lithium to ions. The dynamic behavior of partial charges at the anode-electrolyte interface is a critical aspect showcased in Fig. 5c. Initially, the partial charges of lithium changed almost linearly from the anode to the electrolyte. Over time, this distribution evolved and showed a distinct plateau. This phenomenon could be attributed to the ordered structuring of the SEI, signifying the nucleation of crystalline structures. Ultimately, a stable lithium partial charge plateau was formed between the anode and the electrolyte, spanning a range of 15 to 21.5 nm. This is further supported by the visualized structure depicted in Fig. 5b, where the plateau within the SEI aligns with the Li₂S crystal and amorphous regions. Chen et al.[68] employed AIMD simulations to investigate the radial distribution functions (RDF) of Li-P and Li-S bonds, which concluded that the decompositions of electrolytes in lithium thiophosphates are primarily due to the decomposition of PS₄ tetrahedron by the active lithium metal, leading to the formation of new Li-P and Li-S bonds. Our RDF analysis confirmed the formation of Li-S and Li-P bonds within the SEI and electrolyte as depicted in Fig. 5d. In agreement with the AIMD simulations[68], we observed the emergence of a new Li-P peak within the SEI at approximately 2.5 Å. Furthermore, within the SEI, the Li-S bond displayed RDF characteristics consistent with crystalline Li₂S[75]. For the Li-P bonds, the peak shape closely resembles that of amorphous Li₃P AIMD simulations[76].

As illustrated in Fig. 6a, b, the ultimate decomposition products of P form only short-range ordered structures, in contrast to S, which forms long-range ordered structures. This observation agrees with the measurements where Li₂S crystals were detected but Li₃P crystals were absent in cryogenic transmission electron microscopy (cryo-TEM)

experiments, as noted in the reference[66]. To enhance the understanding of the formation mechanism, the atomic compositions of the SEI in both crystalline and amorphous Li₂S regions were examined. As depicted in Fig. 6c, d, within the first 2 ns, the emergence of amorphous Li₂S and Li₃P regions was attributed to the swift diffusion of P and S atoms. Conversely, the formation of crystalline Li₂S regions was predominantly influenced by the swift diffusion of Li atoms and the slower diffusion of P and S atoms. As the system further evolves to 4 ns, the diffusions slowdown, which also confirmed that the formation of crystalline Li₂S region hindered further atomic diffusion and thus slowed the growth of the SEI. As illustrated in Fig. 5b, during the initial phase of decomposition (0–100 ps), the swift diffusion of P, S, and Li led to the formation of an interface of about 3 nm thick. However, in a later stage between 100 and 2000 ps, the formation of Li₂S nuclei impeded the diffusion of P and S atoms, resulting in a period of sluggish SEI growth. Consequently, the interface experienced only a slight expansion in this period, increasing by only 1 nm. Furthermore, in the following 2 ns, there was essentially no SEI growth. The final interface structure ( ~ 8.5 nm) could be characterized by a 4.5 nm crystalline Li₂S region sandwiched between two 2-nm transitional layers. The cessation of growth at this interface suggests that the crystalline Li₂S and amorphous Li₃P within the crystal region contribute to the stabilization of both the lithium metal anode and the electrolyte, which agrees with the theoretical predictions made by DFT calculations[72].

Additionally, we studied the interface reactions between Li₆PS₅Cl and lithium metal, as detailed in Supplementary Note 13. As shown in Supplementary Fig. 13, at a pressure of 1 atm and a temperature of 300 K, Li₆PS₅Cl formed an interface of about 5-nm thick after 2 ns. This indicates that the Li₆PS₅Cl electrolyte exhibits a higher stability against lithium metal at room temperature compared to Li₃PS₄. Upon elevating the temperature to 500 K, we detected Li₂S at the grain boundaries, which aligns with the cryo-TEM observations[66].

### Finetuning for enhanced accuracy

While our foundation model demonstrates general performance, achieving ab initio accuracy for specific systems requires finetuning. Using Na₈/₉Cl₈⁺ clusters and Au₂ dimers on MgO surfaces from Ko et al.[28] as test cases, we performed targeted finetuning using 20% subsets of configurations and mean squared force error as the sole loss function. After finetuning, our model achieved good agreement with DFT references, accurately distinguishing potential energy surfaces for positive charged clusters and surface conditions, as shown in Supplementary Fig. 12 and Supplementary Table 6. This success demonstrates the effectiveness of the two-step approach: foundation pretraining followed by efficient finetuning, enabling quantum-accurate predictions while maintaining the model's fundamental physical insights and transferability.

### Discussion

In this study, we introduced a framework that integrates equivariant machine learning interatomic potentials with explicit long-range polarizable interactions. QEq-based models inherently struggle with atomic polarization because they do not distinguish between core and shell components. Due to their inability to handle atomic-level responses to external fields, they may exhibit non-physical behaviors in some cases, as described in the reference[47]. Our framework incorporates polarizable long-range interactions and accurately captures atomic polarization in the presence of external electric fields. Since our framework decomposes the total energy into electrostatic energy and remaining energies that only depend on atomic types and positions, any future machine learning techniques and/or high-order electrostatic energy expansion schemes can be readily integrated into it, empowering possible accuracy enhancement.

Our foundation model successfully reproduced DFT potential energy surfaces for Na-Cl/Na-Na dimers across extended interatomic

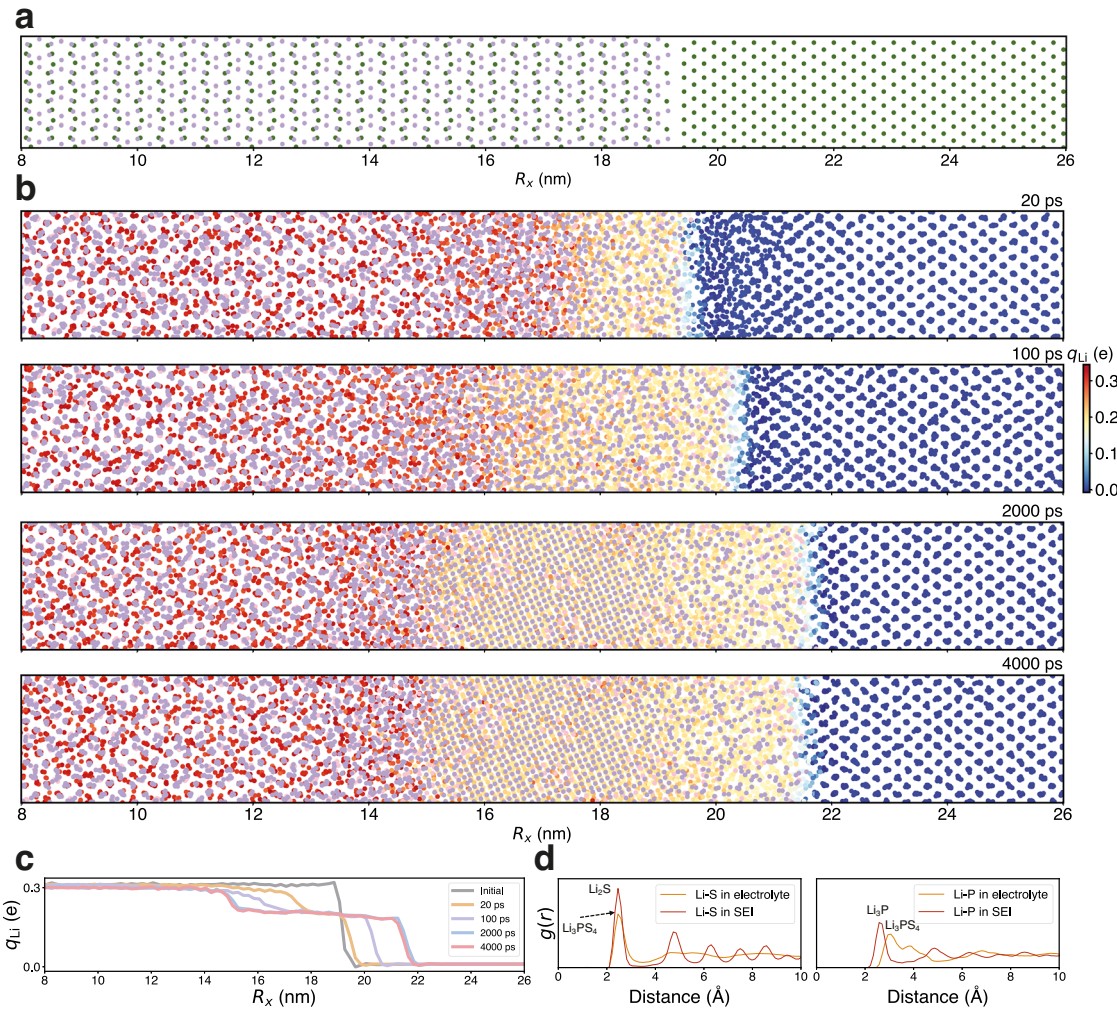

**Fig. 5 | Visualization of solid electrolyte interphase between Li anode and Li₃PS₄ electrolyte. a** The initial structure of $\beta$-Li₃PS₄ (010)/Li (100) interface. The elements are color-coded: Li in green, S in purple, and P in pink. A segment of the primary reaction zone, ranging from 8 to 26 nanometers along the $x$ direction ($R_x$), is displayed. **b** Snapshots at different times during the MD simulations. The partial charges on lithium ions ($q_{Li}$) are represented with color coding to enhance the visibility of structural transformations during the formation of the solid electrolyte interphase (SEI). The SEI layer, approximately 8.5 nm in thickness, forms after 4 ns molecular dynamics (MD) simulations, comprising an amorphous $\beta$-Li₃PS₄/Li₂S interface ( ~ 2 nm), a crystalline Li₂S layer ( ~ 4.5 nm), and an amorphous Li₂S (Li₃P)/Li interface layer ( ~ 2 nm) in sequence. **c** The distributions of the Li partial atomic charges along the $x$-direction at initial, 20, 100, 2000, and 4000 ps state. **d** The radial distribution function ($g(r)$) plots of Li-S and Li-P within the electrolyte and SEI layer.

distances, outperforming conventional message-passing MLIP beyond the cut-off regions. The model demonstrates exceptional versatility across multiple materials science challenges, including accurate prediction of thermal stability in complex layered materials, high-fidelity capture of molecular responses to external electric fields, and reliable prediction of bulk modulus. The model has been also successfully applied to study dynamic properties and reactive molecular dynamics, such as lithium-ion diffusion in solid-state electrolytes and temperature-dependent phase transitions in ferroelectric materials like BaTiO₃, while also elucidating interface formation mechanisms between lithium thiophosphate electrolytes and lithium metal anodes in solid-state batteries.

Beyond these specific applications, we evaluated the model's transferability, a critical attribute for a foundation model. The model accurately reproduces the potential energy surfaces for OH-OH systems, correctly distinguishing between the neutral state and the long-range Coulombic repulsion of the ionic state. The model achieved a remarkable mean absolute error of just 4.57 a.u. for molecular polarizability, highlighting its robust and generalizable learned representation of polarization physics. Finally, we demonstrated our

foundation model's capacity to achieve ab initio accuracy for specific systems through efficient finetuning.

We acknowledge that our model aims to achieve a balance between computational efficiency and accuracy. Several advanced machine learning approaches for long-range interactions may achieve higher accuracy for specific, complex systems, including higher-order charge expansion methods and simultaneous optimization of QEq parameters[32,77] in each step. However, more accurate methods incur higher computational complexity. The method that dynamically solves charge equilibration parameters at each step requires additional cost for MD simulations. It is also observed that the charge equilibration parameters do not change significantly during MD simulations for that method[77], suggesting that expanding polarization charges to higher orders while using fixed, a priori parameters may achieve higher accuracy with tolerable computational overheads. Additionally, several approaches that do not require a charge equilibration process provide valuable insights. Latent Ewald Summation[24] captures long-range interactions by learning hidden variables from local descriptors and applying Ewald summation to these variables. LSR-MP[78] employs a fragmentation-based approach with hierarchical message passing between atoms and

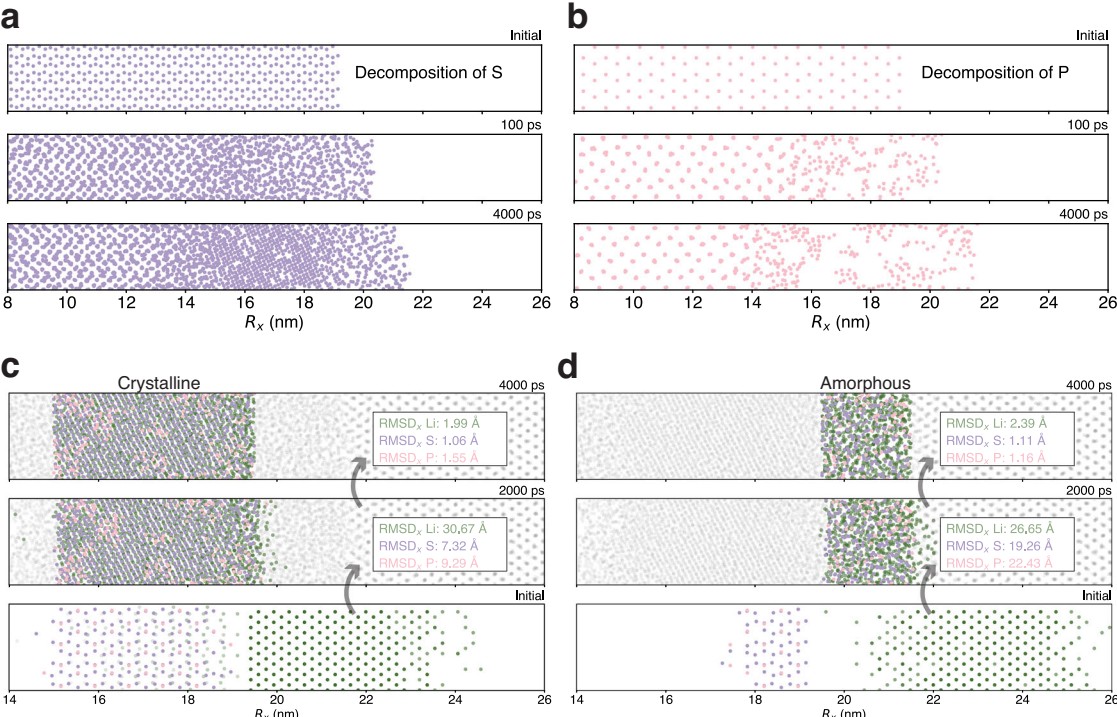

**Fig. 6 | The formation mechanism of Li/Li$_3$PS$_4$ solid electrolyte interphase layer.** The snapshots capture the dynamic decomposition behavior of PS$_4$, showcasing: (**a**) the sulfur, and (**b**) the phosphorus component along the $x$ direction ($R_x$). To elucidate formation mechanism of the solid electrolyte interphase (SEI) layer, atomic component analyses of the (**c**) crystalline, and (**d**) amorphous Li$_2$S regions are compared among 4-ns (upper), 2-ns (middle), and the initial state (lower). The elements are color-coded: Li in green, S in purple, and P in pink. The $x$-component root mean squared displacements (RMSD$_x$) of Li, P, and S between the initial and the 2-ns states and between 4-ns and 2-ns states are also shown.

fragments to learn the long-range interactions. NeuralP³M[79] introduces mesh points alongside atoms to discretize space and capture long-range effects. SO3krates[80] utilizes spherical harmonic coordinates and equivariant attention for interactions at arbitrary length scales. LOREM[81] models charges as equivariant tensors to capture orientation-dependent interactions beyond the standard cutoff radius. In contrast to these approaches, our foundation model explicitly simulates charge redistribution through a physics-driven framework that maintains a transparent connection to the underlying physical principles of charge polarization and electrostatic interactions. This design choice enhances interpretability while preserving computational efficiency. A promising direction for future work lies in exploring strategic integrations of our framework with these complementary methodologies to develop even more comprehensive and powerful foundation models.

## Methods
### Model training details
To train the foundation equivariance neural network potential, we used the MPtrj dataset[16] sourced from Materials Projects[48] as the training dataset. All configurations were calculated using DFT with the PBE[82]/PBE + U[83] exchange-correlation functional and pseudopotential basis. The PBE/PBE + U mixing compatibility correction[84] was applied to ensure energy consistency. Due to transferability issues with the Yb element[85], all data containing Yb in the MPtrj dataset were excluded. The higher-order energy terms and the electrostatic Coulomb interaction energies due to charge fluctuations should be deducted from the total potential energy. Appropriately setting the PQEq parameters is crucial for training a foundation framework. Detailed information on the fitting process and the PQEq parameters used in this work can be found in the Supplementary Note 10 and Supplementary Table 7.

For equivariant neural network training, the NequIP[12] model was utilized, incorporating six equivariant layers with node features set at

$256 \times 0e + 256 \times 1e$. Spherical harmonic basis functions were used to represent interatomic directions, denoted as $1 \times 0e + 1 \times 1o$. Interatomic distances were described using eight Bessel basis functions, and a cut-off value of 5.0 Å was chosen for building the MLIP neighbor list. The dataset is divided into a training set, validation set, and test set in a ratio of 8:1:1.

The total loss function ($\mathcal{L}$) of prediction values $\mathbf{x}^{\mathrm{pred}}$ and reference first principle values $\mathbf{x}^{\mathrm{ref}}$ can be expressed as,

$$L\left(\mathbf{x}^{\mathrm{ref}}, \mathbf{x}^{\mathrm{pred}}\right) = \sum_i^N \frac{1}{N}\left(x_i^{\mathrm{ref}} - x_i^{\mathrm{pred}}\right)^2 \tag{2}$$

$$\mathcal{L} = \omega_E \cdot L\left(\mathbf{E}^{\mathrm{ref}}, \mathbf{E}^{\mathrm{pred}}\right) + \omega_F \cdot L\left(\mathbf{F}^{\mathrm{ref}}, \mathbf{F}^{\mathrm{pred}}\right) + \omega_S \cdot L\left(\mathbf{S}^{\mathrm{ref}}, \mathbf{S}^{\mathrm{pred}}\right) \tag{3}$$

where energies $\mathbf{E}$, forces $\mathbf{F}$, and stresses $\mathbf{S}$ weights were set as $\omega_E = 1$, $\omega_F = 1$, and $\omega_S = 10$ with quantity units of eV, eV·Å$^{-1}$, and eV·Å$^{-3}$. The batch size was selected as 128, and the Adam optimizer was employed with an initial learning rate of $10^{-3}$. All the trainable parameters were initialized with the random seed 3407. The implementation of the entire neural networks and long-range interactions was based on JAX 0.4.20[86]. The training was conducted on a single NVIDIA H100-PCIe GPU, utilizing CUDA version 12.3 and Driver version 545.23.06.

### Molecular dynamics simulations
The Atomic Simulation Environment (ASE)[87] was utilized as the interface for geometry relaxations and MD simulations. Structure relaxations were conducted using the limited-memory BFGS[88] method. Both atomic coordinates and cell vectors were optimized concurrently until the forces fell below the convergence threshold of 0.05 eV/Å.

The c-LLZO with a space group of Ia$\bar{3}$d originated from experimental structures detailed in the reference[50]. The structure was expanded to a $2 \times 2 \times 2$ supercell. Subsequently, 180 lithium atoms were

randomly placed on the 24(d) sites, and an additional 268 lithium atoms on the 96(h) sites. After relaxation, the structure served as the starting point for MD simulations. The isothermal isobaric (NpT) ensemble simulations were performed with the temperature control via Nosé-Hoover thermostat[89,90] and pressure maintained at 1 atm using Parrinello-Rahman barostat[91,92] to determine the lattice constants at various temperatures. Then, the canonical (NVT) ensemble simulations with the Nosé-Hoover thermostat were utilized to ascertain the lithium diffusion properties within c-LLZO. A time step of 2 fs was used for all NpT and NVT simulations. In the case of NVT simulations, following a 100 ps equilibration period, 2-ns trajectories' mean squared displacements of Li ions were employed to calculate the self-diffusion coefficients at various target temperatures. Additionally, we conducted an uncertainty analysis of the diffusion coefficients, applying the empirical error estimation method as detailed in the reference[51].

To construct the phase diagram of the $BaTiO_3$ crystal structure, a supercell consisting of a $10 \times 10 \times 10$ R3m $BaTiO_3$ lattice with 5000 atoms was created. The crystal structure data for $BaTiO_3$ was sourced from the Materials Project[48]. NpT simulations with the Nosé-Hoover thermostat and Parrinello-Rahman barostat were performed with a 1-fs timestep. Throughout the simulations, the pressure was consistently maintained at 1 atmosphere. To regulate temperature, the procedure began at 5 K and involved incrementing the temperature by 10 K every 20 ps. The average lattice constants and the local polarization of the unit cell, denoted as **P**, were determined using data from the final 10 ps of trajectories at each temperature step. For the comprehensive calculation method of the local polarization, please refer to the Supplementary Note 12.

We utilized a 13760-atom structure consisting of a ca. 13.6-nm thick Im3m lithium metal anode with (100) facet, paired with a ca. 19.1 nm thick Pnma $\beta$-$Li_3PS_4$ electrolyte with (010) facet to illustrate the SEI formation across realistic spatial and temporal scales. The system was structured with periodic lateral dimensions measuring 3.1 nm by 2.6 nm. The unit cell of anode and electrolyte structures originated from the Materials Project[48]. The entire cell was subject to periodic boundary conditions, with a 1 nm atomic layer fixed on each side along the x-direction to ensure a singular interface reaction. NpT MD simulations were conducted at a standard temperature of 300 K and a pressure of 1 atm by Nosé-Hoover thermostat and Parrinello-Rahman barostat with a timestep of 2 fs. Initial velocities for the particles were assigned randomly following the Maxwell-Boltzmann distribution, and the simulations were carried out for a duration of 4 ns.

## Data availability

For datasets containing different charge states, the $C_{10}H_2$/$C_{10}H_3^+$, $Na_{8/9}Cl_8^+$, $Ag_3^{+/-}$ and $Au_2$-MgO datasets are publicly available from the reference[93] at https://doi.org/10.24435/materialscloud:f3-yh, while BTA-Cu and BTA($H_2O$)-Cu datasets are available at https://github.com/ahmad-research-group/nequip-charge. The MPtrj dataset is also publicly available from the reference[94] through https://doi.org/10.6084/m9.figshare.23713842. Detailed polarizable charge equilibration parameters are provided in the Supplementary Table 7. The trajectory data of molecular dynamics are available at https://github.com/reaxnet/reaxnet. Source data are provided with this paper.

## Code availability

A JAX implementation code is available under GNU General Public License v3.0 via GitHub at https://github.com/reaxnet/reaxnet or Zenodo (https://doi.org/10.5281/zenodo.17123107[95]).

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

## Acknowledgments

This work was supported by Hong Kong Quantum AI Lab Limited, Air @ InnoHK of Hong Kong Government (G. C. and Z. H.), Seed Fund for Basic Research for New Staff 2022/23 (Grant No. 2201101550 (Z. H.)), the GuangDong Basic and Applied Basic Research Foundation (Grant No. 2024A1515013283 (Z. H.)), the Guangdong Shenzhen Joint Key Fund (Grant No. 2019B1515120045 (C. Y.)), and the National Natural Science Foundation of China (Grant No. 22073007 (C. Y.)).

## Author contributions

R. G.: Conceptualization, investigation, code development, analysis, visualization, and writing. C. Y.: Revision, analysis, and funding acquisition. J. M.: Analysis, review, and discussion. S. C.: Analysis, review, and discussion. G. C.: Review, funding acquisition, and supervision. Z. H.: Conceptualization, analysis, revision, funding acquisition, and supervision.

## Competing interests
The authors declare no competing interests.
