## [Transparent Peer Review file · Nature Communications]

A foundation machine learning potential with polarizable long-range interactions for materials modelling

Corresponding Author: Professor GuanHua Chen

Version 1:

Reviewer comments:

Reviewer #1

(Remarks to the Author)
see attached pdf

(Remarks on code availability)
I did not run the code but it looks OK.

Reviewer #2

(Remarks to the Author)
Overall, a manuscript which should be very heavily revised - but that might decrease any novelty reported such that it would fall outside of the scope of the journal. Maybe best to reject.

- What are the noteworthy results?

There are some interesting simulations using the MLIP technology developed by the authors - but there is no reliable benchmarking carried out to compare to other state-of-the-art methods. Since the focus of the manuscript is a supposedly superior method, this is a claim that should have been proven by the authors.

- Will the work be of significance to the field and related fields? How does it compare to the established literature? If the work is not original, please provide relevant references.

This work is mostly combining other, existing pieces of technology in a slightly different way. In short, a short-range MLIP is combined with Goddard's QEq scheme and D3 dispersion corrections. The closest similar is Behler's 4G-HDNNPs, but these don't use the equivariant message passing short range technology. Results presented here are marginally better, but perhaps only an incremental improvement. Presumably due to the better and more expressive short-range part. Behler's scheme is superior as it uses environment dependent parameters for the QEq component, unlike here, where these are fixed. The CENT method by Goedecker (which uses a similar approach to fit the energies directly) is not even mentioned.

- Does the work support the conclusions and claims, or is additional evidence needed?

A lot of data which the results are compared to here are cherry-picked (there is hardly or no any mention of significant universal force fields, such as MACE-MP and its later improvements). This is a quickly moving field.

- Are there any flaws in the data analysis, interpretation and conclusions? - Do these prohibit publication or require revision?
Plenty of misinterpretation/misrepresentation, listed below. With very careful revision maybe these can be rectified, but most likely it will further reduce any claims of superiority and/or novelty.

- Is the methodology sound? Does the work meet the expected standards in your field?

Mostly.

- Is there enough detail provided in the methods for the work to be reproduced?

Yes, and the code is publicly available (I have not tested, though).

Do not antropomorphise mathematical models (e.g. p2/139)

"subtle fluctuations" - these are not subtle (it is meaningless in this context) - they are dynamic fluctuations

Terms in Eq. 1 should be explained immediately after in the text.

Ref. 24 is incomplete

Eqs. 2 and 3 do not follow. Since the energies also depend on the charges, which are determined self-consistently, it is far from obvious that eqs. 2 and 3 are true. There may be an expression analogous to the Hellmann-Feynman theorem, but not shown here.

Panels 2a, 2b and 2c must be united.

The QEq curve on 2d seems completely flat - this cannot be true, or the QEq must be extremely rigid.

Computational cost being linear: I find that hard to believe. Long range interactions in periodic systems are typically treated using some version of the Ewald summation method or an equivalent approach. While there exists linear scaling implementations, there is no mention of these here.

The sentence "as well as account for the free energy surfaces under the conditions of finite-temperature thermodynamics" does not make sense. The free energy is a state function and not a surface.

"The QEq-based machine learning models inherently struggle with atomic polarization, leading to non-physical results under external electric fields." there is no evidence for that. In the results

quoted by the authors on 4G-HDNNPs they may be less accurate but not unphysical.

"outperforming conventional equivariant GNN-based MLIPs" no evidence is showed. (Ref. 48. is not an MLIP that fits this description).

(Remarks on code availability)

I haven't had the chance to try it out, but it is available, and there are sufficient details to download/use.

Reviewer #3

(Remarks to the Author)

Summary

The manuscript presents a semi-heuristic approach to incorporating long-range interactions, such as polarizability, into machine learning potentials. The proposed framework integrates a physically motivated polarizable charge equilibration scheme within an equivariant graph neural network-based short-range potential. While the idea is clever, it is not entirely new—similar approaches have been explored before, such as in *SpookyNet*, although the current work extends these ideas further. The authors claim their model is universal and exhibits linear scaling in system size while accurately capturing long-range interactions.

While the proposed approach is an interesting idea and appears to perform well on specific test datasets, there are significant concerns regarding the validity of the claims made in the title and throughout the manuscript. The manuscript lacks rigorous benchmarking against existing universal models, and the claim of linear scaling contradicts fundamental physical/mathematical limitations. As such, I cannot recommend publication in its current form as outlined below in more details

Major Concerns

1. **Lack of Rigorous Benchmarking for a "Universal" Model**

The authors make a very strong claim that their model is universal, but the provided evidence does not support this assertion. There are two critical issues with this claim:

1. **Absence of Comparison with Established Models:** The manuscript does not compare the model's performance against other well-known universal machine learning potentials (e.g., MACE-MP-0, MATTERSIm, SevenNet, ORB). Not only do the authors omit numerical comparisons, but these models are not even cited. Instead, the error analysis is restricted to their own training dataset, which is insufficient to validate universality.

2. **Inadequate Testing of Universality:** The authors evaluate their model on the common datasets published by Ko et al. But, the author perform their benchmarks bespoke models rather than testing their claimed universal model. If the model is indeed universal, it should be evaluated directly on these datasets without additional training. A fair comparison would involve evaluating both the universal and bespoke models to determine whether the universal model effectively learns long-range physics.

2. **Misleading Scaling Claims**

One of the most problematic claims in the manuscript is that the model achieves linear scaling for long-range interactions:

- **Fundamental Computational Limits:** Long-range interactions require solving the Poisson equation, which fundamentally cannot be computed in linear time. The best possible scaling for such a computation is $O(N \log N)$ using Ewald summation on a grid, while traditional solvers scale as $O(N^{3/2})$ or worse.

- **Questionable Interpretation of Figure S4:** The claim of $O(N)$ scaling is based on Figure S4, but the presented data does not conclusively support this claim. If the authors indeed achieve linear scaling, they must be using a cutoff solver, which is

known to introduce severe errors in energy and force computations.

Given these issues, the scaling behavior of the proposed model needs significant clarification.

3. **Concerns Regarding Specific Applications**

The manuscript presents results on well-known test cases, but the methodology used for comparison raises concerns:

- **Barium Titanate (BTO):** The authors compare phase transition temperatures in Table 2, but this comparison is misleading because the transition temperature is highly dependent on system size. The authors use a $10 \times 10 \times 10$ cell, whereas the reference MLIP from Gigli et al. models use a $4 \times 4 \times 4$ cell. The comparison should be made at equal system sizes or, at the very least, the system size used should be explicitly stated in the table with a comprehensive discussion.
- **Lithium Phosphorous Sulfide (LPS):** The claim that their model captures long-range interactions in LPS is questionable. In this material, interactions are largely screened, and no significant long-range charge transfer is expected to occur. This system has already been simulated successfully with non-long-range potentials (e.g., *Phys. Rev. Materials* 8, 065403*), raising doubts about the necessity and impact of the proposed approach.
- **Lack of Fine-Tuning Capabilities:** Universal models are only widely useful if they can be fine-tuned to specific systems. This is common practice in other domains (e.g., stable diffusion for images or MATTERSIm, MACE-MP0 for materials), yet the authors provide no evidence that their model supports such fine-tuning. Examination of the code suggests that this feature is not even planned.

Minor Issues and Suggested Revisions

1. **Dataset Clarifications:** Clearly state that MPT_{rj} is the target training dataset in the main text (Page 6, Line 169).
2. **Consistency in Units:** Ensure uniform usage of units, likely $\text{V}/\text{\AA}$, in all relevant sections (e.g., Page 7, Line 221).
3. **Charge Values for Water Molecule:** Provide explicit charge values from Figure 2d for easier comparison, similar to what is done in Figure S1a.
4. **Thermostat Choice:** The authors use a Nosé-Hoover thermostat in their molecular dynamics simulations, which does not properly sample the canonical ensemble for small systems and can artificially suppress fluctuations. A stochastic thermostat (e.g., Langevin or Bussi-Donadio-Parrinello) would be more appropriate to ensure correct statistical sampling.

(Remarks on code availability)

The provided repository includes a README file with installation instructions, and the code appears to be installable. However, the core model presented in the manuscript is not included in the repository, preventing direct verification of the results.

Version 2:

Reviewer comments:

Reviewer #1

(Remarks to the Author)

The authors have addressed most of my concerns. One remaining point remains.

Q6. "Figure 2 b-c: the authors should increase the x-axis range to 12 Å to actually see the long-range scaling behaviour."

I do not see "other publications also do it like this" as a valid argument. Understandably, the surrogate model cannot be more accurate than the data-generating DFT model. So, I'm not sure why the authors are trying to hide this.

Thus, the authors should add the results. If not in the main paper then at least in the supplementary information.

(Remarks on code availability)

Reviewer #2

(Remarks to the Author)

Thanks to the authors for their reply and reviewing the manuscript. Factual errors have been mostly corrected, but there remains some open questions. These should be clarified and/or corrected before considering this manuscript any further. Here are some comments:

Table S5 does not represent any significant improvements over other methods. A selling point could be increased transferability etc. but no evidence for that.

Table S6. I have no stake in MACE-MP-0 but I can immediately tell that those results can't be right and I am sure the MACE-MP-0 authors will challenge those as soon as they see this table. It seems extremely unlikely that in all the other metrics MACE-MP-0 does well and suddenly the errors become huge.

About the forces/stresses. Since the forces depend on the charges, which depend on the positions, one must take the chain rule and calculate the appropriate term. Table R2 is not convincing at all - this compares to another implementation presumably using the same formulae. Did the authors compare their forces to forces obtained using finite differences?

Regarding Fig. 2b. and R3 yes, it is clear now that due to the specific direction the electric field was applied this is indeed the case. However, I disagree with the general statement "This is a fundamental limitation of the traditional QEq-based model—it lacks the capability to respond to external electric fields when atomic positions remain fixed." If the electric field had components in the y or z directions, the energy would change.

"The term "free energy landscape" is the standard terminology for describing how free energy varies across configuration space." this is not quite right. This term describes the free energy as a function of a collective variable or set of collective variables. But the manuscript text is OK.

(Remarks on code availability)

Reviewer #3

(Remarks to the Author)

The authors have addressed many of the concerns raised in my previous review, and the manuscript has improved in several respects. For example, the addition of Table 1 is a valuable contribution. However, I still find that the very strong claims made in the title and abstract—particularly the notion of a universal long-range model—are not sufficiently supported by the presented evidence.

A central issue remains: while the model is trained to be universal, the authors do not convincingly demonstrate that it has actually learned the correct long-range physics, as opposed to simply benefitting from increased expressivity due to the added long-range block. This point was raised in my initial review, but the response did not adequately address it.

The authors argue that:

> "we cannot directly validate a universal model trained with VASP/PBE+U functional and pseudopotential basis sets on the datasets proposed by Ko et al."

While technically true, this is not a convincing justification for avoiding such validation. The Ko et al. datasets contain only a few hundred small structures, and recomputing them using the authors' own DFT setup (VASP/PBE+U) is computationally feasible—especially in the context of training a large-scale universal model. Recomputing small datasets for consistent evaluation is standard practice in the field.

The further claim that:

> "so far, no universal model can reach ~1 meV/atom of precision."

misses the point. The Ko et al. datasets include physically motivated test cases, such as displacement scans of NaCl and Au₂ systems, which reveal whether a model correctly captures energy and force trends with interatomic distance. The authors should at least reproduce such physical validation tests with both their universal and bespoke models to support their claims.

Additionally, while Table 1 shows the model's performance on the 4G dataset, the comparison is made only against a basic bespoke model from the original benchmark. Given the field's rapid progress, this is no longer a sufficient baseline. For instance, the recent model by Kim et al. (<https://arxiv.org/abs/2412.15455>) demonstrates significantly better performance for some systems (e.g., an energy error of 0.073 meV/atom and force error of 0.008 eV/Å for the gold dimer on MgO). A fair evaluation should include comparisons to such modern long-range models.

In summary, while the manuscript has improved, the main claims still require more rigorous and convincing validation. I would strongly recommend the authors perform standard physicality tests and include comparisons to other state-of-the-art universal models to support their conclusions.

(Remarks on code availability)

Version 3:

Reviewer comments:

Reviewer #2

(Remarks to the Author)

This revision of the manuscript has brought further improvements and the authors have now clarified most pertinent points raised by the reviewers. The manuscript is publishable.

(Remarks on code availability)

I haven't run the code but the repository exists with examples and instructions.

Reviewer #3

(Remarks to the Author)

The manuscript has improved substantially compared to the previous version. The results now look mostly solid and trustworthy, and many of my earlier concerns have been addressed. I only have a few remaining comments, including one point that still puzzles me.

Major comment:

Physical decay of long-range interactions (Fig. S8). I recommend plotting this figure on a log–log scale in order to assess whether the correct power-law decay of interactions is reproduced. From visual inspection, it seems that the extracted exponent p is too small; for Coulombic interactions, one expects $p = 1$. Could the authors comment on this discrepancy, especially given the claim that their model reproduces correct long-range physics?

Minor comments:

1. Fine-tuning strategy (S11). The additional comparison is very helpful and the results look convincing. Could the authors comment on their fine-tuning strategy in more detail? For example, is this full fine-tuning, multi-head fine-tuning, or a parameter-efficient method such as LoRA? In this context, I am also curious about the performance of the fine-tuned models on the original training tasks—did the authors observe catastrophic forgetting?

2. Comparison table (S10). Thank you for adding this comparison. The results look strong and convincing. It also appears that the Ko dataset is becoming a widely used benchmark. I would like to point out another recent paper (<https://arxiv.org/abs/2507.19382>) which further raises the benchmark results; it may be worth discussing this in the context of related work.

(Remarks on code availability)

Version 4:

Reviewer comments:

Reviewer #3

(Remarks to the Author)

I thank the authors for their revision and improvements. I am happy with the manuscript and support publication in its current form.

(Remarks on code availability)

Response to the reviewers

We sincerely thank the three reviewers for their time and invaluable comments. We have revised our manuscript in accordance with their advice and criticisms and have clarified the issues raised by the reviewers. We have provided point-by-point responses to the questions raised by the reviewers in this document. The modifications to the manuscript are highlighted in **dark red**.

Table of Contents

Response to reviewer #1	2
Response to reviewer #2	13
Response to reviewer #3	30

Response to reviewer #1

Q1. “The authors present a Machine Learning (ML) potential where the short-range graph neural network prediction is combined with the long-range electrostatic and dispersion interactions. While similar approaches were proposed previously, the novelty of this manuscript is in using the polarizable charge equilibration (PQEq) method instead of the charge equilibration (QEq) method. In addition, the authors do not fit the partial charges and thus avoid problems with charge partitioning schemes. Tackling long-range interactions with ML potentials is an active research area, and this manuscript presents an important contribution to the field. Nevertheless, the authors should address the following points before publication.”

Authors’ reply:

We would like to thank the reviewer for the thorough review and kind comments. We will carefully address all the points raised and incorporate the necessary improvements in our revised manuscript. Your expert insights are invaluable in enhancing the quality of our manuscript.

Q2. “In message-passing graph neural networks, the graph cut-off should not be confused with the cut-off of classical force fields since each message passing/interaction layer increases the receptive field. The authors use a NequIP model with six equivariant layers. Thus, the effective cut-off is not 5 Å but can extend up to $6 * 5$ Å. This point should be made clear in the text, i.e., in the introduction where the authors discuss the “MLIPs with a cut-off of around 5 Å” and in the results section when the authors are comparing their results with strictly local ML potentials such as HDNNP, where the employed cut-off for the symmetry functions is the same as the effective cut-off.”

Authors’ reply:

Thank you for your valuable comment regarding the interpretation of cut-off distances in message-passing graph neural networks. We will revise our manuscript to clarify this distinction in the following ways.

In the **Introduction** section, we now explicitly explain how message-passing mechanisms can extend the cut-off through layer-wise propagation. This addresses the potential confusion between our nominal cut-off and the actual effective receptive field of message-passing graph neural network potential. Subsequently, we further demonstrate that implicit message-passing mechanisms may also face challenges in two molecules separated by a distance beyond the cut-off value,

While existing local MLIPs with a cut-off of around 5 Å perform well in simulating interactions within localized chemical environments, they may fail to capture long-range phenomena. This hinders their ability to understand and elucidate the behaviours of complex materials [22]. One approach is to *implicitly* incorporate the effects of long-range interactions through message-passing mechanisms [23] which *can extend the cut-off through layer-wise*

propagation. However, if certain parts of the system are disconnected on the graph, such as two molecules separated by a distance beyond the cut-off value, the message-passing scheme becomes ineffective [24]. A promising direction involves investigating the potential advantages of explicitly including long-range interactions in the model formulation.

And in the **Results** section,

... We demonstrate the advantage of integrating equivariant message-passing networks with explicitly polarizable long-range interaction models by benchmarking against both 4G-HDNNP, *a local MLIP with explicit long-range interactions*, and NequIP, which *implicitly* captures long-range effects through its *message-passing mechanism*...

Q3. “Related to the above, the test cases developed by Ko *et al.* involve very small systems and make sense for strictly local ML potentials but make little sense with a 6-layer graph neural network (GNN), where the receptive field likely encompasses the whole system. To identify the PQEq contribution, the authors should extend **Table 1** to include results using only GNN and using GNN+DFT-D3. On the other hand, a comparison with 2G-HDNNP and 3G-HDNNP is not necessary since these results were already published.”

Authors’ reply:

Thank you for this insightful comment which raises an important methodological consideration about our test systems and evaluation approach. We agree that clearer isolation of the PQEq contribution would strengthen our analysis. Following your suggestion, we have extended **Table 1** to include results using only the GNN component (Maruf’s NequIP, arXiv:2503.17949 <https://arxiv.org/html/2503.17949v1>). We do not include D3 correction to any of the methods because the dataset does not incorporate it. This additional analysis will provide a more comprehensive picture of how each component contributes to the model’s performance and will help readers better understand the specific benefits of the PQEq component in our approach.

Regarding the comparison with 2G-HDNNP and 3G-HDNNP, we appreciate your guidance, and we have removed these comparisons from **Table 1** to focus on the novel aspects of our work.

In the **Results** section,

Table 1. The comparison of root means square test error metrics on the different charge-state datasets.

Dataset		4G-HDNNP	Maruf’s NequIP	This work
C ₁₀ H ₂ /C ₁₀ H ₃ ⁺	Energy (meV/atom)	1.19	1.33	0.44
	Force (eV/Å)	0.078	0.071	0.023
Na _{8/9} Cl ₈ ⁺	Energy (meV/atom)	0.48	N. A.	0.23
	Force (eV/Å)	0.033	N. A.	0.006

Dataset		4G-HDNNP	Maruf's NequIP	This work
Ag ₃ ^{+/-}	Energy (meV/atom)	1.32	498.39	4.87
	Force (eV/Å)	0.032	2.145	0.028
Au ₂ -MgO	Energy (meV/atom)	0.22	1.03	0.14
	Force (eV/Å)	0.066	0.034	0.021
BTA-Cu	Energy (meV/atom)	N. A.	0.48	0.30
	Force (eV/Å)	N. A.	0.008	0.004
BTA (H ₂ O)-Cu	Energy (meV/atom)	N. A.	0.71	0.19
	Force (eV/Å)	N. A.	0.010	0.005

...

As for the charged Ag clusters, due to their small system size and lacking long-range effects, the entire structure falls within the typical cutoff radius used by the MLIP. The energy dependence on the overall charge state of clusters leads to degeneracy between atomic structures and potential energy surfaces, resulting in poor performance of charge-independent NequIP model. We acknowledge that for such small, highly charged systems, 4G-HDNNP approach to learning charge equilibration parameters is more convenient. Nonetheless, we also achieved reduced force error with the fixed physical parameters compared to 4G-HDNNP. We re-optimized the geometries of Ag₃^{+/-} using our model, with results shown in **Fig. S1 c**. Our model accurately predicts the structures, with root mean squared displacements of only 0.002 Å and 0.006 Å compared to DFT-optimized structures.

...

Our comprehensive evaluation demonstrates that explicit incorporation of physical long-range interactions significantly enhances the performance of MLIPs across diverse charge-state systems. Notably, our proposed framework outperforms 4G-HDNNP in most cases for energies and forces without fitting partial charges. This advantage likely stems from a more physically meaningful partition of the total energy.

Q3. “It is unclear if the resulting partial charges should be interpreted as physical partial charges, or a quantity correlated with the partial charge. The authors should report the errors for partial charge prediction for Ko *et al.* cases. Of course, the agreement cannot be as good as when training on a partial charge, and given the differences between the charge partitioning schemes, this is also not the goal. Nevertheless, the authors should compare the partial charge prediction errors with differences between charge partitioning schemes to clarify the meaning of predicted partial charges.”

Authors' reply:

Thank you for your insightful comment regarding the interpretation of partial charges in our model. The partial charges in our model should be interpreted primarily as parameters that

effectively reproduce the electrostatic contribution to the total energy while maintaining physically reasonable characteristics.

In the **Results** section,

... Although PQEq partial charges are not directly comparable to DFT Hirshfeld ones, we compare them to show their relevance. *Varying degrees of agreement* across different chemical systems are observed as shown in **Table S2**...

And in **SI**,

When comparing our PQEq partial charges with reference DFT Hirshfeld charges, we observe varying degrees of agreement across different chemical systems as shown in **Table S2**. While *some systems show remarkably agreement*, particularly carbon-based systems and metal-oxide interfaces, it is important to acknowledge that direct numerical comparisons between different charge partitioning schemes have inherent limitations. The different partitioning philosophies mean that *perfect agreement is neither expected nor necessarily desirable as a validation metric*.

Table S2. The mean absolute difference of DFT Hirshfeld charges and PQEq partial charges for the different charge state dataset, expressed in e.

	C ₁₀ H ₂ /C ₁₀ H ₃ ⁺	Na _{8/9} Cl ₈ ⁺	Ag ₃ ^{+/-}	Au ₂ -MgO
Difference	0.0182	0.396	0.210	0.0364

Furthermore, we would like to clarify that comparing the absolute values of partial charges has limited significance. Instead, the *relative trends* and patterns in charge distribution can meaningfully reflect the consistency of physical models. When applying an electric field perpendicular to the molecular plane of water molecules, we observed that the changes in PQEq partial charges closely track those of DFT Hirshfeld charges, despite differences in their absolute values. Rather, it is the response of these charges to perturbations and their collective contribution to system properties that provides more meaningful validation of our model's physical consistency.

In the **Results** section,

... We also compared the response of O atom partial charges across DFT Hirshfeld calculations, our model (PQEq model), and the conventional QEq model as shown in **Fig. S4**. *Excellent agreement between PQEq model and DFT* is observed, while QEq model completely lacks response...

And in **SI**,

In the polarization of static water molecules under an electrostatic field shown in **Fig. 2b**, the charge equilibration (QEq) model demonstrates limitations in handling intramolecular polarization when electric field strength varies in the *x*-axis. The QEq model represents atoms as either point charges or Gaussian charges, which restricts its ability to accurately capture molecular polarization effects. In contrast, the PQEq model separates atomic cores and shells

(Eq. S6), enabling it to effectively account for intramolecular polarization phenomena. To illustrate this difference, we present in Fig. S4 the variations in O atom partial charge differences ($q - q_{\min}$) across different models as a function of the applied electric field strengths. It is important to note that our analysis does not directly compare the absolute values of partial charges, but rather employs *relative values as a reference point*. Under different charge partitioning schemes, comparing absolute values lacks meaningful interpretation, whereas the relative changes more accurately reflect how various methods respond to the applied electrostatic fields. Fig. S4 demonstrates that *our model shows consistency with first principle calculations* in terms of charge variations. This comparison validates that our approach accurately captures the electronic response behaviour under various conditions.

Fig. S4 The variations in O atom partial charge differences ($q - q_{\min}$) between DFT Hirshfeld calculations, our model (PQEq model), and the conventional QEq model.

Q5. “For the universal machine learning potential model with and without polarizable long-range interactions, the energy/force/stress errors on the test set are very similar. The authors should include the results of the universal model without polarizable long-range interactions also for the reported applications, *i.e.*, Mechanical properties, Ionic diffusivity *etc.*”

Authors’ reply:

Thank you for this insightful comment regarding the comparative performance of our models. While our universal models with and without polarizable long-range interactions show similar error metrics on the test set, the model incorporating long-range interactions consistently achieves lower errors. Following your suggestion, we extend our analysis to include results from the universal model without the interactions (*w/o*-lr model) for the reported applications, where the *superiority of our model with the interactions still holds*.

For the mechanical properties, in the **Results** section,

... **Fig. 2 e-g** illustrate the comparison of the bulk modulus B determined by the pretrained model and DFT. Our model demonstrated impressive performance, achieving an R^2 of 0.94 with the Voigt approach [49] B_V and Hill average method [50] B_{VRH} , and 0.93 using the Reuss

method [51] B_R . In comparison, as shown in **Fig. S2**, the *w/o-lr* model achieved lower R^2 values of 0.89, 0.86, and 0.88 for B_V , B_R , and B_{VRH} , respectively.

Fig. 2 c-e, Comparison of bulk modulus benchmarking from first principles calculations and our model: **c**, Voigt approach, **d**, Reuss method, and **e**, Hill average.

And **SI** detailed the bulk modulus predicted by the *w/o-lr* model,

Fig. S2 Comparison of bulk modulus from first principles calculations and the model without polarizable long-range interactions (*w/o-lr* model): **a**, Voigt approach, **b**, Reuss method, and **c**, Hill average.

For the ionic diffusivity, in the **Results** section,

... Owing to the efficiency of our pretrained model, we are able to conduct MD simulations on the c-LLZO comprising 64 formula units, with a duration of 2 ns for each temperature range from 800 K to 1800K. **Fig. 3 a** presents the Arrhenius plots derived from our model, *w/o-lr* model, and AIMD [52] simulations. The diffusion coefficients were determined from the slope of the logarithmic mean square displacements (MSD) versus logarithmic time within the Fickian regime [54]. MSD curves of our model are depicted in **Fig. 3 b**. Compared to previous work using AIMD simulations, our model clearly reproduces the diffusion coefficients and activation energy. *The performance surpasses that of the w/o-lr model which lacks explicit long-range interactions.*

Fig. 3 a, Crystal structure of $Ia\bar{3}d$ $\text{Li}_7\text{La}_3\text{Zr}_2\text{O}_{12}$ and Arrhenius plots depicting the lithium-ion diffusion coefficients across varying temperatures. The dark blue polyhedron signifies La located at the 24(c) site and the light brown polyhedron indicates Zr at the 16(a) site. Li fraction occupies the 24(d) and 96(h) sites. Predicted diffusion coefficients of AIMD [51], our model, and *w/o-lr* model complete with error bars, are presented to calculate activation energies. **b**, 2-ns mean square displacements using our model of lithium-ion in c-LLZO with different temperatures ranging from 800 K to 1800 K in an increment of 200K. The linear dashed grey lines, with a slope of 1, are also plotted.

Q6. “Figure 2 b-c: the authors should increase the x -axis range to 12 Å to actually see the long-range scaling behaviour.”

Authors’ reply:

We appreciate the reviewer’s valuable suggestion to extend the x -axis range. However, we would like to clarify two important points regarding our current presentation. First, our primary focus in these figures is to demonstrate the behaviour *beyond the graph cut-off distance of 5 Å*, specifically highlighting how the model without long-range interactions (*w/o-lr* model) *fail* to distinguish between Na-Na and Na-Cl dimer systems. Second, even DFT, our target, *fails to correctly describe molecular dissociation limit* [*Chem Rev* **112**, 289 (2012) <https://doi.org/10.1021/cr200107z>].

We note that our choice of upper limit is consistent with other recent publications in the field. For example, a recent study by Shaidu *et al.* [*npj Comput Mater* **10**, 47 (2024) <https://doi.org/10.1038/s41524-024-01225-6>] similarly presents DFT calculations only up to 5.3 Å when demonstrating their QEq-based model’s capability to differentiate dimer interactions (**Fig. 5** of that reference).

Given these constraints, we hope the reviewer acknowledge that the current range can adequately demonstrate the critical difference between the models while maintaining scientific integrity regarding our reference data.

Q7. “Are the model training details described in the Methods section the same for the universal equivariance neural network potential and the test cases developed by Ko *et al.*? If not, please add training details for Ko *et al.* cases.”

Authors’ reply:

We thank the reviewer for pointing out this important issue. Indeed, we used different structures in our initial manuscript. We now provide all the details in **Table S3, SI**.

As in **Q3**, we have included results on Ko’s cases from a model containing only the GNN component, taken from the referenced work [arXiv:2503.17949 <https://arxiv.org/html/2503.17949v1>]. For comparison, we retrained our long-range interaction model using the same number of GNN layers as specified in that reference to ensure a fair comparison.

In the **Results** section,

... The results are presented in **Table 1**, *with detailed neural network architectural specifications* provided in the **Table S3**.

And in **SI**,

As for training the different charge state dataset, the neural network architectures are provided in **Table S3**. For feature multiplicity, we consistently employed 32 across all models to maintain all parameters in accordance with the reference [6].

Table S3. The architectures for training different charge state dataset. l_{\max} is the maximum rotation order.

Dataset	Settings	Maruf’s NequIP	This work
$C_{10}H_2/C_{10}H_3^+$	Cutoff (Å)	5.00	5.00
	Number of layers	8	8
	l_{\max}	1	1
$Na_{8/9}Cl_8^+$	Cutoff (Å)	N.A.	5.00
	Number of layers	N.A.	5
	l_{\max}	N.A.	2
$Ag_3^{+/-}$	Cutoff (Å)	5.29	5.29
	Number of layers	8	8
	l_{\max}	2	2
Au_2-MgO	Cutoff (Å)	5.50	5.50
	Number of layers	6	6
	l_{\max}	2	2
BTA-Cu	Cutoff (Å)	5.00	5.00
	Number of layers	6	6
	l_{\max}	1	1
BTA (H ₂ O)-Cu	Cutoff (Å)	5.00	5.00
	Number of layers	6	6
	l_{\max}	1	1

We appreciate the reviewer's attention to this important methodological detail, which allows for a more rigorous comparison between the different model architectures. This expanded information provides readers with a clearer understanding of the modelling approaches and strengthens the validity of our comparative analysis.

Q8. "... MPtrj Dataset sourced from Materials Projects as the pretraining dataset ..." Why do you call this pretraining? which dataset was used in the subsequent training?"

Authors' reply:

Thank you for raising this terminology issue. Our use of the term "*pretraining*" was intended to convey that we have developed a universal model that can be directly applied for molecular dynamics simulations and other applications without further training. To clarify, the model trained on MPtrj serves as a general-purpose, ready-to-use potential. In the revised manuscript we all use "*training*".

In the **Methods** section,

To train the universal equivariance neural network potential, we used the MPtrj dataset [16] sourced from Materials Projects [48] as the *training* dataset...

Additionally, in the revised manuscript, we include examples of *fine-tuning*, which show how this pretrained model can be further refined for more specialized tasks requiring higher precision. This illustrates the versatility of our approach: the model can be used as-is for many applications but can also serve as a starting point for more specific applications through fine-tuning.

In the **Results** section,

... Our model also exhibits remarkable adaptability through *fine-tuning* on high-temperature datasets, with performance metrics detailed in **Table S6**.

And in **SI**,

... To evaluate our model's capacity to learn high-temperature data representations, we fine-tuned our pretrained universal model on the MPF-TP dataset. During this fine-tuning process, only the final two convolutional layers and the multilayer perceptron layers were set as trainable parameters, while all remaining layers were frozen.

Our fine-tuning results demonstrate remarkable improvements on high-temperature, non-equilibrium systems as shown in **Table S6**. The fine-tuned model achieves *state-of-the-art accuracy* across all prediction targets on the MPF-TP dataset...

Q9. "Why is the shell charge of all atoms set to 1?"

Authors' reply:

The shell charge value of 1 for all atoms was chosen following established precedent in the field of original PQEq paper [*J. Chem. Phys.* **146**, 124117 (2017) <https://doi.org/10.1063/1.4978891>]. In the **Method** section of this paper, subsection A “**Polarizable charge equilibration (PQEq) model**”, they explicitly state: “...*A shell charge of $Z_i = 1$ is used for all atoms.*”.

We add the reference in **SI**,

... Here z is set to 1 for all atoms [1].

This standardized value simplifies the model while still adequately capturing polarization effects across different chemical environments. The physical interpretation is that the model *adjusts the relative displacement of the shell rather than varying the shell charge itself*, effectively modulating the induced dipole moment through positional shifts rather than charge magnitude changes.

Q10. “Missing directly related reference: 10.48550/arXiv.2501.19179”

Authors’ reply:

Thank you for bringing this reference to our attention. This paper combines the QEq method with equivariant message-passing graph neural networks, which is indeed related to our approach. We have added this reference as suggested to provide readers with a more complete view of the current landscape in this field. We appreciate the opportunity to acknowledge this relevant work and position our research within the broader context of recent developments in combining charge equilibration methods with machine learning frameworks.

In the **Introduction** section,

... In these methods, charges are usually obtained through charge equilibration (QEq) [25] principles, including the Charge Equilibration via Neural Network Technique (CENT) method [26], Fourth-Generation [27, 28] High-Dimensional Neural Network Potential (4G-HDNNP), and *Charge Equilibration Layer for Long-range Interactions (CELLI)* [29] method.

29. Fuchs, P., M. Sanocki, and J. Zavadlav, *Learning Non-Local Molecular Interactions via Equivariant Local Representations and Charge Equilibration*. arXiv preprint arXiv:2501.19179, 2025.

Q11. “The authors should relate their work with alternative long-range approaches not employing the charge equilibration schemes, e.g., Ewald summation, Latent Ewald Summation, LSR-MP, NeuralP3M, and So3krates”

Authors’ reply:

We appreciate the suggestion to relate our work with alternative long-range approaches that do not employ charge equilibration schemes. We have added them in the **Discussion** section to address this importance.

... Additionally, several approaches that do not require charge equilibration process provide valuable insights. Latent Ewald Summation [24] captures long-range interactions by learning hidden variables from local descriptors and applying Ewald summation to these variable. LSR-MP [79] employs a fragmentation-based approach with hierarchical message passing between atoms and fragments to learn the long-range interactions. NeuralP³M [80] introduces mesh points alongside atoms to discretize space and capture long-range effects. SO3krates [81] utilizes spherical harmonic coordinates and equivariant attention for interactions at arbitrary length scales. In contrast to these approaches, our universal model explicitly simulates charge redistribution through a physics-driven framework that maintains a transparent connection to the underlying physical principles of charge polarization and electrostatic interactions. This design choice enhances interpretability while preserving computational efficiency. A promising direction for future work lies in exploring strategic integrations of our approach with these complementary methodologies to develop even more comprehensive and powerful universal models.

Q12. “Remarks on code availability:

I did not run the code but it looks OK.”

Authors’ reply:

We appreciate the reviewer’s assessment of our code. We have open-sourced the complete model codebase in the new repository and supplemented it with additional tutorials and fine-tuning examples to facilitate implementation for users.

Code availability

The JAX implementation code is available through <https://github.com/reaxnet/reaxnet>.

Response to reviewer #2

Q1. “Overall, a manuscript which should be very heavily revised - but that might decrease any novelty reported such that it would fall outside of the scope of the journal. Maybe best to reject.”

Authors’ reply:

Thank you for your feedback on our manuscript. We deeply respect your thoughtful review and have taken your comments very seriously. We have made substantial revisions to address the concerns you raised, and we believe the revised manuscript still demonstrates adequate novelty.

Q2. “- What are the noteworthy results?”

There are some interesting simulations using the MLIP technology developed by the authors - but there is no reliable benchmarking carried out to compare to other state-of-the-art methods. Since the focus of the manuscript is a supposedly superior method, this is a claim that should have been proven by the authors.”

Authors’ reply:

We appreciate the reviewer’s feedback regarding the need for comprehensive benchmarking against state-of-the-art methods. In response to this important point, we have significantly expanded our comparative analysis in the revised manuscript.

We have conducted additional analyses, comparing our performance with CHGNet and MACE-MP-0, both of which underwent similar training on the zero-temperature MPtrj dataset. These comparative evaluations were performed across the benchmark datasets proposed in MatterSim. Our findings demonstrate that our model consistently achieves *lower force errors* across all datasets compared to these two universal models, while maintaining reasonable energies and stresses errors.

In the **Results** section,

We also perform benchmarking of our model with other universal models on the test sets provided by the reference [19]. **Table S5** demonstrates the exceptional performance of our model across diverse test sets in *force prediction capabilities, consistently outperforming* both CHGNet [16] and MACE-MP-0 [18], while maintaining reasonable energy and stress predictions. We excluded MatterSim [19] results from our comparative analysis as its training dataset incorporates high-temperature data.

And in **SI**,

We also conducted benchmarking with other universal models on the test sets provided by the reference [11]. For near-equilibrium evaluations, two randomly sampled collections are used: 1,000 structures from the MPtrj dataset [7] (MPtrj-random-1k) and 1,000 from

Alexandria dataset [12] (Alexandria-1k). These assess model accuracy on energetically favourable configurations, which is crucial for predicting material stability. Three increasingly challenging thermodynamic perturbation (TP) test sets evaluate performance under varied temperature and pressure conditions with non-equilibrium atomic configurations. MPF-Alkali-TP focuses on potential ionic conductors, comprising 50 randomly selected compounds containing at least one alkali metal paired with electronegative elements (N, O, P, S, Se, or halogens) from MPF2021 dataset [13]. This dataset specifically tests model accuracy for materials with significant ionic character. MPF-TP broadens the scope with 50 randomly selected compounds from the MPF2021 dataset without elemental restrictions, presenting a more diverse chemical challenge. Random-TP represents the most demanding test, constructed by arbitrarily positioning 20 atoms of random elements within simulation boxes, creating hypothetical structures that test extreme generalization capability. **Table S5** demonstrates the performance of our model across diverse test sets, *particularly highlighting its superior force prediction capabilities*. Our model achieves remarkable force prediction accuracy, consistently outperforming both CHGNet [7] and MACE-MP-0 [14] (all three models were trained on MPtrj dataset) across all benchmark datasets with significantly lower MAEs – nearly half the error of competing models in many cases. Impressively, our model demonstrated robust generalization capability across increasingly challenging thermodynamic perturbation scenarios. Although we observe slightly higher energy prediction MAEs compared to the two models, its performance across the entire spectrum of materials conditions is improved particularly for force predictions critical for molecular dynamics simulations. It is important to note that our comparison was limited to universal models trained exclusively on zero-temperature dataset (MPtrj) and did not include MatterSim [11], which incorporates high-temperature datasets...

Table S5. Comparison of mean absolute errors (MAE) in the test set generated by the reference [11].

Test set	MAE	CHGNet	MACE-MP-0	Our model
MPtrj-random-1k	Energy (eV/atom)	0.027	0.015	0.019
	Force (eV/Å)	0.120	0.117	0.066
	Stress (GPa)	0.290	0.468	0.271
Alexandria-1k	Energy (eV/atom)	0.150	0.092	0.115
	Force (eV/Å)	0.108	0.095	0.064
	Stress (GPa)	1.643	0.160	1.055
MPF-Alkali-TP	Energy (eV/atom)	0.250	1.351	0.280
	Force (eV/Å)	1.636	15.819	0.717
	Stress (GPa)	12.625	25.723	13.841
MPF-TP	Energy (eV/atom)	0.254	256.340	0.521
	Force (eV/Å)	3.313	1506.854	0.768
	Stress (GPa)	25.208	202.093	15.445
Random-TP	Energy (eV/atom)	0.506	9.184	3.338
	Force (eV/Å)	3.950	88.327	1.199
	Stress (GPa)	7.230	19.224	33.282

Furthermore, after fine-tuning, our model can achieve even more accurate predictions on high-temperature datasets, demonstrating its adaptability to diverse thermodynamic conditions.

In the Results section,

... Our model also exhibits remarkable adaptability through fine-tuning on high-temperature datasets, with performance metrics detailed in **Table S6**.

And in **SI**,

Our fine-tuning results demonstrate remarkable improvements on high-temperature, non-equilibrium systems as shown in **Table S6**. The fine-tuned model *achieves state-of-the-art accuracy* across all prediction targets on the MPF-TP dataset. This suggests that while our universal model can maintain reasonable force accuracy under a wide range of conditions, targeted fine-tuning is more likely to balance broad applicability with the superior accuracy required for specific applications.

Table S6. Comparison of mean absolute errors in the MPF-TP test set of the universal models and finetuned model.

	Energy (eV/atom)	Force (eV/Å)	Stress (GPa)
CHGNet	0.254	3.313	25.208
MACE-MP-0	256.340	1506.854	202.093
MatterSim (M3GNet)	0.036	0.431	1.318
MatterSim (Graphormer)	0.040	0.421	1.917
Our universal model	0.521	0.768	15.445
Fine-tuned model	0.027	0.203	0.976

We believe these additional benchmarking results now provide substantial evidence to support our claims regarding the method's capabilities.

Q3. “- Will the work be of significance to the field and related fields? How does it compare to the established literature? If the work is not original, please provide relevant references.

This work is mostly combining other, existing pieces of technology in a slightly different way. In short, a short-range MLIP is combined with Goddard's QEq scheme and D3 dispersion corrections. The closest similar is Behler's 4G-HDNNPs, but these don't use the equivariant message passing short range technology. Results presented here are marginally better, but perhaps only an incremental improvement. Presumably due to the better and more expressive short-range part. Behler's scheme is superior as it uses environment dependent parameters for the QEq component, unlike here, where these are fixed. The CENT method by Goedecker (which uses a similar approach to fit the energies directly) is not even mentioned.”

Authors' reply:

We appreciate the reviewer's insightful comparison of our work with existing technologies in the field. To address the points raised, we have expanded our benchmarking analysis to better demonstrate the novel contributions of our approach.

(1) For the improvement over existing works, to show that it is not just because of a more expressive short-range part, we compared to a very recent arXiv work [arXiv: 2503.17949v1 <https://arxiv.org/html/2503.17949v1>], which used the same NequIP neural network as us, yet similar long-range interaction scheme like 4G-HDNNP (marked as arXiv-lr work). We extracted their results and compared to ours as shown in **Table R1**.

Table R1 Energy (meV/atom) and force (eV/Å) root mean squared errors of the arXiv-lr work and our work.

Dataset		arXiv-lr work	This work
C ₁₀ H ₂ /C ₁₀ H ₃ ⁺	Energy (meV/atom)	0.84	0.44
	Force (eV/Å)	0.055	0.023
Ag ₃ ^{+/-}	Energy (meV/atom)	0.82	4.87
	Force (eV/Å)	0.021	0.028
Au ₂ -MgO	Energy (meV/atom)	0.17	0.14
	Force (eV/Å)	0.022	0.021
BTA-Cu	Energy (meV/atom)	0.43	0.30
	Force (eV/Å)	0.002	0.004
BTA (H ₂ O)-Cu	Energy (meV/atom)	0.28	0.19
	Force (eV/Å)	0.011	0.005

We achieved some improvements *with the fixed PQEq parameters* over that work, indicating that our combined approach can indeed provide insights into this field.

(2) Regarding the 4G-HDNNP/CENT methods, which learn environment-dependent QEq parameters, we have provided a detailed discussion in the **Discussion** section. While employing an additional neural network to learn environmental parameters could potentially achieve higher accuracy, efficiency remains critically important for universal models designed for large-scale molecular dynamics simulations. The literature also indicates that *QEq parameters do not change significantly* during molecular dynamics simulations. A key focus of our future work will be combining efficiency and accuracy to develop even more generalizable models that maintain computational performance while further enhancing predictive capabilities.

In the **Discussion** section,

We acknowledge that our model aims to achieve a balance between computational efficiency and accuracy. Several advanced machine learning approaches for long-range interactions may achieve higher accuracy for specific, complex systems, including higher-order charge expansion methods and simultaneous optimization of QEq parameters [32, 79] in each

step. However, the more accurate methods incur higher computational complexity. The method that dynamically solve charge equilibration parameters at each step require additional cost for MD simulations. It is also observed that the *charge equilibration parameters do not change significantly* during MD simulations for that method [79], suggesting that expanding polarization charges to higher orders while using fixed, *a priori* parameters may achieve higher accuracy with tolerable computational overheads.

For reference, the dynamic changes of QEq parameters (χ and η) during molecular dynamics simulations in the reference [arXiv:2411.04062 <https://arxiv.org/html/2411.04062v1>] are reproduced in **Fig. R1**,

Fig. R1 Variation with time of χ and η at the C, H and O sites through molecular dynamics. The figure is extracted from arXiv:2411.04062.

For these parameters in different systems, fixed parameters work generally well (as **Q2** detailed and shown in **Table S5**). For extreme cases (small, charged structures like Ag clusters), we admit adjusting parameters would give better results, as in the revised manuscript.

In the **Results** section,

...As for the charged Ag clusters, due to their small system size and lacking long-range effects, the entire structure falls within the typical cutoff radius used by the MLIP. The energy dependence on the overall charge state of clusters leads to degeneracy between atomic structures and potential energy surfaces, resulting in poor performance of charge-independent NequIP model. We acknowledge that for such small, highly charged systems, *4G-HDNNP approach to learning charge equilibration parameters is more convenient*. Nonetheless, we also *achieved reduced force error* with the *fixed physical parameters* compared to 4G-HDNNP. We re-optimized the geometries of $\text{Ag}_3^{+/-}$ using our model, with results shown in **Fig. S1 c**. Our model accurately predicts the structures, with root mean squared displacements of only 0.002 Å and 0.006 Å compared to DFT-optimized structures...

(3) In the **Introduction** section, the CENT method is mentioned as it points out that learning partial charges is not necessary,

... In these methods, charges are usually obtained through charge equilibration (QEq) [25] principles, including *the Charge Equilibration via Neural Network Technique (CENT)* method [26], Fourth-Generation [27, 28] High-Dimensional Neural Network Potential (4G-HDNNP), and Charge Equilibration Layer for Long-range Interactions (CELLI) [29] method. ... The ambiguity in DFT-assigned charges suggests that directly learning of them *may be inessential* for constructing accurate interatomic potentials [26, 31-32] ...

Q4. “- Does the work support the conclusions and claims, or is additional evidence needed?

A lot of data which the results are compared to here are cherry-picked (there is hardly or no any mention of significant universal force fields, such as MACE-MP and its later improvements). This is a quickly moving field.”

Authors’ reply:

We appreciate your advice. As mentioned above, we have supplemented our benchmarking with CHGNet and MACE-MP-0, which were also trained on zero-temperature datasets, using the challenging test sets proposed by MatterSim. We did not compare with MatterSim itself because it includes high-temperature datasets in its training process, which would not allow for a fair comparison with our model. Notably, in these additional benchmark tests, our model demonstrates lower force errors than both CHGNet and MACE-MP-0 while maintaining reasonable energy and stress errors, further validating the effectiveness and advantages of our approach.

In **SI**,

We also conducted benchmarking with other universal models on the test sets provided by the reference [11]. For near-equilibrium evaluations, two randomly sampled collections are used: 1,000 structures from the MPtrj dataset [7] (MPtrj-random-1k) and 1,000 from Alexandria dataset [12] (Alexandria-1k). These assess model accuracy on energetically favourable configurations, which is crucial for predicting material stability. Three increasingly challenging thermodynamic perturbation (TP) test sets evaluate performance under varied temperature and pressure conditions with non-equilibrium atomic configurations. MPF-Alkali-TP focuses on potential ionic conductors, comprising 50 randomly selected compounds containing at least one alkali metal paired with electronegative elements (N, O, P, S, Se, or halogens) from MPF2021 dataset [13]. This dataset specifically tests model accuracy for materials with significant ionic character. MPF-TP broadens the scope with 50 randomly selected compounds from the MPF2021 dataset without elemental restrictions, presenting a more diverse chemical challenge. Random-TP represents the most demanding test, constructed by arbitrarily positioning 20 atoms of random elements within simulation boxes, creating hypothetical structures that test extreme generalization capability. **Table S5** demonstrates the performance of our model across diverse test sets, *particularly highlighting its superior force prediction capabilities*. Our model achieves remarkable force prediction accuracy, consistently outperforming both CHGNet [7] and MACE-MP-0 [14] (all three models were trained on MPtrj dataset) across all benchmark datasets with significantly lower MAEs – nearly half the error of

competing models in many cases. *Impressively, our model demonstrated robust generalization capability across increasingly challenging thermodynamic perturbation scenarios.* Although we observe slightly higher energy prediction MAEs compared to the two models, its performance across the entire spectrum of materials conditions is improved particularly for force predictions critical for molecular dynamics simulations. It is important to note that our comparison was limited to universal models trained exclusively on zero-temperature dataset (MPtrj) and did not include MatterSim [11], which incorporates high-temperature datasets ...

Table S5. Comparison of mean absolute errors (MAE) in the test set generated by the reference [11].

Test set	MAE	CHGNet	MACE-MP-0	Our model
MPtrj-random-1k	Energy (eV/atom)	0.027	0.015	0.019
	Force (eV/Å)	0.120	0.117	0.066
	Stress (GPa)	0.290	0.468	0.271
Alexandria-1k	Energy (eV/atom)	0.150	0.092	0.115
	Force (eV/Å)	0.108	0.095	0.064
	Stress (GPa)	1.643	0.160	1.055
MPF-Alkali-TP	Energy (eV/atom)	0.250	1.351	0.280
	Force (eV/Å)	1.636	15.819	0.717
	Stress (GPa)	12.625	25.723	13.841
MPF-TP	Energy (eV/atom)	0.254	256.340	0.521
	Force (eV/Å)	3.313	1506.854	0.768
	Stress (GPa)	25.208	202.093	15.445
Random-TP	Energy (eV/atom)	0.506	9.184	3.338
	Force (eV/Å)	3.950	88.327	1.199
	Stress (GPa)	7.230	19.224	33.282

Q5. “- Are there any flaws in the data analysis, interpretation and conclusions? - Do these prohibit publication or require revision?”

Plenty of misinterpretation/misrepresentation, listed below. With very careful revision maybe these can be rectified, but most likely it will further reduce any claims of superiority and/or novelty.”

Authors’ reply:

We sincerely thank the reviewer for their thorough assessment and for highlighting potential issues with our data analysis, interpretation, and conclusions. We take these concerns very seriously and have carefully addressed each point raised in the detailed comments below. Universal machine learning potentials with polarizable long-range interactions are indeed still in their nascent stage, which underscores the significant contribution of our work. Through our meticulous revisions, we believe that this novelty is not diminished.

Q6. “- Is the methodology sound? Does the work meet the expected standards in your field?

Mostly.”

Authors’ reply:

We appreciate the reviewer’s acknowledgment that our methodology is generally sound. We have further revised our manuscript to ensure that it aligns with your expectations.

Q7. “- Is there enough detail provided in the methods for the work to be reproduced?

Yes, and the code is publicly available (I have not tested, though).”

Authors’ reply:

We appreciate the reviewer’s assessment of our code. We have open-sourced the complete model codebase in the new repository and supplemented it with additional tutorials and fine-tuning examples to facilitate implementation for users.

Code availability

The JAX implementation code is available through <https://github.com/reaxnet/reaxnet>.

Q8. “Do not anthropomorphize mathematical models (e.g. p2/139)

‘subtle fluctuations’ - these are not subtle (it is meaningless in this context) - they are dynamic fluctuations”

Authors’ reply:

We thank the reviewer for this feedback regarding our language. We have adopted this precise scientific terminology as in the **Introduction** section,

... These behaviours encompass electrostatic interactions, involving forces between charged particles, and dispersion terms, which arise from *dynamic fluctuations* in electron distribution within molecules or atoms.

Q9. “Terms in Eq. 1 should be explained immediately after in the text.”

Authors’ reply:

We thank the reviewer for this suggestion regarding the clarity of our presentation. We have reorganized the structure of our manuscript to ensure that all terms in **Eq. 1** are explained in the text immediately following the equation.

In the **Results** section,

...The potential energy is expressed as the sum of the second-order expansion [30] with respect to charge fluctuations and the density functional theory [39] D3 (DFT-D3) van der Waals dispersion energies correction term [40, 41] E_{D3} ,

$$\begin{aligned}
 E_{\text{pot}} &= \sum_i \left(E_0^i(\mathbf{r}_i, Z_i) + \chi_i^0 q_i + \frac{1}{2} \eta_i^0 q_i^2 + \frac{1}{2} K_s^i |\mathbf{r}_{ic} - \mathbf{r}_{is}|^2 \right) \\
 &\quad + \sum_{ik>jl} C_{ik,jl}(\mathbf{r}_{ik,jl}) q_{ik} q_{jl} + E_{D3} \\
 &= \sum_i E_0^i + E_{\text{PQEq}} + E_{D3}.
 \end{aligned}
 \tag{1}$$

The zeroth-order atomic energy E_0^i corresponds the final layer scalar output of the equivariant graph neural networks, while the higher-order self- and interatomic Coulomb interactions represent the polarizable long-range electrostatic interactions E_{PQEq} . To account for charge transfer and polarization effects, **partial atomic charge** q_i of atom i is the sum of the **nuclear core charge** q_{ic} and **shell charge** q_{is} , both of which assume a Gaussian charge distribution form. The first-order coefficients χ^0 are **electronegativities**, commonly defined as half of the sum of ionization potential (IP) and electron affinities (EA). The second-order coefficients η^0 signifies **idempotential** or chemical **hardness**, defined as $IP - EA$. The **spring constant** K_s^i denotes the isotropic harmonic connectivity between the **shell position** \mathbf{r}_{is} and **core position** \mathbf{r}_{ic} (equal to \mathbf{r}_i) of atom i . The **Gaussian electrostatic energy** is given by $C(\mathbf{r}_{ik,jl}) q_{ik} q_{jl}$, where i and j are the atomic indices, and k and l represent the core (c) or shell (s), respectively. The displacement vector is given by $\mathbf{r}_{ik,jl} = \mathbf{r}_{ik} - \mathbf{r}_{jl}$. The detailed derivation of the atomic force and its response to the external electric field can be found in the **SI...**

Q10. “Ref. 24 is incomplete”

Authors’ reply:

We sincerely thank the reviewer for bringing this bibliographic issue to our attention. We have added the missing page number to the reference,

Naserifar, S., et al., *Polarizable charge equilibration model for predicting accurate electrostatic interactions in molecules and solids*. The Journal of Chemical Physics, 2017. **146**(12): p. 124117.

Q11. “Eqs. 2 and 3 do not follow. Since the energies also depend on the charges, which are determined self-consistently, it is far from obvious that eqs. 2 and 3 are true. There may be an expression analogous to the Hellmann-Feynman theorem but not shown here.”

Authors’ reply:

Thank you for raising this important point. Indeed, the energy does depend on the self-consistent determination of charges, whereby atomic types and positions determine both the core charges and shell displacements. We understand your concern that when calculating forces, we shall compute the negative gradient of energy with respect to position, while *automatic differentiation might calculate the total derivative*, causing the forces to be affected by the self-consistency process.

To be specific, in our implementation,

$$E_{\text{pot}} = E_{\text{pot}}(\mathbf{r}_{ic}; \mathbf{r}_{is}, q_i) \quad (\text{R1})$$

$$\mathbf{F}_i = -\partial_{\mathbf{r}_{ic}} E_{\text{pot}}, \quad (\text{R2})$$

where \mathbf{r}_{is}, q_i both depend on \mathbf{r}_{ic} and are solved self consistently. In our code, we *explicitly stop the gradient updates* with respect to \mathbf{r}_{ic} during this process. This approach ensures that the *forces are calculated correctly after the charge distribution has been converged*. For the stress (Eq. 3 in the main text), similar argues hold.

In Results section, we have refined our language,

... . After partial charges are obtained self-consistently, the forces on atoms \mathbf{F}_i and stress \mathbf{S}_{ab} on the cell can be calculated via automatic differentiation of potential energy (Eq. 1) with respect to atomic coordinates \mathbf{r}_i and strain \mathbf{s} . Note that *the gradient updates are stopped in the charge equilibration process* to ensure partial derivatives are taken ...

We also provided an example as in examples/basic.ipynb, section 5. Define the total energy function of our github repository (<https://github.com/reaxnet/reaxnet>),

```
def energy_fn(positions,
              nn_nbr,
              nb_nbr,
              ):
    """
    Total potential energy function

    Args:
    ----
    positions: jnp.array
               Atomic positions in Angstrom
    """

    # Update the neighbor lists first
    nn_nbr = nn_nbr.update(positions)
    nb_nbr = nb_nbr.update(positions)
    pe_nn = energy_fn_nn(positions, nn_nbr)

    charges, r_shell = charges_fn(jax.lax.stop_gradient(positions),
                                  nb_nbr)
    pe_nb = energy_fn_nb(positions, nb_nbr, r_shell, charges)

    return pe_nn + pe_nb, (charges, r_shell)
```

The `jax.lax.stop_gradient()` function will stop gradient updates when calculating the force via automatic differentiation.

To ensure that our equations and implementation are correct, we test the PQEq part of forces of ours and LAMMPS' (<https://github.com/sabernaserifar/RexPoN>) on the $\text{Li}_2\text{PO}_2\text{N}$ (mp-1020019) system. The results are listed in **Table R2** and **Table R3**.

Table R2. Forces comparison of LAMMPS and our code for $\text{Li}_2\text{PO}_2\text{N}$ (mp-1020019).

	Forces (eV/Å)					
	F_x^{LAMMPS}	F_x^{ours}	F_y^{LAMMPS}	F_y^{ours}	F_z^{LAMMPS}	F_z^{ours}
Li	0.036	0.036	-0.044	-0.044	0.001	0.002
Li	0.036	0.036	0.044	0.044	0.001	0.001
Li	-0.036	-0.036	0.044	0.044	0.001	0.002
Li	-0.036	-0.036	-0.044	-0.044	0.001	0.001
Li	0.036	0.036	-0.044	-0.044	0.001	0.001
Li	0.036	0.036	0.044	0.044	0.001	0.002
Li	-0.036	-0.036	0.044	0.044	0.001	0.001
Li	-0.036	-0.036	-0.044	-0.044	0.001	0.002
P	-0.002	-0.002	0.000	0.000	0.000	0.000
P	0.002	0.002	0.000	0.000	0.000	0.000
P	-0.002	-0.003	0.000	0.000	0.000	0.000
P	0.002	0.003	0.000	0.000	0.000	0.000
N	-0.057	-0.057	0.000	0.000	0.012	0.012
N	0.057	0.057	0.000	0.000	0.012	0.012
N	-0.057	-0.057	0.000	0.000	0.012	0.012
N	0.057	0.057	0.000	0.000	0.012	0.012
O	0.050	0.050	0.069	0.069	-0.008	-0.008
O	0.050	0.050	-0.069	-0.069	-0.008	-0.008
O	-0.050	-0.050	-0.069	-0.069	-0.008	-0.008
O	-0.050	-0.050	0.069	0.069	-0.008	-0.008
O	0.050	0.050	0.069	0.069	-0.008	-0.008
O	0.050	0.050	-0.069	-0.069	-0.008	-0.008
O	-0.050	-0.050	-0.069	-0.069	-0.008	-0.008
O	-0.050	-0.050	0.069	0.069	-0.008	-0.008

Table R3. Stresses comparison of LAMMPS and our code for $\text{Li}_2\text{PO}_2\text{N}$ (mp-1020019).

	S_{xx} (GPa)	S_{yy} (GPa)	S_{zz} (GPa)	S_{yz} (GPa)	S_{xz} (GPa)	S_{xy} (GPa)
Our code	0.3586	0.8228	0.0164	0.0000	0.0000	0.0000
LAMMPS	0.3586	0.8228	0.0164	0.0000	0.0000	0.0000

Q12. “Panels 2a, 2b and 2c must be united.”

Authors' reply:

We thank the reviewer for this suggestion on improving the presentation of our results. We have revised **Fig. 2** by uniting panels 2a, 2b, and 2c into a single cohesive panel with a shared axis system.

In the **Results** section,

... As shown in **Fig. 2a**, both models (our model and *w/o*-lr model) trained on MPtrj dataset accurately reproduce the equilibrium bond lengths of the two dimers. However, the *w/o*-lr model failed to differentiate interaction energies beyond the MLIPs cutoff distance, *i.e.*, 5.0 Å, while our model successfully captured the DFT potential energy surface throughout the entire distance range, demonstrating its capability in describing extended ionic interactions...

Fig. 2 a, The interaction energies for Na-Na (green) and Na-Cl (purple) dimers predicted by our model, *w/o*-lr model, and reference calculations using DFT.

Q13. “The QEq curve on 2d seems completely flat - this cannot be true, or the QEq must be extremely rigid.”

Authors’ reply:

We appreciate you raising this question. Indeed, the QEq model curve is completely flat, with only negligible floating-point errors that are virtually undetectable. We would like to emphasize that this example represents a *static* calculation of a water molecule under different electric field strengths, primarily designed to demonstrate the response of atomic polarization within the molecule to external electric fields. The definition of electric potential energy can be expressed as,

$$E_{\epsilon} = -\sum q_i \epsilon \cdot \mathbf{r}_i. \quad (\text{R3})$$

The water molecule *consistently remains in the yz-plane* with the electric field direction consistently along the *x-axis*. Given that the water dipole is perpendicular to the direction of the electric field, the electrostatic energy of QEq-based model *is always zero*.

This is a fundamental limitation of the traditional QEq-based model—it lacks the capability to respond to external electric fields when atomic positions remain fixed. In contrast, our

polarizable model captures this physical phenomenon correctly, as clearly demonstrated by their non-flat energy curves in **Fig. 2d (now Fig. 2b)**. This response highlights an important advantage of our model for simulations involving external electric fields or environments with varying electrostatic characteristics.

In the **Results** section,

... Due to the external *electrostatic field direction consistently being orthogonal to the molecule*, the electrostatic energy in the QEq-based model (like 4G-HDNNP) is 0 and is completely independent of field strength (*i.e.*, $\boldsymbol{\mu} \cdot \boldsymbol{\epsilon} = 0$, where $\boldsymbol{\mu}$, $\boldsymbol{\epsilon}$ are dipole moment and electric field, respectively). This is because the QEq model treats atoms in the molecule as point charges or Gaussian charges, *without distinguishing between core and shell components*...

Q14. “Computational cost being linear: I find that hard to believe. Long range interactions in periodic systems are typically treated using some version of the Ewald summation method or an equivalent approach. While there exist linear scaling implementations, there is no mention of these here.”

Authors’ reply:

We sincerely thank the reviewer for their correction. We regretfully acknowledge our oversight regarding the computational scaling. Due to the matrix inversion operations in our code, the scaling is indeed cubic rather than linear as we initially stated. We did not measure the scaling, instead, we unintentionally took the apparent trend as the y -axis upper limit was too large but the x -axis upper limit was too small. To fix this misleading presentation, we extend the system sizes being tested to reflect the cubic scaling behaviour. However, we would like to clarify that for the systems of tens of thousands of atoms that we have simulated, the computational cost remains acceptable, and we have still achieved high computational efficiency. This practical performance is sufficient for the material systems and properties investigated in this work.

For future work, we plan to improve our algorithm by implementing more efficient methods for solving the linear equation systems, such as conjugate gradient methods with appropriate preconditioning, which could potentially reduce the scaling behaviour. We have corrected the manuscript to accurately reflect the computational scaling of our current implementation and to properly contextualize its practical performance for the systems studied.

Again, we are extremely sorry for this misrepresentation.

In the **Results** section,

Despite the incorporation of additional long-range interaction calculations, our model maintains high computational efficiency. Performance benchmarks were conducted on a single NVIDIA H100 GPU to quantitatively assess the computational cost. As illustrated in **Fig. S6**, for a periodic system comprising 2160 atoms, our model requires approximately 0.07 s per

molecular dynamics time step, representing a significant improvement over conventional universal MLIPs which require approximately 0.91 s. This computational advantage extends to larger systems, as demonstrated by simulations of a 24000-atom system, where our model maintains efficiency with only 1.49 s per time step. Such computational performance enables nanosecond-scale molecular dynamics simulations of systems containing tens of thousands of atoms, making it practical for more realistic modelling of materials.

And in SI,

Our model introduces explicit polarizable long-range interactions while still maintaining good computational efficiency. To highlight the performance of this model, benchmarking was done on molecular dynamics simulations. Supercell models were generated from $\text{Li}_2\text{PO}_2\text{N}$ (mp-1020019) from Materials Project [8] and simulations were conducted. Additionally, the PyTorch-based universal machine learning interatomic potentials CHGNet [7] was used for comparison. All molecular dynamics simulations were carried out under the canonical ensemble simulations with a Nosé-Hoover thermostat [16, 17] at 300 K temperature. All the computations were performed with a single NVIDIA H100 GPU.

Fig. S6 Benchmarking on the computational efficiency involved in molecular dynamics simulations and geometry optimizations. All the calculations are performed on a single NVIDIA H100 GPU. The time consumption per step during 100 steps of molecular dynamics simulations for our model is presented. Results for CHGNet are shown in blue dots; however, the calculation for systems with more than 10 thousand atoms was halted due to out of GPU memory in the calculation environment.

As shown in **Fig. S6**, we present efficiency in terms of time consumption per simulation step. Although our model demonstrates $O(N^3)$ scaling due to inverting the **Eq. S3**, it successfully handles large-scale systems exceeding 20,000 atoms, highlighting its applicability to complex materials simulations. In our studied reaction systems containing tens of thousands of atoms, the inversion method provides acceptable computational efficiency. Nevertheless, in future work, we plan to further refine our algorithms, such as implementing iterative solution

methods to reduce complexity, thereby achieving more computationally efficient implementations and lowering resource consumption.

Q15. “The sentence ‘as well as account for the free energy surfaces under the conditions of finite-temperature thermodynamics’ does not make sense. The free energy is a state function and not a surface.”

Authors’ reply:

We thank the reviewer for this correction regarding our terminology. We realise that this sentence is scientifically imprecise. Free energy is indeed a state function, not a surface. In **Results** section, we have revised as,

...Accurately simulating phase changes in ferroelectric substances requires precise potential energy functions that can respond to the small atomic shifts and structural changes, as well as account for the *free energy landscape* under the conditions of finite-temperature thermodynamics [56] ...

The term “*free energy landscape*” is the standard terminology for describing how free energy varies across configuration space. We appreciate the reviewer’s attention to this detail, which helps improve the scientific rigour of our manuscript.

Q16. ““The QEq-based machine learning models inherently struggle with atomic polarization, leading to non-physical results under external electric fields.’ there is no evidence for that. In the results quoted by the authors on 4G-HDNNPs they may be less accurate but not unphysical.”

Authors’ reply:

We thank the reviewer for this important point regarding our characterization of QEq-based machine learning models. We have revised this sentence in the **Discussion** section,

... QEq-based models *inherently struggle with atomic polarization* because they *do not distinguish between core and shell components*. Due to their inability to handle *atomic-level responses* to external fields, they may exhibit non-physical behaviours in some cases as described in the reference [47].

Because the QEq model assumes the system is a *continuous conductor*, allowing charge transfer to occur at any distance, it produces problematic results for insulators or semiconductors, where long-range charge transfer should not occur in response to small differences in potential. As demonstrated in the *reference 47* [*J. Chem. Theory Comput.* **18**, 1, 580-594 (2022) <https://doi.org/10.1021/acs.jctc.1c00975>], the QEq model indeed predicts non-physical charge distributions. We tested our model under identical conditions and found that water molecules do not exhibit non-physical charge distributions.

In the **Results** section,

...As illustrated in **Fig. S5**, we conducted additional investigations of two water molecules under an external electric field of 0.25 V/\AA . In charge distribution models, it is essential to allow systems to form an internal electric field opposing the external field. The QEq model assumes the system behaves as a perfect conductor without penalizing charge transfer as a function of distance. This limitation of the QEq method leads to non-physical charge transfer between molecules separated by large distances [47]. In contrast, within PQEq model, the individual water molecules can polarize, *generating internal electric fields that counteract the external field*, thereby capturing more realistic electrostatic responses in molecular systems. Under QEq scheme, the water molecules accumulate non-zero net charges and migrate to the top and bottom of the simulation box. In contrast, our model maintains charge neutrality of the water molecules, which remain stationary despite the applied electric field, aligning with DFT results...

And in **SI**,

We also examine the limitations of the QEq method when applied to water molecules under an external electric field, particularly its tendency to predict physically unrealistic charge distributions. To illustrate the distinctions between QEq and PQEq charge distribution methods in the presence of an electric field, we performed a simple demonstration following the protocol outlined in reference [15]. **Fig. S5 a** presents visualizations of water molecules using both QEq and PQEq methods. **Fig. S5 b** and **c** depict the temporal evolution of corresponding z -positions and molecular charges within the simulation box for both methods at an electric field of 0.25 V/\AA . Under QEq, the water molecules accumulate non-zero net charges and migrate to the top and bottom of the box [15]. In contrast, using our model, the water molecules maintain charge neutrality and remain stationary despite the applied electric field. This behaviour aligns with DFT calculations, which similarly predict no molecular displacement for the two water molecules.

Fig. S5 a, Visualization of two interacting water molecules (M1 and M2) under an applied electric field of 0.25 V/Å oriented along the positive z -axis. The water molecules model oriented in the yz -plane, with oxygen and hydrogen atoms shown in red and white, respectively. **b**, Temporal evolution of z -positions difference ($|R_{M1,z} - R_{M2,z}|$) of the two water molecules and **c**, charge dynamics of both water molecules calculated using QEq method (yellow curves), our universal model (blue dots and curves), and DFT reference (green curves). Solid and dash lines represent two molecules, respectively. The QEq and DFT results are extracted from reference [15].

Q17. “outperforming conventional equivariant GNN-based MLIPs’ no evidence is showed. (Ref. 48. Is not an MLIP that fits this description).”

Authors’ reply:

We thank the reviewer for their important observation. Upon review, we realize that our statement was imprecise and potentially misleading. What we intended to emphasize was the difference in Na-Na/Na-Cl dimers resolution between our model and message-passing GNNs without long-range interactions that we trained ourselves, *not the model in the reference 48*. We have modified our statement to be more precise and accurately reflect our comparative analysis in the **Discussion** section,

Our pretrained universal model successfully reproduced DFT potential energy surfaces for *Na-Cl/Na-Na dimers across extended interatomic distances*, outperforming conventional message-passing MLIP beyond the cut-off regions.

Q18. “Remarks on code availability:

I haven’t had the chance to try it out, but it is available, and there are sufficient details to download/use.”

Authors’ reply:

We thank the reviewer for verifying the availability of our code and acknowledging that sufficient details are provided for downloading and usage. To further facilitate ease of use, we have open-sourced our entire model framework in a new repository.

Code availability

The JAX implementation code is available through <https://github.com/reaxnet/reaxnet>.

Response to reviewer #3

Q1. “### Summary

The manuscript presents a semi-heuristic approach to incorporating long-range interactions, such as polarizability, into machine learning potentials. The proposed framework integrates a physically motivated polarizable charge equilibration scheme within an equivariant graph neural network-based short-range potential. While the idea is clever, it is not entirely new—similar approaches have been explored before, such as in *SpookyNet*, although the current work extends these ideas further. The authors claim their model is universal and exhibits linear scaling in system size while accurately capturing long-range interactions.

While the proposed approach is an interesting idea and appears to perform well on specific test datasets, there are significant concerns regarding the validity of the claims made in the title and throughout the manuscript. The manuscript lacks rigorous benchmarking against existing universal models, and the claim of linear scaling contradicts fundamental physical / mathematical limitations. As such, I cannot recommend publication in its current form as outlined below in more details.”

Authors’ reply:

Thank you for your thorough review of our manuscript. We deeply respect your expert assessment and have carefully considered all your points. We have made substantial revisions to address the concerns you raised, particularly by adding comprehensive benchmarking against existing universal models as you suggested.

(1) Regarding SpookyNet, it directly incorporates the physically observable *dipole moment* into the loss function rather than fitting partial charges. However, its fundamental representation still relies on *point charges*, which significantly limits its ability to respond accurately to external fields.

We add discussions in the **Introduction** section,

... Another way to solve the partition-dependent issue is to *take dipole moment into the loss function*, as done by SpookyNet [37]. However, it still uses *point-like charges*, making it behaves similar to Qeq-based methods when dealing with *polarization effects*.

(2) For the *scaling assertion*, we are deeply sorry that we unintentionally made incorrect claim. We now provide a mathematical analysis that clarifies the actual cubic complexity (inverting the matrix), acknowledging the fundamental physical/mathematical limitations you pointed out.

(3) For the *universality*, we have included additional benchmarking against relevant universal models to substantiate our comparative performance claims. We believe these substantial revisions address the core concerns while maintaining the scientific value of our contribution. We appreciate your critical feedback which has significantly improved the accuracy and rigour of our manuscript.

Q2. “### Major Concerns

1. **Lack of Rigorous Benchmarking for a “Universal” Model**

The authors make a very strong claim that their model is universal, but the provided evidence does not support this assertion. There are two critical issues with this claim:

****Absence of Comparison with Established Models:**** The manuscript does not compare the model’s performance against other well-known universal machine learning potentials (e.g., MACE-MP-0, MATTERSim, SevenNet, ORB). Not only do the authors omit numerical comparisons, but these models are not even cited. Instead, the error analysis is restricted to their own training dataset, which is insufficient to validate universality.”

Authors’ reply:

We thank the reviewer for this important observation regarding comparisons with established models. We acknowledge this limitation in our current manuscript and agree that a more comprehensive comparison is necessary to substantiate our claims of universality. We have cited the universal models mentioned by the reviewer in the **Introduction** section,

... Moreover, universal models trained on the periodic table such as M3GNet [15], CHGNet [16], GNoME [17], *MACE-MP-0* [18], *MatterSim* [19], *SevenNet* [20], and *ORB* [21] have emerged, showing remarkable prospects for materials discovery.

We now conduct benchmarks to compare our model with these advanced models. It is important to note that we only compare *models trained on the MPtrj dataset* (CHGNet and MACE-MP-0). Although SevenNet was also trained on the MPtrj dataset, it did not split the data into training, validation, and test sets during training (see https://github.com/MDIL-SNU/SevenNet/tree/main/sevonn/pretrained_potentials/SevenNet_0_11Jul2024#sevennet-0-11july2024). We employ the challenging test sets proposed by MatterSim for benchmarking and found that our energy and stress errors are reasonable, and our *force error is better than both universal models on all sets*. Since forces are the most relevant metrics for molecular dynamics simulations, this demonstrates the robustness of our model.

In the **Results** section,

We also perform benchmarking of our model with other universal models on the test sets provided by the reference [19]. **Table S5** demonstrates the exceptional performance of our model across diverse test sets in *force prediction capabilities, consistently outperforming* both CHGNet [16] and MACE-MP-0 [18], while maintaining reasonable energy and stress predictions. We excluded MatterSim [19] results from our comparative analysis as its training dataset incorporates high-temperature data.

And in **SI**,

We also conducted benchmarking with other universal models on the test sets provided by the reference [11]. For near-equilibrium evaluations, two randomly sampled collections are used: 1,000 structures from the MPtrj dataset [7] (MPtrj-random-1k) and 1,000 from

Alexandria dataset [12] (Alexandria-1k). These assess model accuracy on energetically favourable configurations, which is crucial for predicting material stability. Three increasingly challenging thermodynamic perturbation (TP) test sets evaluate performance under varied temperature and pressure conditions with non-equilibrium atomic configurations ... **Table S5** demonstrates the performance of our model across diverse test sets, *particularly highlighting its superior force prediction capabilities*. Our model achieves remarkable force prediction accuracy, consistently outperforming both CHGNet [7] and MACE-MP-0 [14] (all three models were trained on MPtrj dataset) across all benchmark datasets with significantly lower MAEs – nearly half the error of competing models in many cases. Impressively, our model demonstrated robust generalization capability across increasingly challenging thermodynamic perturbation scenarios. Although we observe slightly higher energy prediction MAEs compared to the two models, its performance across the entire spectrum of materials conditions is improved particularly for force predictions critical for molecular dynamics simulations.

Table S5. Comparison of mean absolute errors (MAE) in the test set generated by the reference [11].

Test set	MAE	CHGNet	MACE-MP-0	Our model
MPtrj-random-1k	Energy (eV/atom)	0.027	0.015	0.019
	Force (eV/Å)	0.120	0.117	0.066
	Stress (GPa)	0.290	0.468	0.271
Alexandria-1k	Energy (eV/atom)	0.150	0.092	0.115
	Force (eV/Å)	0.108	0.095	0.064
	Stress (GPa)	1.643	0.160	1.055
MPF-Alkali-TP	Energy (eV/atom)	0.250	1.351	0.280
	Force (eV/Å)	1.636	15.819	0.717
	Stress (GPa)	12.625	25.723	13.841
MPF-TP	Energy (eV/atom)	0.254	256.340	0.521
	Force (eV/Å)	3.313	1506.854	0.768
	Stress (GPa)	25.208	202.093	15.445
Random-TP	Energy (eV/atom)	0.506	9.184	3.338
	Force (eV/Å)	3.950	88.327	1.199
	Stress (GPa)	7.230	19.224	33.282

We believe these additions significantly strengthen the manuscript and provide the necessary context for evaluating our model’s universality claims.

Q3. “2. ****Inadequate Testing of Universality:**** The authors evaluate their model on the common datasets published by Ko *et al.* But, the author performs their benchmarks bespoke models rather than testing their claimed universal model. If the model is indeed universal, it should be evaluated directly on these datasets without additional training. A fair comparison would involve evaluating both the universal and bespoke models to determine whether the universal model effectively learns long-range physics.”

Authors’ reply:

We appreciate the reviewer’s important point regarding the evaluation of our model’s universality. We would like to clarify that due to differences in basis sets, functionals, and computational software, we cannot directly validate a universal model trained with VASP/PBE+U functional and pseudopotential basis sets on the datasets proposed by Ko *et al.* The fundamental reason we adopted Ko *et al.*’s datasets was to verify the feasibility of our framework. Our universal model achieves a test error of 18 meV/atom on the MPtrj dataset, which is significantly larger than the ~ 1 meV/atom error that would be achieved by the model trained on the specific datasets proposed by Ko *et al.* To the best of our knowledge, so far, ***no universal model can reach ~ 1 meV/atom of precision.***

A fairer comparison might be to progressively ablate our models, including benchmarking models containing only GNN components against our framework model, to demonstrate that the introduction of long-range interactions further improves model accuracy. This approach allows us to isolate and quantify the specific contribution of the long-range interaction components within our framework.

In the **Results** section,

Benchmarking against different-charge-state dataset. To evaluate the capability of our framework in capturing different charge states and long-range interactions, we validated our framework against the dataset developed by Ko *et al.* [28] and Maruf *et al.* [38], which encompass various charge states and charge transfer systems ... We demonstrate the advantage of integrating equivariant message-passing networks with explicitly polarizable long-range interaction models by benchmarking against both 4G-HDNNP, a local MLIP with explicit long-range interactions, and NequIP, which implicitly captures long-range effects through its message-passing mechanism. 4G-HDNNP differentiates charged systems by explicitly training DFT partial charges and incorporating them as descriptors into neural networks for short-range interactions. We employed the NequIP models trained in the reference [38] (Maruf’s NequIP) as our baseline equivariant model. Our framework explicitly incorporates long-range interactions while adopts Maruf’s NequIP as the E_0 component without touching its settings, thereby ensuring a fair comparative analysis. The results are presented in **Table 1**, with detailed neural network architectural specifications provided in the **Table S3**.

Table 1. The comparison of root means square test error metrics on the different charge-state datasets.

Dataset		4G-HDNNP	Maruf’s NequIP	This work
$C_{10}H_2/C_{10}H_3^+$	Energy (meV/atom)	1.19	1.33	0.44
	Force (eV/Å)	0.078	0.071	0.023
$Na_{8/9}Cl_8^+$	Energy (meV/atom)	0.48	N. A.	0.23
	Force (eV/Å)	0.033	N. A.	0.006
$Ag_3^{+/-}$	Energy (meV/atom)	1.32	498.39	4.87
	Force (eV/Å)	0.032	2.145	0.028

Dataset		4G-HDNNP	Maruf's NequIP	This work
Au ₂ -MgO	Energy (meV/atom)	0.22	1.03	0.14
	Force (eV/Å)	0.066	0.034	0.021
BTA-Cu	Energy (meV/atom)	N. A.	0.48	0.30
	Force (eV/Å)	N. A.	0.008	0.004
BTA (H ₂ O)-Cu	Energy (meV/atom)	N. A.	0.71	0.19
	Force (eV/Å)	N. A.	0.010	0.005

... In the linear C₁₀H₂ molecule, ... after explicitly incorporating physical long-range interactions, our framework achieved additional improvements in error metrics compared to NequIP. For positively charged NaCl clusters, the predicted energy and forces by our model are still more accurate compared to 4G-HDNNP, as well as the potential energy surface shown in **Fig. S1 b**. For Au₂ adsorption on MgO surfaces and benzotriazole interactions with Cu-(111) surfaces in dry and aqueous environments, our model outperforms NequIP significantly. Additionally, our model correctly predicted the preferential adsorption geometries for both doped and undoped MgO, as listed in **Table S1**.

As for the charged Ag clusters, due to their small system size and lacking long-range effects, the entire structure falls within the typical cutoff radius used by the MLIP. The energy dependence on the overall charge state of clusters leads to degeneracy between atomic structures and potential energy surfaces, resulting in poor performance of charge-independent NequIP model. We acknowledge that for such small, highly charged systems, 4G-HDNNP approach to learning charge equilibration parameters is more convenient. Nonetheless, we also achieved reduced force error with the fixed physical parameters compared to 4G-HDNNP. We re-optimized the geometries of Ag₃^{+/-} using our model, with results shown in **Fig. S1 c**. Our model accurately predicts the structures, with root mean squared displacements of only 0.002 Å and 0.006 Å compared to DFT-optimized structures.

Our comprehensive evaluation demonstrates that explicit incorporation of physical long-range interactions significantly enhances the performance of MLIPs across diverse charge-state systems. Notably, our proposed framework outperforms 4G-HDNNP in most cases for energies and forces without fitting partial charges. This advantage likely stems from a more physically meaningful partition of the total energy.

After verifying that our model could perform well on the challenging dataset, we then focused on training a universal model. As mentioned above, we have supplemented our universal model with benchmarks to validate its generalizability across different datasets. These additional evaluations demonstrate that our approach not only excels on specific test cases but also maintains good performance when applied broadly across diverse chemical systems and properties.

Q4. “##### 2. **Misleading Scaling Claims**

One of the most problematic claims in the manuscript is that the model achieves linear scaling for long-range interactions:

- **Fundamental Computational Limits:** Long-range interactions require solving the Poisson equation, which fundamentally cannot be computed in linear time. The best possible scaling for such a computation is $O(N \log N)$ using Ewald summation on a grid, while traditional solvers scale as $O(N^{3/2})$ or worse.
- **Questionable Interpretation of Figure S4:** The claim of $O(N)$ scaling is based on Figure S4, but the presented data does not conclusively support this claim. If the authors indeed achieve linear scaling, they must be using a cutoff solver, which is known to introduce severe errors in energy and force computations.

Given these issues, the scaling behaviour of the proposed model needs significant clarification.”

Authors' reply:

We sincerely appreciate the reviewer's expert insights. We apologize for being misled by our original **Fig. S4**, where the large y -axis values and insufficient number of atoms incorrectly suggested linear scaling. Given that our approach involves matrix inversion, it should exhibit cubic scaling. We did not measure the scaling, instead, we unintentionally took the apparent but wrong trend. To fix this misleading presentation, we extend the system sizes to include up to 24,000 atoms being tested to reflect the cubic scaling behaviour.

We apologize again for this oversight. Nevertheless, for the reaction systems containing tens of thousands of atoms that we studied, the computational cost of cubic scaling remains acceptable, and the absolute computation time is still significantly better than CHGNet. In future work, we plan to update our algorithms by implementing iterative methods to solve the linear equation systems, which will achieve lower computational complexity.

In the **Results** section,

Computational cost. Despite the incorporation of additional long-range interaction calculations, our model maintains high computational efficiency. Performance benchmarks were conducted on a single NVIDIA H100 GPU to quantitatively assess the computational cost. As illustrated in **Fig. S6**, for a periodic system comprising 2160 atoms, our model requires approximately 0.07 s per molecular dynamics time step, representing a significant improvement over conventional universal MLIPs which require approximately 0.91 s. This computational advantage extends to larger systems, as demonstrated by simulations of a 24000-atom system, where our model maintains efficiency with only 1.49 s per time step. Such computational performance enables nanosecond-scale molecular dynamics simulations of systems containing tens of thousands of atoms, making it practical for more realistic modelling of materials.

And in **SI**,

Our model introduces explicit polarizable long-range interactions while still maintaining good computational efficiency. To highlight the performance of this model, benchmarking was

done on molecular dynamics simulations. Supercell models were generated from $\text{Li}_2\text{PO}_2\text{N}$ (mp-1020019) from Materials Project [8] and simulations were conducted. Additionally, the PyTorch-based universal machine learning interatomic potentials CHGNet [7] was used for comparison. All molecular dynamics simulations were carried out under the canonical ensemble simulations with a Nosé-Hoover thermostat [16, 17] at 300 K temperature. All the computations were performed with a single NVIDIA H100 GPU.

Fig. S6 Benchmarking on the computational efficiency involved in molecular dynamics simulations and geometry optimizations. All the calculations are performed on a single NVIDIA H100 GPU. The time consumption per step during 100 steps of molecular dynamics simulations for our model is presented. Results for CHGNet are shown in blue dots; however, the calculation for systems with more than 10 thousand atoms was halted due to out of GPU memory in the calculation environment.

As shown in **Fig. S6**, we present efficiency in terms of time consumption per simulation step. Although our model demonstrates $O(N^3)$ scaling due to inverting the **Eq. S3**, it successfully handles large-scale systems exceeding 20,000 atoms, highlighting its applicability to complex materials simulations. In our studied reaction systems containing tens of thousands of atoms, the inversion method provides acceptable computational efficiency. Nevertheless, in future work, we plan to further refine our algorithms, such as implementing iterative solution methods to reduce complexity, thereby achieving more computationally efficient implementations and lowering resource consumption.

Q5. “##### 3. **Concerns Regarding Specific Applications**

The manuscript presents results on well-known test cases, but the methodology used for comparison raises concerns:

- **Barium Titanate (BTO):** The authors compare phase transition temperatures in **Table 2**, but this comparison is misleading because the transition temperature is highly dependent on system size. The authors use a $10\times 10\times 10$ cell, whereas the reference MLIP from Gigli *et al.* models use a $4\times 4\times 4$ cell. The comparison should be made at equal system sizes or, at the very least, the system size used should be explicitly stated in the table with a comprehensive discussion.”

Authors’ reply:

We appreciate the reviewer’s critical observation regarding the comparison of phase transition temperatures for Barium Titanate in **Table 2**. The reviewer is correct that transition temperatures can be significantly affected by system size, making direct comparisons potentially misleading without proper context.

In response to this valuable feedback, we have revised **Table 2** to *explicitly include the system size* used for each model’s calculation, making the comparison transparent. Our $10\times 10\times 10$ supercell size is now clearly indicated alongside the $4\times 4\times 4$ cell used in the Gigli *et al.* reference.

In the **Results** section,

Table 2. The phase transition temperatures of BaTiO₃ obtained by different methods and experiments. The size of supercell is also displayed.

Method	$T_{c,R-O}$ (K)	$T_{c,O-T}$ (K)	$T_{c,T-C}$ (K)
This work ($10\times 10\times 10$)	145	205	285
Effective Hamiltonian ($12\times 12\times 12$) [62]	150 ± 10	195 ± 5	265 ± 5
Second Principles ($16\times 16\times 16$) [63]	140	180	224
Effective Hamiltonian ($16\times 16\times 16$) [58]	119	158	257
ReaxFF ($6\times 6\times 6$) [64]	N.A.	N.A.	240
MLIPs ($4\times 4\times 4$) [56]	18.6	91.4	182.4
Experiments [65]	183	278	403

We have attempted to simulate phase transitions using our model with a $4\times 4\times 4$ supercell to match the reference work by Gigli *et al.* However, we found that such small systems exhibit significant temperature fluctuations, making it challenging to accurately capture the free energy landscape and precise transition temperatures. To address this issue, we have added the following discussion to the manuscript.

In the **Results** section,

... To detect lattice distortions and variations in free energy that distinguish different phases, a $10\times 10\times 10$ supercell of BaTiO₃ was simulated. The use of larger simulation cells ($10\times 10\times 10$ in our work versus $4\times 4\times 4$ in Gigli *et al.* [56]) may provide advantages for phase transition studies. *Larger supercells can effectively reduce the impact of temperature fluctuations*, enabling better temperature control for sampling the free energy landscape, thereby yielding results with greater statistical significance.

We would like to clarify that our intention was *not to directly compare* our results with those of Gigli *et al.* We do not consider results from different supercell sizes to be directly comparable but rather *aim to demonstrate that our model can reasonably predict phase transition behaviour* in complex material systems.

In the **Results** section,

...It worth noting that the *supercell size can affect phase transition temperatures*, so that direct comparison between different simulations *is not expected*...

Q6. “- **Lithium Phosphorous Sulfide (LPS):** The claim that their model captures long-range interactions in LPS is questionable. In this material, interactions are largely screened, and no significant long-range charge transfer is expected to occur. This system has already been simulated successfully with non-long-range potentials (e.g., *Phys. Rev. Materials* **8**, 065403*), raising doubts about the necessity and impact of the proposed approach.”

Authors' reply:

We thank the reviewer for the valuable insights. There may be a misunderstanding, as we did not emphasize that our model captures long-range interactions in LPS, specifically in our application examples of Li₃PS₄-Li and Li₆PS₅Cl-Li electrolyte-electrode interface reactive molecular dynamics. Regarding the reference [*Phys. Rev. Materials* **8**, 065403 (2024) <https://doi.org/10.1103/PhysRevMaterials.8.065403>], it studied the *bulk properties* (thermal conductivity) of different LPS phases using MLIP, but not *interfacial reactions*.

Instead, we used these examples to compare with experimentally observed phenomena and some first-principles simulations merely to validate the robustness of our universal model – whether a universal model trained solely on the MPtrj dataset (*no interfacial data*), without incorporating specific reaction data, can correctly describe reaction molecular dynamics and even reproduce certain experimental phenomena.

Additionally, our model can clearly reflect changes in *partial charges* during the molecular dynamics process, as shown in **Fig. 5c**, facilitating our analysis of *SEI formation processes*. The distribution of Li partial charges gradually forms a stable plateau as the SEI forms.

In the **Results** section,

With the growth of the SEI layer, electrons were transferred from the Li anode to the Li₃PS₄ electrolyte, which causes a gradual increase of the partial charges of Li as they migrate from the anode, across the SEI, and into the electrolyte. This results in *a transition from metal lithium to ions*. The dynamic behaviour of partial charges at the anode-electrolyte interface is a critical aspect showcased in **Fig. 5c**. Initially, *the partial charges of lithium changed almost linearly* from the anode to the electrolyte. Over time, this distribution evolved and showed a distinct plateau. This phenomenon could be attributed to the ordered structuring of the SEI, signifying the nucleation of crystalline structures. Ultimately, *a stable lithium partial charges plateau* was formed between the anode and the electrolyte, spanning a range of 15 to 21.5 nm.

This is further supported by the visualized structure depicted in **Fig. 5 b**, where the plateau within the SEI aligns with the Li_2S crystal and amorphous regions.

Fig. 5 c, The distributions of the Li partial atomic charges along the x -direction at initial, 20, 100, 2000, and 4000 ps states.

There have been previous examples [*Energy Environ. Sci.* **17**, 2743-2752 (2024) <https://doi.org/10.1039/D3EE03536K>] using MLIP to study interfacial reactions, but these were based on *AIMD data of Li_3PS_4 -Li interface reactions*, whereas our model successfully reproduces experimentally observed results through training on MPtrj dataset. Furthermore, we have also used our universal model to study *$\text{Li}_6\text{PS}_5\text{Cl}$ -Li interfacial reactions*, reproducing the *polycrystalline structure* observed in experiments, as shown in **Fig. S7**. This represents an advantage over specialized MLIP models that would typically require *retraining to include the Cl element*.

Fig. S7 The formation of the $\text{Li}_6\text{PS}_5\text{Cl}/\text{Li}$ SEI layer at different temperatures. The elements are colour-coded for initial structures: Li in green, S in purple, P in pink, and Cl in light green. For final structures at 300 K, 400 K, and 500 K, the partial charges on lithium ions are represented with colour coding to enhance the visibility of structural transformations during the formation of SEI.

Q7. “- ****Lack of Fine-Tuning Capabilities:**** Universal models are only widely useful if they can be fine-tuned to specific systems. This is common practice in other domains (e.g., stable diffusion for images or MATTERSIm, MACE-MP0 for materials), yet the authors provide no

evidence that their model supports such fine-tuning. Examination of the code suggests that this feature is not even planned.”

Authors’ reply:

We thank the reviewer for raising this important point about *fine-tuning* capabilities in universal models. We agree that the ability to fine-tune a universal model for specific systems is valuable for enhancing performance in targeted applications. We have included a demonstration example in the supplementary materials showing how our model can be fine-tuned for high-temperature dataset, with quantitative improvements in property prediction accuracy.

In SI,

... To evaluate our model’s capacity to learn high-temperature data representations, we fine-tuned our pretrained universal model on the MPF-TP dataset. During this fine-tuning process, only the final two convolutional layers and the multilayer perceptron layers were set as trainable parameters, while all remaining layers were frozen.

Our fine-tuning results demonstrate remarkable improvements on high-temperature, non-equilibrium systems as shown in **Table S6**. The fine-tuned model *achieves state-of-the-art accuracy* across all prediction targets on the MPF-TP dataset. This suggests that while our universal model can maintain reasonable force accuracy under a wide range of conditions, targeted fine-tuning is more likely to balance broad applicability with the superior accuracy required for specific applications.

Table S6. Comparison of mean absolute errors in the MPF-TP test set of the universal models and finetuned model.

	Energy (eV/atom)	Force (eV/Å)	Stress (GPa)
CHGNet	0.254	3.313	25.208
MACE-MP-0	256.340	1506.854	202.093
MatterSim (M3GNet)	0.036	0.431	1.318
MatterSim (Graphormer)	0.040	0.421	1.917
Our universal model	0.521	0.768	15.445
Fine-tuned model	0.027	0.203	0.976

We also included a demonstration *example in our code repository* showing how our model can be fine-tuned for specific material systems.

We believe this addition significantly enhances the practical utility of our universal model, and we appreciate the reviewer highlighting this important aspect of model development.

Q8. “### Minor Issues and Suggested Revisions

1. **Dataset Clarifications:** Clearly state that MPtrj is the target training dataset in the main text (Page 6, Line 169).”

Authors’ reply:

We thank the reviewer for this suggestion. We have revised it to clearly state that MPtrj is our target training dataset.

In the **Results** section,

... We trained a universal model for all the periodic table elements up to Pu *using MPtrj dataset* [16] following our framework (our model), as described in the **Methods** section ...

And in the **Methods** section,

To train the universal equivariance neural network potential, we used the MPtrj dataset [16] sourced from Materials Project [48] *as the training dataset* ...

Q9. “2. **Consistency in Units:** Ensure uniform usage of units, likely $V/\text{\AA}$, in all relevant sections (e.g., Page 7, Line 221).”

Authors’ reply:

We thank the reviewer for highlighting this important issue regarding consistency in units. We have carefully reviewed the manuscript and standardized all electric field units to $V/\text{\AA}$.

In the **Results** section,

... As illustrated in **Fig. S5**, we conducted additional investigations of two water molecules under an external electric field of $0.25 V/\text{\AA}$.

In **SI**,

... **Fig. S5 b** and **c** depict the temporal evolution of corresponding z -positions and molecular charges within the simulation box for both methods at an electric field of $0.25 V/\text{\AA}$.

Regarding the units of PQEq parameters, to maintain consistency with original PQEq paper, we have retained the conventional units where χ and η are expressed in eV, while the spring constant is expressed in kcal/mol/ \AA^2 . All other energy units throughout the work have been standardized to eV (meV) for consistency. For forces and stresses units, $eV/\text{\AA}$ and GPa are used, respectively.

Q10. “3. **Charge Values for Water Molecule:** Provide explicit charge values from Figure 2d for easier comparison, similar to what is done in **Figure S1a**.”

Authors' reply:

We thank the reviewer for this helpful suggestion. We have added supplementary materials to illustrate the charge variations. We believe that due to differences in charge partitioning methods, comparing absolute charge values may not be meaningful; however, ***the trends in charge variations*** can effectively reflect our model's response to external electric fields. The supplementary information now includes a detailed analysis of how partial charges evolve under different electric field strengths.

In SI,

.... To illustrate this difference, we present in **Fig. S4** the variations in O atom partial charge differences ($q - q_{\min}$) across different models as a function of the applied electric field strengths ...

Fig. S4 The variations in O atom partial charge differences ($q - q_{\min}$) between DFT Hirshfeld calculations, our model (PQEq model), and the conventional QEq model.

Q11. “4. **Thermostat Choice**: The authors use a Nosé-Hoover thermostat in their molecular dynamics simulations, which does not properly sample the canonical ensemble for small systems and can artificially suppress fluctuations. A stochastic thermostat (e.g., Langevin or Bussi-Donadio-Parrinello) would be more appropriate to ensure correct statistical sampling.”

Authors' reply:

We thank the reviewer for this insightful comment regarding thermostat choice in our molecular dynamics simulations. We agree that the choice of thermostat can significantly impact the quality of statistical sampling, especially for small systems. However, we would like to clarify that the smallest system in this manuscript underwent MD simulations contains 1,536 atoms. For systems of this size, we have conducted ***additional comparative simulations*** that demonstrate minimal differences in results between the Nosé-Hoover thermostat and stochastic thermostats such as Langevin. Our supplementary tests confirm this equivalence, as shown in the **Fig. R2** and **Table R2**.

Fig. R2 Evolution in temperature, potential energy, and pressure during molecular dynamics simulations of 1536-atom LLZO using **a**, Nosé-Hoover, and **b**, Langevin thermostats at 1000K.

Table R4 Confidence intervals of temperature, potential energy, and pressure during molecular dynamics simulations of 1536-atom LLZO with Nosé-Hoover and Langevin thermostats.

Thermostat	Temperature (K)	Potential energy (eV)	Pressure (GPa)
Nosé-Hoover	1000.29 ± 21.40	-11627.45 ± 3.97	-0.01 ± 0.20
Langevin	997.51 ± 20.60	-11628.02 ± 3.94	0.01 ± 0.20

As **Fig. R2** and **Table R4** demonstrate, the differences in temperature, potential energy, and pressure between the two thermostats *are negligible and well within statistical uncertainty* for our systems. While we acknowledge that stochastic thermostats offer theoretical advantages for canonical ensemble sampling in smaller systems, our specific applications involve sufficiently large simulation cells where the Nosé-Hoover thermostat performs adequately for the properties of interest.

Q12. “Remarks on code availability:

The provided repository includes a README file with installation instructions, and the code appears to be installable. However, the core model presented in the manuscript is not included in the repository, preventing direct verification of the results.”

Authors’ reply:

We thank the reviewer for their feedback regarding code availability. We have open-sourced *the complete model codebase* in the new repository and supplemented it with additional tutorials and fine-tuning examples to facilitate implementation for users.

Code availability

The JAX implementation code is available through <https://github.com/reaxnet/reaxnet>.

Response to the reviewers

We sincerely thank the three reviewers for their continued time and effort in this second round of review. Their additional feedback has been invaluable, and we have revised the manuscript to address all remaining points. We have provided point-by-point responses to the questions raised by the reviewers in this document. The modifications to the manuscript are highlighted in **dark red**.

Table of Contents

Response to reviewer #1	2
Response to reviewer #2	3
Response to reviewer #3	10

Response to reviewer #1

“The authors have addressed most of my concerns. One remaining point remains.

Q6. “Figure 2 b-c: the authors should increase the x-axis range to 12 Å to actually see the long-range scaling behaviour.”

I do not see “other publications also do it like this” as a valid argument. Understandably, the surrogate model cannot be more accurate than the data-generating DFT model. So, I’m not sure why the authors are trying to hide this. Thus, the authors should add the results. If not in the main paper then at least in the supplementary information.”

Authors’ reply:

We sincerely thank the reviewer for acknowledging our previous revisions. We also appreciate your valuable suggestion, which has prompted us to further clarify an important scientific issue. We agree with your point that simply citing ‘standard practice’ of other publications is an insufficient justification, and we apologize for the lack of clarity in our previous response. Your suggestion to extend the x-axis to 12 Å has indeed provided a much better perspective for a comprehensive evaluation of our model’s performance.

In the **main text**, we *added the entire distance up to 12 Å* for comparison,

“**Fig. 2 a**, The interaction energies for Na-Na (green) and Na-Cl (purple) dimers predicted by our model, w/o-Ir model, and reference calculations using DFT.

...We analysed interaction energies for Na-Na and Na-Cl dimers across interatomic distances ranging from 1.8 Å to 12.0 Å...”

We believe these additions fully address the reviewer’s point and have made the manuscript stronger. We thank the reviewer again for pushing us to include this important validation.

Response to reviewer #2

“Thanks to the authors for their reply and reviewing the manuscript. Factual errors have been mostly corrected, but there remains some open questions. These should be clarified and/or corrected before considering this manuscript any further. Here are some comments:”

Authors’ reply:

We thank the reviewer for their feedback and for taking the time to review our revised manuscript. We appreciate the acknowledgment that some factual errors have been corrected. We have carefully considered the remaining questions and will address each point thoroughly in the updated manuscript to provide the necessary clarifications and corrections.

“**Table S5** does not represent any significant improvements over other methods. A selling point could be increased transferability etc. but no evidence for that.”

Authors’ reply:

We thank the reviewer for this constructive and valuable feedback. We agree that the original manuscript did not sufficiently demonstrate the transferability of our method, which is indeed a key distinctive feature. Following the reviewer’s suggestion, we have performed additional analyses and included substantial new evidence to explicitly showcase the superior transferability of our approach. Specifically,

1. **Transferability of long-range physics.** First, we demonstrate our model’s ability to accurately describe dimer interactions across multiple distance scales. As shown in the new **Fig. S7**, our model correctly reproduces the potential energy surfaces for both Na-Na and Na-Cl dimers. In addition, our model outperforms CHGNet and MACE-MP-0 on both short-range and long-range distances. Second, we investigated the OH-OH system in both its neutral (weakly interacting) and ionic (charge-repulsive) states. We showcase our model’s unique ability to distinguish between different charge states and correctly model their corresponding long-range physics—a fundamental challenge for other universal models.
2. **Transferability of polarization.** To further provide evidence of our model’s transferability in terms of polarization, we tested its ability to predict responses to external electric fields—a crucial test of physical fidelity that reveals limitations in other universal models. First, in a periodic system of 64 water molecules, our model accurately reproduced the quadratic energy dependence under a varying electric field, showing outstanding agreement with DFT reference calculations (**Fig. S9**). Notably, its performance matched that of FIREANN, a state-of-the-art model specifically trained on that system, whereas leading universal models like CHGNet and MACE-MP-0 were completely unresponsive to the field. In addition, we validated our model’s ability to predict molecular polarizability for 7,211 diverse molecules from the QM-7b dataset, a task beyond the capabilities of other universal models. Our model achieved a low mean absolute error of 4.57 a.u. against high-level theory reference values (**Fig. S10**).

These additions provide the quantitative evidence for transferability of our foundation model. We are confident that these results now clearly establish a significant improvement over other methods, not just in the metrics of **Table S5**, but in the crucial dimension of physical transferability.

In **SI**, we detail the transferability,

“In comparison with existing universal models, our model demonstrates transferability in describing dimer interactions. As mentioned in the main text, regarding the potential energy surface capturing of Na-Cl and Na-Na dimers, we have discovered that our model not only accurately reproduces the long-range behaviour predicted by DFT/PBE [18] calculations, but also effectively captures the potential energy surfaces of these dimers in the short-range region compared to CHGNet and MACE-MP-0, as illustrated in **Fig. S7**. The enhanced transferability of our model across different dimers suggests its potential utility for a broad spectrum of applications requiring accurate representation of interatomic forces across multiple distance scales.

Another example that showcases the transferability of our model is its ability to distinguish between different charge states and accurately simulate their corresponding long-range interactions. We selected the OH-OH system as a case study, investigating the potential energy curves when the system consists of two neutral hydroxyl radicals versus when it carries a single negative charge. As illustrated in **Fig. S8**, the plot compares the performance of our model against DFT (the ground truth), MACE-MP-0, and CHGNet for the system in both “ionic” and “neutral” states as a function of distance.

Fig. S7 The interaction energies for Na-Na (green) and Na-Cl (purple) dimers predicted by our model, CHGNet [7], MACE-MP-0 (large version) [11], and reference DFT calculations.

For the neutral state (dashed lines), which represents the physical scenario of a weak, non-bonded interaction, the DFT reference shows a very shallow potential well with relative energies. Our model demonstrates good agreement with the DFT reference, accurately capturing the subtle attractive interaction. In contrast, MACE-MP-0 and CHGNet struggle to capture this long-range interaction; they appear less transferable than our model as they fail to reproduce the precise shape and depth of the DFT curve. For the ionic state (solid lines), which represents a strong, long-range Coulombic repulsion between charged species, the DFT reference shows a significant positive (repulsive) energy that decays slowly with distance. Our

model correctly identifies the ionic state and qualitatively reproduces the long-range repulsive energy curve from DFT. Despite a slight overestimation of the energy, the model captures the physical trend and magnitude of the interaction across the entire distance range. Conversely, MACE-MP-0 and CHGNet are completely unable to distinguish the ionic from the neutral state. This represents a fundamental physical problem, indicating that their underlying architectures cannot capture charge-state-dependent long-range physics.

Fig. S8 Relative interaction energy versus distance for an OH-OH system in its neutral (dashed lines) and ionic (solid lines) states. Our model’s predictions are compared with DFT, MACE-MP-0, and CHGNet. The results demonstrate that our model can reproduce the DFT reference for both the weak interaction of the neutral state and, crucially, the long-range Coulombic repulsion of the ionic state. Universal models like MACE-MP-0 and CHGNet fail to capture the long-range physics of the ionic state.

A crucial test of a model’s physical fidelity is its response to external fields. We evaluated our model’s ability to predict the energy response of a system to a uniform electric field. We adopted a periodic system containing 64 water molecules with each cell length measured 12.4185 Å, as described in reference [19]. All the water molecules are within yz -plane. The first and third layers are same while the second and fourth are same. An electric field was then applied along the x -axis with varying strength.

Fig. S9 Evaluation of polarizable interactions in the periodic water model. The plot compares the performance of our model against DFT [19] (reference), FIREANN [19], CHGNet, and MACE-MP-0.

The results are presented in the **Fig. S9**, which plots the system energy as a function of field strength. Our model demonstrates good agreement with the DFT reference, accurately reproducing the energy curve across the entire field range from -2 to 2 V/Å. Notably, the performance of our foundation model is on par with that of FIREANN [19], a state-of-the-art model that was specifically trained on this water system. This highlights the transferability of our model, as it correctly captures complex polarization physics without explicit training for this task. In contrast, the CHGNet and MACE-MP-0 models are entirely unresponsive to the field, predicting a constant energy. This represents a fundamental failure to capture one of the most essential physical interactions, underscoring the superior predictive power and physical realism of our model.

Fig. S10 Comparison of the predicted isotropic polarizability (α_{iso}) against reference coupled-cluster theory including single and double excitations values for 7,211 molecules from the QM-7b dataset [20]. The model achieves a mean absolute error (MAE) of 4.57 a.u., demonstrating its strong capability to generalize and predict complex electronic response properties accurately.

Conventional universal machine learning potentials like MACE-MP-0 and CHGNet are typically trained on field-free energies and forces, and their architectures lack the components to describe fundamental electrostatic properties like the dipole moment. Consequently, they are inherently incapable of predicting response properties such as polarizability, which is defined by the change in energy or dipole moment under an applied field. We further validated the powerful transferability of our model by assessing its ability to predict molecular polarizability, a key electronic response property. To accomplish this, we curated a challenging test set comprising 7,211 diverse small molecules from the QM-7b dataset [20-23] ... This involved calculating the system's dipole response to a small, externally applied electric field—a physical process the model was not explicitly trained to reproduce. The results of this evaluation are

presented in the **Fig. S10**. The plot, which compares our model’s predictions against the reference values, demonstrates good agreement. The model achieves a MAE of just 4.57 a.u. across this vast and chemically diverse dataset. This high level of accuracy, achieved without direct training on polarizability data, provides compelling evidence for the model’s strong generalization capabilities and its ability to capture the fundamental physics governing molecular electronic response, a capability that lies beyond the scope of other universal machine learning potential models.”

“**Table S6**. I have no stake in MACE-MP-0 but I can immediately tell that those results can’t be right and I am sure the MACE-MP-0 authors will challenge those as soon as they see this table. It seems extremely unlikely that in all the other metrics MACE-MP-0 does well and suddenly the errors become huge.”

Authors’ reply:

We thank the reviewer for their careful scrutiny of the MACE-MP-0 results and for raising this important point. We understand the concern, as the reported errors are indeed strikingly large. The large errors reported are indeed correct and stem from the specific nature of the benchmark, which is designed to evaluate models under challenging thermodynamic perturbation datasets. The large version of MACE-MP-0 trained on the MPtrj dataset consistently demonstrates limitations when evaluated on such extreme and challenging test cases.

Furthermore, to ensure a rigorous and fair comparison, we conducted a thorough review of the evaluation metrics. We discovered that the definition for MAE of forces and stresses used in the MatterSim paper (from which we cited their results) differs from the definition used during their training. We have therefore re-evaluated all models using a single, unified definition to ensure consistency.

To guarantee full transparency and reproducibility, we have made our complete evaluation script for MACE-MP-0 available on Figshare at the following link: <https://figshare.com/s/7ecc56e3538d83c83010>. We believe this clarification and the provided script will fully address the reviewer’s concerns about the validity of our findings.

“About the forces/stresses. Since the forces depend on the charges, which depend on the positions, one must take the chain rule and calculate the appropriate term. **Table R2** is not convincing at all - this compares to another implementation presumably using the same formulae. Did the authors compare their forces to forces obtained using finite differences?”

Authors’ reply:

We sincerely thank the reviewer for their insight on this critical point. After a careful re-derivation, we identified that it has a Hellmann-Feynman-like term which will vanish as you mentioned before. To provide a definitive and convincing validation, we have also followed your suggestion and compared our auto-gradient forces against the forces obtained using finite

differences for a water molecule as described in **Table R1**. As depicted in **Table R2**, the auto-gradient forces and finite differences results show excellent agreement, confirming the correctness of our implementation. We are grateful for the reviewer’s suggestion, which has significantly improved the rigor of our work.

The complete and detailed derivation has been added to the **SI**,

“The total forces of our model are,

$$\mathbf{F}_i = -\frac{\partial E_{\text{pot}}}{\partial \mathbf{r}_i} - \sum_j \frac{\partial E_{\text{pot}}}{\partial q_j} \frac{\partial q_j}{\partial \mathbf{r}_i}, \quad (\text{S9})$$

where E_{pot} , \mathbf{r}_i , and q_i are total potential energy as described in **Eq. 1**, atomic coordinates, and partial charges, respectively. The Lagrange multiplier method ($E_\lambda = E_{\text{pot}} - \lambda \sum q_i$) requires

$$-\frac{\partial E_\lambda}{\partial q_j} = 0, \quad (\text{S10})$$

so that,

$$\frac{\partial E_{\text{pot}}}{\partial q_j} = \lambda. \quad (\text{S11})$$

Thus, the forces are simplified as,

$$\mathbf{F}_i = -\frac{\partial E_{\text{pot}}}{\partial \mathbf{r}_i} - \lambda \sum_j \frac{\partial q_j}{\partial \mathbf{r}_i}. \quad (\text{S12})$$

And with the constraint,

$$\sum q_j = Q, \quad (\text{S13})$$

we have $\frac{\partial Q}{\partial \mathbf{r}_i} = 0$ since the net charge Q is a constant. Finally, the atomic forces are,

$$\mathbf{F}_i = -\frac{\partial E_{\text{pot}}}{\partial \mathbf{r}_i}. \quad (\text{S14})$$

...”

Table R1 Water molecule coordinates.

	x (Å)	y (Å)	z (Å)
O	-1.464	0.099	0.300
H	-1.956	0.624	-0.340
H	-1.797	-0.799	0.206

Table R2 Comparison of the auto-gradient and finite differences forces on the water molecule.

	Forces (eV/Å)					
	$F_x^{\text{Numerical}}$	F_x^{Auto}	$F_y^{\text{Numerical}}$	F_y^{Auto}	$F_z^{\text{Numerical}}$	F_z^{Auto}
O	-0.13566780	-0.13566779	-0.06141234	-0.06141233	-0.12068319	-0.12068319
H	0.06315283	0.06315281	0.07240831	0.07240830	0.04431891	0.04431891
H	0.07251500	0.07251498	-0.01099598	-0.01099597	0.07636429	0.07636429

“Regarding **Fig. 2b.** and **R3** yes, it is clear now that due to the specific direction the electric field was applied this is indeed the case. However, I disagree with the general statement “This is a fundamental limitation of the traditional QEq-based model—it lacks the capability to respond to external electric fields when atomic positions remain fixed.” If the electric field had components in the y or z directions, the energy would change.”

Authors’ reply:

We thank the reviewer for this insightful comment and for pointing out the imprecision in our original statement. The reviewer is absolutely correct. A traditional QEq-based model would indeed show an energy change if the electric field applied has a component parallel to the direction of molecular dipole moment. We are grateful to the reviewer for pointing this out. We have revised it as in the **main text**,

“...Conventional QEq-based models have limitations in their response to an external electric field applied orthogonally to the molecular dipole...”

““The term “free energy landscape” is the standard terminology for describing how free energy varies across configuration space.” this is not quite right. This term describes the free energy as a function of a collective variable or set of collective variables. But the manuscript text is OK.”

Authors’ reply:

We thank the reviewer for the valuable clarification on the term “free energy landscape” and for the approval of our manuscript’s content. Reviewer’s rigorous approach and expertise in mathematical physics are, once again, truly impressive.

Response to reviewer #3

“The authors have addressed many of the concerns raised in my previous review, and the manuscript has improved in several respects. For example, the addition of **Table 1** is a valuable contribution. However, I still find that the very strong claims made in the title and abstract—particularly the notion of a universal long-range model—are not sufficiently supported by the presented evidence.”

Authors’ reply:

We sincerely thank the reviewer for acknowledging the improvements we made in the manuscript, particularly for recognizing the value of the newly added **Table 1**. Our model can demonstrate the capability to capture accurate long-range physics in certain scenarios. For instance, it accurately models interactions across various distance scales, reproducing the potential energy surfaces for Na-Na and Na-Cl dimers (new **Fig. S7**) while outperforming benchmarks like CHGNet and MACE-MP-0. Crucially, our model also addresses a key limitation of other universal potentials by successfully distinguishing between different charge states. This is highlighted by its ability to correctly model the distinct long-range physics of the OH-OH system in both its neutral (weakly interacting) and ionic (charge-repulsive) forms.

However, we recognize that for more challenging and diverse systems, such as the dataset proposed by Ko *et al.*, our model serves as a foundation model that requires further *finetuning* to achieve high accuracy. We will elaborate on this specific point in our response to the reviewer’s next question. We now position our work as a “**foundation model**”. This term more accurately reflects our contribution: a robust, physically-informed starting point that is designed to be efficiently finetuned to achieve *ab initio* accuracy for specific, challenging applications. This two-step approach—pretraining and finetuning—represents, in our view, a more pragmatic and powerful paradigm for modern materials modelling. The revised title is now,

“A *foundation* machine learning potential with polarizable long-range interactions for materials modelling”

And in the abstract, we further address the foundation model and finetuning,

“...*The foundation model*, trained across the periodic table up to Pu, demonstrates strong performance across key materials modelling challenges... Furthermore, we show that as a foundation model, it can be efficiently *finetuned to achieve high-level accuracy for specific challenging systems.*”

Detailed physical test of long-range physics in **SI**,

“In comparison with existing universal models, our model demonstrates transferability in describing dimer interactions. As mentioned in the main text, regarding the potential energy surface capturing of Na-Cl and Na-Na dimers, we have discovered that our model not only accurately reproduces the long-range behaviour predicted by DFT/PBE [18] calculations, but also effectively captures the potential energy surfaces of these dimers in the short-range region compared to CHGNet and MACE-MP-0, as illustrated in **Fig. S7**. The enhanced transferability of our model across different dimers suggests its potential utility for a broad spectrum of

applications requiring accurate representation of interatomic forces across multiple distance scales.

Fig. S7 The interaction energies for Na-Na (green) and Na-Cl (purple) dimers predicted by our model, CHGNet [7], MACE-MP-0 (large version) [11], and reference DFT calculations.

Another example that showcases the transferability of our model is its ability to distinguish between different charge states and accurately simulate their corresponding long-range interactions. We selected the OH-OH system as a case study, investigating the potential energy curves when the system consists of two neutral hydroxyl radicals versus when it carries a single negative charge. As illustrated in **Fig. S8**, the plot compares the performance of our model against DFT (the ground truth), MACE-MP-0, and CHGNet for the system in both “ionic” and “neutral” states as a function of distance.

Fig. S8 Relative interaction energy versus distance (Å) for an OH-OH system in its neutral (dashed lines) and ionic (solid lines) states. Our model’s predictions are compared with DFT, MACE-MP-0, and CHGNet. The results demonstrate that our model can reproduce the DFT reference for both the weak interaction of the neutral state and, crucially, the long-range Coulombic repulsion of the ionic state. Universal models like MACE-MP-0 and CHGNet fail to capture the long-range physics of the ionic state.

For the neutral state (dashed lines), which represents the physical scenario of a weak, non-bonded interaction, the DFT reference shows a very shallow potential well with relative energies. Our model demonstrates good agreement with the DFT reference, accurately capturing the subtle attractive interaction. In contrast, MACE-MP-0 and CHGNet struggle to capture this long-range interaction; they appear less transferable than our model as they fail to reproduce the precise shape and depth of the DFT curve. For the ionic state (solid lines), which represents a strong, long-range Coulombic repulsion between charged species, the DFT reference shows a significant positive (repulsive) energy that decays slowly with distance. Our model correctly identifies the ionic state and qualitatively reproduces the long-range repulsive energy curve from DFT. Despite a slight and consistent overestimation of the energy, the model captures the physical trend and magnitude of the interaction across the entire distance range. Conversely, MACE-MP-0 and CHGNet are completely unable to distinguish the ionic from the neutral state. This represents a fundamental physical problem, indicating that their underlying architectures cannot capture charge-state-dependent long-range physics.”

“A central issue remains: while the model is trained to be universal, the authors do not convincingly demonstrate that it has actually learned the correct long-range physics, as opposed to simply benefitting from increased expressivity due to the added long-range block. This point was raised in my initial review, but the response did not adequately address it.

The authors argue that:

> “we cannot directly validate a universal model trained with VASP/PBE+U functional and pseudopotential basis sets on the datasets proposed by Ko et al.”

While technically true, this is not a convincing justification for avoiding such validation. The Ko et al. datasets contain only a few hundred small structures, and recomputing them using the authors’ own DFT setup (VASP/PBE+U) is computationally feasible—especially in the context of training a large-scale universal model. Recomputing small datasets for consistent evaluation is standard practice in the field.

The further claim that: > “so far, no universal model can reach ~ 1 meV/atom of precision.” misses the point. The Ko et al. datasets include physically motivated test cases, such as displacement scans of NaCl and Au₂ systems, which reveal whether a model correctly captures energy and force trends with interatomic distance. The authors should at least reproduce such physical validation tests with both their universal and bespoke models to support their claims.”

Authors’ reply:

We sincerely thank the reviewer for this critical and insightful feedback. We agree that our previous response did not adequately address the central issue: whether our model has learned the correct long-range physics or is merely benefiting from increased architectural expressivity. We agree that performing validation on the potential energy surfaces of NaCl clusters and Au₂ systems is indispensable.

As the reviewer previously pointed out, a pretrained foundation model should demonstrate its capacity for learning challenging physics through finetuning. We first evaluated various models on the potential energy surfaces (PES) of the Na₈Cl₈⁺ and Na₉Cl₈⁺ charged clusters from the Ko *et al.* dataset. As illustrated in **Fig. S11 (a, c, and e)**, all three foundation models

(ours, CHGNet, and MACE-MP-0) fail to reproduce the distinct DFT potential energy surfaces for these two systems. This confirms the reviewer’s scepticism about the out-of-the-box performance of universal models on such specific, challenging tasks. However, even at this stage, our model showed a nascent ability to distinguish between the two clusters, whereas the others did not. We then finetuned all three models using a small subset (20%) of the configurations. The same algorithm was used for all models, and only the mean squared force loss was used to update the weights. After finetuning, PES of our finetuned model for both Na_8Cl_8^+ and Na_9Cl_8^+ achieve good agreements with the DFT reference (**Fig. S11 b**). In contrast, while finetuning reduced the overall error for CHGNet and MACE-MP-0, their final predicted energy curves settled into an intermediate position, averaging the two distinct DFT reference curves (**Fig. S11 d and f**). They remain unable to differentiate between the two distinct clusters. We may attribute this limitation to the absence of explicit descriptors for the system’s net charge within their architectures, which prevents them from tweaking the unique energetic signature of each cluster. We then extended our investigation to the complex periodic system of an Au_2 dimer adsorbed on both Al-doped and undoped MgO surfaces. Following a similar finetuning protocol, we found that message-passing-based models are indeed capable of learning these precise potential energy surfaces. As summarized in **Table S6**, both finetuned CHGNet and our model successfully reproduce the DFT equilibrium bond lengths on both surfaces, with our model achieving slightly higher fidelity. The inferior performance of MACE-MP-0 in this case may be attributed to its limited effective receptive field (arising from only two message-passing layers).

Our new results not only validate this approach but also reveal the critical importance of architectural design—specifically, the inclusion of net charge as a descriptor—in successfully capturing complex electrostatic interactions.

In **SI**,

“While our model provides a foundational understanding, achieving *ab initio* accuracy for specific and challenging systems like charged clusters necessitates a refinement of the potential energy surface. To rigorously assess the physical fidelity of the foundation models, we conducted an evaluation on the potential energy surfaces (PES) of charged sodium chloride clusters from the dataset proposed by Ko *et al.* [4]. The initial test involved scanning the PES of the $\text{Na}_{8/9}\text{Cl}_8^+$ cluster systems along a coordinate defined by varying the distance between two sodium atoms. As illustrated in **Fig. S11 (a, c, and e)**, a comparison with DFT calculations reveals that all three foundation models fail to accurately describe the PES for these systems. Notably, our model demonstrated a potential to distinguish between the two clusters due to the charge state descriptor, whereas other models did not.

We acknowledge that capturing such a refined PES requires model specialization, so we performed a finetuning procedure on all three models. A subset was created by randomly selecting 20% of the total 5000 configurations from the $\text{Na}_{8/9}\text{Cl}_8^+$ dataset. For our model and CHGNet, a parameter-efficient approach was used where only the final two convolutional layers, and the multilayer perceptron were updated. For MACE-MP-0, the entire model was subjected to finetuning, as implemented in its own code. The finetuning protocol was as follows: the subset was partitioned into training and validation sets with a 4:1 ratio. The optimization objective was exclusively the atomic forces, using the Mean Squared Error (MSE) as the loss

function. All models were trained for 1000 epochs with a batch size of 10 and a learning rate of 5×10^{-4} .

After finetuning, PES of our model for $\text{Na}_{8/9}\text{Cl}_8^+$ clusters achieve a good agreement with the DFT reference as shown in **Fig. S11 b**. This demonstrates that by accurately learning the forces, the model implicitly reconstructs a highly precise and physically consistent potential energy surface, capturing the subtle energetic details that were absent in the foundation model. On the other hand, following the finetuning procedure, both CHGNet and MACE-MP-0 exhibited a reduction in error as shown in **Fig. S11 d** and **f**. However, their final predicted potential energy surfaces settled into an intermediate position, averaging the two distinct DFT reference surfaces for two clusters. This indicates that while finetuning enables these foundation models to learn a more refined potential energy surface to some extent, they remain unable to differentiate between the two distinct clusters. We may attribute this limitation to the absence of explicit descriptors for the system's net charge within their architectures, which prevents them from tweaking the unique energetic signature of each cluster.

Fig. S11 Potential energy surfaces of the $\text{Na}_{8/9}\text{Cl}_8^+$ cluster system. The surfaces were generated by varying the position of the first Na atom relative to the second Na atom as described in **Fig. S1**. Left column: foundation models' results of **a**, our model, **c**, CHGNet, and **e**, MACE-MP-0 (large version). Right column: the finetuned models' results of **b**, our model, **d**, CHGNet, and **f**, MACE-MP-0. Our model achieves good agreement with the DFT reference, demonstrating its superior capability to refine the potential energy surface for complex charged systems.

Having established the efficacy of finetuning for isolated clusters, we extended our investigation to more complex periodic systems to assess the models' ability to capture subtle

surface chemistry. We focused on the refined potential energy surface of an Au₂ dimer adsorbed on both Al-doped and undoped MgO surfaces, using the dataset provided by Ko *et al.* [4]. We found that message-passing-based foundation models, after finetuning, are indeed capable of learning these precise potential energy surfaces. The finetuning protocol was consistent with our previous tests: a subset comprising 20% of the configurations was randomly selected and partitioned into a 4:1 training-to-validation ratio. The models were then trained for 1000 epochs with a 5×10^{-4} learning rate, exclusively minimizing the MSE loss on atomic forces. As summarized in **Table S6**, both finetuned CHGNet and our model successfully reproduce the DFT equilibrium bond lengths on both doped and undoped structures. Nonetheless, our model achieves slightly higher fidelity with the DFT reference. In contrast, the inferior performance of MACE-MP-0 may be attributed to its limited effective receptive field arising from only two message-passing layers, which is likely insufficient to model the complex interactions modulated by the dopant.

Table S6 The equilibrium bond lengths of Au₂ dimer absorbed on both Al-doped and undoped MgO surfaces.

		Au ₂ on Al-doped MgO (Å)	Au ₂ on MgO (Å)
True value of DFT		2.332	2.190
Relative value compared to DFT	4G-HDNNP	+0.010	-0.003
	Bespoke model	-0.002	0.000
	Our foundation model	-0.120	+0.038
	Foundation CHGNet	-0.098	+0.040
	Foundation MACE-MP-0	+0.028	+0.154
	Finetuned our model	+0.002	-0.002
	Finetuned CHGNet	-0.002	-0.004
	Finetuned MACE-MP-0	-0.068	+0.034

In summary, our work establishes a powerful two-step approach: first, pretraining on a universal dataset to build a robust foundation model, and second, performing targeted, data-efficient finetuning to achieve *ab initio* accuracy for specific applications. The success of this second step is, however, contingent on the inherent capabilities of the model’s architecture. A successful foundation model must not only learn general physics but also possess the necessary descriptors and structural flexibility—such as handling net charge and capturing long-range effects—to be effectively specialized for the diverse and challenging problems in modern materials science.”

And the physical test of bespoke model of NaCl cluster is also provided in **SI**,

“**Fig. S1 b**, Potential energy surfaces of the $\text{Na}_{8/9}\text{Cl}_8^+$ cluster system, comparing DFT results with our model predictions. The surfaces were generated by varying the position of the first Na atom relative to the second Na atom as shown in dashed lines, with Na and Cl atoms represented in yellow and green, respectively.”

As for the physical test of Au_2 dimer absorption of the bespoke model, it has been already provided in **Table S6**. All the bespoke models achieve near-perfect agreements with DFT.

We demonstrated that our model is able to learn the correct underlying physics like in charged NaCl clusters rather than simply fitting the data through increased expressivity. We are grateful to the reviewer for pushing us to perform these experiments, which have significantly strengthened the conclusions of our manuscript.

“Additionally, while **Table 1** shows the model’s performance on the 4G dataset, the comparison is made only against a basic bespoke model from the original benchmark. Given the field’s rapid progress, this is no longer a sufficient baseline. For instance, the recent model by Kim *et al.* (<https://arxiv.org/abs/2412.15455>) demonstrates significantly better performance for some systems (e.g., an energy error of 0.073 meV/atom and force error of 0.008 eV/Å for the gold dimer on MgO). A fair evaluation should include comparisons to such modern long-range models.”

Authors’ reply:

We thank the reviewer for raising this important point. We agree that a fair evaluation requires benchmarking against the state-of-the-art, and the comparison against a basic bespoke model is no longer sufficient. The work by Kim *et al.* is indeed a strong and relevant baseline.

To provide a more rigorous and contemporary evaluation, we have conducted benchmark comparisons against the long-range capable model from Kim *et al.* (referred to as cace-lr) on Ko *et al.* datasets. For the $\text{Ag}_3^{+/-}$ and Au_2 -MgO systems, the polarizable charge equilibration (PQEq) parameters for Ag and Au were reoptimized, allowing the model to learn the specific long-range interactions more accurately. The results of this direct comparison are summarized in **Table S10**. In summary, the new benchmarks demonstrate that our methodology is highly competitive with the very recent reference although using the PQEq parameters invariant to geometries. Future work could focus on two key avenues: first, developing dynamic models where PQEq parameters become a function of the local atomic environment, and second,

exploring non-charge-equilibration frameworks, such as those proposed by Kim *et al.*, to capture long-range physics with potentially greater efficiency and expressive power.

In SI,

“**Table S10** Comparison of our bespoke models and cace-lr models.

Dataset		cace-lr	This work
$C_{10}H_2/C_{10}H_3^+$	Energy (meV/atom)	0.73	0.44
	Force (eV/Å)	0.037	0.023
$Na_{8/9}Cl_8^+$	Energy (meV/atom)	0.21	0.16
	Force (eV/Å)	0.010	0.005
$Ag_3^{+/-}$	Energy (meV/atom)	0.162	4.87
	Force (eV/Å)	0.029	0.028
Au_2-MgO	Energy (meV/atom)	0.073	0.13
	Force (eV/Å)	0.008	0.006

Table S11 Final PQEq parameters used for Ag and Au in the bespoke models for this section.

Atom	χ^0 (eV)	η^0 (eV)	Z	Radius (Å)	K_s (kcal/mol/Å ²)
Ag	3.50747	4.80698	1.00000	1.17602	221.26430
Au	5.76708	6.91695	1.00000	0.55600	740.00000

”

“In summary, while the manuscript has improved, the main claims still require more rigorous and convincing validation. I would strongly recommend the authors perform standard physicality tests and include comparisons to other state-of-the-art universal models to support their conclusions.”

Authors’ reply:

We sincerely thank the reviewer for their constructive feedback throughout this process. We fully agree that the main claims of a manuscript require rigorous and convincing validation. In response to the reviewer’s strong recommendation, we have performed an extensive series of experiments to systematically address these points. We are confident that these additions now provide the robust validation that was requested. Specifically, we have addressed the reviewer’s two primary recommendations including physical test and comparison with other universal models as follows.

1. **Fundamental interactions.** We validated that our foundation model possesses the transferability to describe correct long-range physics, successfully capturing potential energy surfaces for both dimer interactions and systems with different charge states, as described in the first question.
2. **Challenging validations.** We then subjected our model to the challenging tests from the Ko *et al.* dataset. Through finetuning, we demonstrated that our model, unlike others, can learn the subtle differences in the potential energy surfaces of the $Na_8Cl_8^+$ and $Na_9Cl_8^+$ clusters, proving it correctly interprets the system’s physics, as described in the second question.

3. **Unique polarization test.** We introduced a novel physicality test that other universal models typically cannot address: response to an external electric field. Our model successfully reproduces the DFT potential energy surface under varying field strengths, demonstrating a genuine understanding of polarization.

In summary, we have systematically performed the requested physical tests and comparisons with other universal models. These new results provide robust and convincing validation for our main claims. We are grateful to the reviewer for guidance, which has significantly strengthened the manuscript.

For the polarization test, details are provided in **SI**,

“...The results are presented in the **Fig. S9**, which plots the system energy as a function of field strength. Our model demonstrates good agreement with the DFT reference, accurately reproducing the energy curve across the entire field range from -2 to 2 V/Å. Notably, the performance of our foundation model is on par with that of FIREANN [19], a state-of-the-art model that was specifically trained on this water system. This highlights the transferability of our model, as it correctly captures complex polarization physics without explicit training for this task. In contrast, the CHGNet and MACE-MP-0 models are entirely unresponsive to the field, predicting a constant energy. This represents a fundamental failure to capture one of the most essential physical interactions, underscoring the superior predictive power and physical realism of our approach.

Fig. S9 Evaluation of polarizable interactions in the periodic water model. The plot compares the performance of our model against DFT [19] (reference), FIREANN [19], CHGNet, and MACE-MP-0.”

Response to the reviewers

We are sincerely grateful to the two reviewers for their valuable time and insightful feedback during this round of review. We have carefully considered all the comments and have provided detailed, point-by-point responses in this document. The modifications to the manuscript are highlighted in **dark red**.

Table of Contents

Response to reviewer #2	2
Response to reviewer #3	3

Response to reviewer #2

“Remarks to the Author:

This revision of the manuscript has brought further improvements and the authors have now clarified most pertinent points raised by the reviewers. The manuscript is publishable.”

Authors’ reply:

We would like to express our gratitude for your positive and encouraging feedback on our revised manuscript. We are very pleased to hear that you found our revisions effective and that the manuscript is now considered publishable. Your expert comments and guidance have been invaluable in improving the quality of our work, and we appreciate the time and effort you dedicated to the review process.

“Remarks on code availability:

I haven’t run the code but the repository exists with examples and instructions.”

Authors’ reply:

We appreciate you taking the time to check our code repository and for confirming that the examples and instructions are available.

Response to reviewer #3

“The manuscript has improved substantially compared to the previous version. The results now look mostly solid and trustworthy, and many of my earlier concerns have been addressed. I only have a few remaining comments, including one point that still puzzles me.”

Authors’ reply:

We sincerely thank you for your thoughtful and constructive feedback on our revised manuscript. We are very encouraged to hear that you find it has improved substantially and that our results are now mostly solid and trustworthy. We have carefully noted your remaining comments and will address them as follows.

“Major comment:

Physical decay of long-range interactions (**Fig. S8**). I recommend plotting this figure on a log–log scale in order to assess whether the correct power-law decay of interactions is reproduced. From visual inspection, it seems that the extracted exponent p is too small; for Coulombic interactions, one expects $p = 1$. Could the authors comment on this discrepancy, especially given the claim that their model reproduces correct long-range physics?”

Authors’ reply:

We thank the reviewer for this insightful comment and valuable suggestion. The reviewer is correct that a log-log plot is the most appropriate way to analyse the power-law decay of long-range interactions. Following this recommendation, we have generated a new log-log plot, now included in the new **Fig. S9**. The reviewer correctly observed that the interaction does not follow a simple $1/r$ decay across the entire distance range. This is expected in the distance range we plotted in **Fig. S8**, as contributions from higher-order electrostatic terms cannot be ignored. To show that our method captures the correct asymptotic behaviour, we extended the distance to more than 16 \AA as shown in **Fig. S9**. Indeed, the curve from our model is tangent to the line with slope = -1 , confirming that it successfully reproduces the correct $p = 1$ power-law decay in the long-range limit.

In **Supplementary Information**, we have included the log-log plot of our model in extended distance range,

“**Fig. S9** A log-log plot showing the interaction energy as a function of distance for an OH-OH dimer in an ionic state. The data from our model (solid green line) is compared with a reference line representing a pure $1/r$ decay (dashed black line).”

“Minor comments:

1. Fine-tuning strategy (**S11**). The additional comparison is very helpful and the results look convincing. Could the authors comment on their fine-tuning strategy in more detail? For example, is this full fine-tuning, multi-head fine-tuning, or a parameter-efficient method such as LoRA? In this context, I am also curious about the performance of the fine-tuned models on the original training tasks—did the authors observe catastrophic forgetting?”

Authors’ reply:

We sincerely thank the reviewer for their positive feedback on our supplementary results and for asking these insightful questions regarding our finetuning strategy and the issue of catastrophic forgetting.

Regarding our finetuning strategy, we adopted “full finetuning” approach. For our model and CHGNet, we unfroze and updated the parameters of the final two convolutional layers and the subsequent Multilayer Perceptron (MLP) layers. For the MACE-MP model, we finetuned all model parameters, following the officially recommended practice for this architecture. Our decision to employ these methods, rather than using multi-head finetuning or parameter-efficient techniques such as LoRA, was driven by the high-precision nature of the Ko *et al.* dataset. We hypothesized that a more flexible approach—achieved by unfreezing some convolutional and MLP layers—was necessary to provide the models with sufficient capacity to capture these subtle but critical features of the potential energy surfaces with high fidelity.

It is important to clarify that our primary goal in finetuning the foundation model on these challenging systems was to demonstrate its inherent capability to capture these complex interactions. That said, the high-precision nature of the potential energy surfaces in Ko *et al.* datasets meant that we did observe a degree of catastrophic forgetting. However, the extent of this effect was dependent on the similarity between the finetuning system and the domains covered in the original training data. For the Au₂-MgO system, the task involves modelling a surface system, which does not include isolated, charged clusters. This represents a relatively modest domain shift from our original MPtrj dataset. Consequently, we found that our model’s performance on the original test set is acceptable after finetuning (MAE for forces and stresses were 0.87 eV/Å and 0.64 GPa on the MPtrj test set, respectively). Conversely, the Na_{8/9}Cl₈⁺ systems represent a significant departure from our original MPtrj dataset, as they involve isolated, charged clusters with unique electrostatic characteristics. Finetuning on this domain, which is substantially different, led to more pronounced catastrophic forgetting. Based on this analysis, we conclude that for highly distinct systems like these charged clusters, the most reliable path to achieving a highly accurate description is to train a bespoke model.

In the **Supplementary Information**, we included details about the finetuning strategy,

“...Regarding our finetuning strategy, we adopted a full finetuning approach. For our model and CHGNet, only the final two convolutional layers, and the multilayer perceptron were updated. For MACE-MP-0, the entire model was subjected to finetuning, as implemented in its own code...”

“2. Comparison table (S10). Thank you for adding this comparison. The results look strong and convincing. It also appears that the Ko dataset is becoming a widely used benchmark. I would like to point out another recent paper (<https://arxiv.org/abs/2507.19382>) which further raises the benchmark results; it may be worth discussing this in the context of related work.”

Authors’ reply:

We thank the reviewer for their positive feedback and for bringing the recent and highly relevant LOREM paper [<https://arxiv.org/abs/2507.19382>] to our attention. We agree that its methodology which treats charges as equivariant tensors provides an important contrast to other long-range approaches. Since the LOREM framework also does not require a charge equilibration process, we have integrated a description of this preprint into our manuscript’s **Discussion** section, highlighting it among other methods that bypass this step.

In the **main text**,

“...LOREM [84] models charges as equivariant tensors to capture orientation-dependent interactions beyond the standard cutoff radius...”

84. Rumiantsev, E., et al., *Learning Long-Range Representations with Equivariant Messages*. arXiv preprint [arXiv:2507.19382](https://arxiv.org/abs/2507.19382), 2025.”

Response to reviewer #3

“I thank the authors for their revision and improvements. I am happy with the manuscript and support publication in its current form.”

Authors' reply:

We sincerely thank the reviewer for their positive feedback and for endorsing the publication of our manuscript in its current form. We are grateful for their time and evaluation.

The authors present a Machine Learning (ML) potential where the short-range graph neural network prediction is combined with the long-range electrostatic and dispersion interactions. While similar approaches were proposed previously, the novelty of this manuscript is in using the polarizable charge equilibration (PQEq) method instead of the charge equilibration (QEq) method. In addition, the authors do not fit the partial charges and thus avoid problems with charge partitioning schemes. Tackling long-range interactions with ML potentials is an active research area, and this manuscript presents an important contribution to the field. Nevertheless, the authors should address the following points before publication:

1. In message-passing graph neural networks, the graph cut-off should not be confused with the cut-off of classical force fields since each message-passing/interaction layer increases the receptive field. The authors use a NequIP model with six equivariant layers. Thus, the effective cut-off is not 5 Å but can extend up to 6 x 5 Å. This point should be made clear in the text, i.e., in the introduction where the authors discuss the "MLIPs with a cut-off of around 5 Å" and in the results section when the authors are comparing their results with strictly local ML potentials such as HDNNP, where the employed cut-off for the symmetry functions is the same as the effective cut-off.
2. Related to the above, the test cases developed by Ko et al. involve very small systems and make sense for strictly local ML potentials but make little sense with a 6-layer graph neural network (GNN), where the receptive field likely encompasses the whole system. To identify the PQEq contribution, the authors should extend Table 1 to include results using only GNN and using GNN+ DFT-D3. On the other hand, a comparison with 2G-HDNNP and 3G-HDNNP is not necessary since these results were already published.
3. It is unclear if the resulting partial charges should be interpreted as physical partial charges or a quantity correlated with the partial charge. The authors should report the errors for partial charge prediction for Ko et al. cases. Of course, the agreement cannot be as good as when training on a partial charge, and given the differences between the charge partitioning schemes, this is also not the goal. Nevertheless, the authors should compare the partial charge prediction errors with differences between charge partitioning schemes to clarify the meaning of predicted partial charges.
4. For the universal machine learning potential model with and without polarizable long-range interactions, the energy/force/stress errors on the test

set are very similar. The authors should include the results of the universal model without polarizable long-range interactions also for the reported applications, i.e., Mechanical properties, Ionic diffusivity etc.

5. Figure 2 b-c: the authors should increase the x-axis range to 12Å to actually see the long-range scaling behavior.
6. Are the model training details described in the Methods section the same for the universal equivariance neural network potential and the test cases developed by Ko et. al? If not, please add training details for Ko et. al cases.
7. "... MPtrj Dataset sourced from Materials Projects as the pretraining dataset..."
Why do you call this pretraining? which dataset was used in the subsequent training?
8. why is the shell charge of all atoms set to 1?
9. missing directly related reference: [10.48550/arXiv.2501.19179](https://arxiv.org/abs/10.48550/arXiv.2501.19179)
10. the authors should relate their work with alternative long-range approaches not employing the charge equilibration schemes, e.g., Ewald summation, Latent Ewald Summation, LSR-MP, NeuralP3M, and So3krates